# An information-theoretic quantification of the content of communication between brain regions

**Marco Celotto**
University Medical Center Hamburg-Eppendorf
Istituto Italiano di Tecnologia
University of Bologna
marco.celotto@iit.it

**Jan Bím**
Datamole,
Prague

**Alejandro Tlaie**
Istituto Italiano di Tecnologia

**Vito De Feo**
University of Essex

**Alessandro Toso**
University Medical Center
Hamburg-Eppendorf

**Stefan M. Lemke**
University of North Carolina

**Daniel Chicharro**
City, University of London

**Hamed Nili**
University Medical Center
Hamburg-Eppendorf

**Malte Bieler**
University College Kristiania

**Tobias H. Donner**
University Medical Center
Hamburg-Eppendorf

**Ileana L. Hanganu-Opatz**
University Medical Center
Hamburg-Eppendorf

**Andrea Brovelli**
Aix Marseille Université
CNRS

**Stefano Panzeri**
University Medical Center Hamburg-Eppendorf
Istituto Italiano di Tecnologia
s.panzeri@uke.de

## Abstract

Quantifying the amount, content and direction of communication between brain regions is key to understanding brain function. Traditional methods to analyze brain activity based on the Wiener-Granger causality principle quantify the overall information propagated by neural activity between simultaneously recorded brain regions, but do not reveal the information flow about specific features of interest (such as sensory stimuli). Here, we develop a new information theoretic measure termed Feature-specific Information Transfer (FIT), quantifying how much information about a specific feature flows between two regions. FIT merges the Wiener-Granger causality principle with information-content specificity. We first derive FIT and prove analytically its key properties. We then illustrate and test them with simulations of neural activity, demonstrating that FIT identifies, within the total information propagated between regions, the information that is transmitted about specific features. We then analyze three neural datasets obtained with different recording methods, magneto- and electro-encephalography, and spiking activity, to demonstrate the ability of FIT to uncover the content and direction of information flow between brain regions beyond what can be discerned with traditional analytical methods. FIT can improve our understanding of how brain regions communicate by uncovering previously unaddressed feature-specific information flow.

37th Conference on Neural Information Processing Systems (NeurIPS 2023).

# 1    Introduction

Cognitive functions, such as perception and action, emerge from the processing and routing of information across brain regions [10, 48, 56, 57, 41]. Methods to study within-brain communication [11, 12, 51] are often based on the Wiener-Granger causality principle, which identifies propagation of information between simultaneously recorded brain regions as the ability to predict the current activity of a putative receiving region from the past activity of a putative sending region, discounting the self-prediction from the past activity of the receiving region [21, 61]. While early measures implementing this principle, such as Granger causality [51], capture only linear interactions, successive information theoretic measures (the closely-related Directed Information [35] and Transfer Entropy [50]) are capable of capturing both linear and nonlinear time-lagged interactions between brain regions [6, 7, 58]. Using such measures has advanced our understanding of brain communication [7, 9, 12, 31, 54, 55, 59, 44]; however, they are designed to capture only the overall information propagated across regions, and are insensitive to the content of information flow. Assessing the content of information flow, not only its presence, would be invaluable to understand how complex brain functions, involving distributed processing and flow of different types of information, arise.

Here, we leverage recent progress in Partial Information Decomposition (PID; [62, 32]) to develop a new non-negative measure (Feature-specific Information Transfer; FIT) that quantifies the directed flow of information about a specific feature of interest between neural populations (Fig. 1A). The PID decomposes the total information that a set of source variables encodes about a specific target variable into components representing shared (redundant) encoding between the variables, unique encoding by some of the variables, or synergistic encoding in the combination of different variables. FIT isolates features-specific information flowing from one region to another by identifying the part of the feature information encoded in the current activity of the receiving region that is shared (redundant) with information present in the past activity of the sending region (because a piece of transmitted information is first found in the sender and then in the receiver) and that is new and unique with respect to the information encoded in the past activity of the receiver (because information already encoded would not have come from the sender).

We first mathematically derive a definition of FIT based on PID. We then use it to demonstrate, on simulated data, that it is specifically sensitive to the flow of information about specific features, correctly discarding feature-unrelated transmission. We then demonstrate that FIT is able to track the feature-specific content and direction of information flow using three different types of simultaneous multi-region brain recordings (electroencephalography - EEG, magnetoencephalography - MEG, and spiking activity). We also address how introducing appropriate null hypotheses and defining conditioned versions of FIT can deal with potential confounding factors, such as the time-lagged encoding of information in two regions without actual communication between them.

## 2    Defining and Computing Feature-specific Information Transfer (FIT)

We consider two time-series of neural activity $X$ and $Y$ simultaneously recorded from two brain regions over several experimental trials. $X$ and $Y$ might carry information about a feature $S$ varying from trial to trial, e.g. a feature of a sensory stimulus or a certain action. The activity measured in each region, $X$ and $Y$, may be any type of brain signal, e.g. the spiking activity of single or multiple neurons, or the aggregate activity of neural populations, such as EEG or MEG. We call $Y_{pres}$ the activity of $Y$ at the present time point $t$ , and $X_{past}$ and $Y_{past}$ the past activity of $X$ and $Y$ respectively (Fig. 1). Established information theoretic measures such as TE [50] use the Wiener-Granger principle to quantify the overall information propagated from a putative sender $X$ to a putative receiver $Y$ as the mutual information $I$ between the receiver's present neural activity $Y_{pres}$ and the sender's past activity $X_{past}$, conditioned on the receiver's past activity $Y_{past}$ (Fig. 1):

$$TE(X \rightarrow Y) = I(X_{past}; Y_{pres}|Y_{past}) \tag{1}$$

(see Supplementary Material, SM1.1 for how TE depends on probabilities of past and present activity). TE captures the overall information propagated across regions but lacks the ability to isolate information flow about specific external variables. To overcome this limitation, here we define FIT, which quantifies the flow of information specifically about a feature $S$ from a putative sending area $X$ to a putative receiving area $Y$ (Fig. 1A). We define FIT using the PID [62]. PID decomposes the joint mutual information $I(S; \underline{X})$, that a set of $N$ source variables $\underline{X} = (X_1, X_2, ..., X_N)$ carries

about a target variable $S$, into non-negative components called information atoms (see SM1.2). For $N = 2$, PID breaks down the joint mutual information $I(S; X_1, X_2)$ into four atoms: the Shared (or redundant) Information $SI(S : X_1, X_2)$ that both $X_1$ and $X_2$ encode about $S$; the two pieces of Unique Information about $S$, $UI(S : X_1 \setminus X_2)$ and $UI(S : X_2 \setminus X_1)$, provided by one source variable but not by the other ; and the Complementary (synergistic) information about $S$, $CI(S : X_1, X_2)$, encoded in the combination of $X_1$ and $X_2$. Several measures have been proposed to quantify information atoms [62, 5, 18, 30]. Here we use the measure $I_{min}$ originally defined in [62], which guarantees non-negative values for information atoms for any $N$ (see SM1.2).

Using $I_{min}$, the Shared Information that $X_1$ and $X_2$ carry about $S$ is defined as follows:

$$SI(S : X_1, X_2) = \sum_{s \in S} p(s) \min_{X_i \in \{X_1, X_2\}} I(S = s; X_i) \tag{2}$$

where $I(S = s; X_i)$ is the specific information that source $X_i$ carries about a specific outcome of the target variable $s \in S$, and is defined as:

$$I(S = s; X_i) = \sum_{x_i \in X_i} p(x_i|s) \Big[ \log \frac{p(s|x_i)}{p(s)} \Big] \tag{3}$$

Intuitively, the shared information computed as in eq. 2 quantifies redundancy as the similarity between $X_1$ and $X_2$ in discriminating individual values of the feature $S$. In the general case of $N$ source variables, information atoms are hierarchically ordered in a lattice structure, and $I_{min}$ can be used to quantify any atom in the decomposition (including the Unique and Complementary information atoms introduced above for the case $N = 2$; see SM1.2).

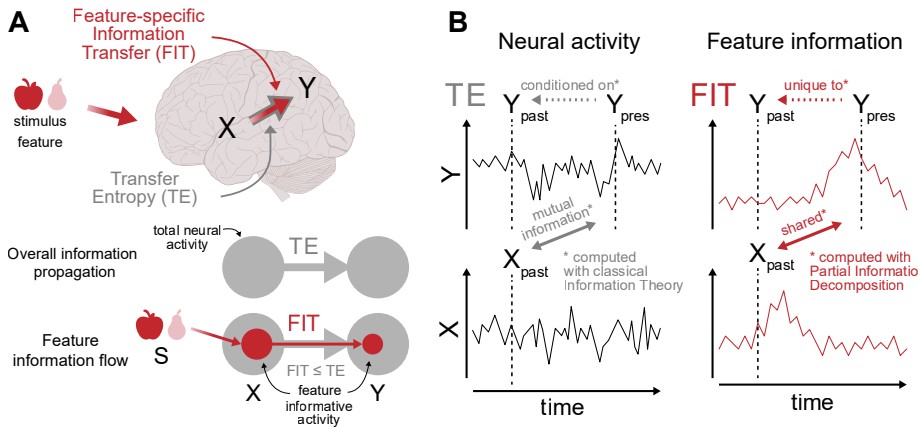

Figure 1: Sketch of FIT and TE. (A) TE is the established information-theoretic measure to quantify the overall information propagated between two simultaneously recorded brain regions $X$ (sender) and $Y$ (receiver). FIT measures the information flowing from $X$ to $Y$ about the stimulus feature S. (B) TE and FIT incorporate content-unspecific and content-specific versions of the Wiener-Granger causality principle. TE is the mutual information between the past activity of $X$ and the present activity of $Y$ conditioned on the past of $Y$. FIT is the feature information in the present of $Y$ shared with the past information of $X$ and unique with respect to the past information of $Y$.

We wanted FIT to measure the directed flow of information about $S$ between $X$ and $Y$, rather than the overall propagation of information measured by TE (Fig. 1A). We thus isolated the information about a feature $S$ in the past of the sender $X$ that $Y$ receives at time $t$. Because of the Wiener-Granger causality principle, such information should not have been present in the past activity of the receiver $Y$. Therefore, we performed the PID in the space of four variables $S$, $X_{past}$, $Y_{past}$, and $Y_{pres}$ to compute information atoms that combine Shared, Unique and Complementary Information carried by three sources about one target [62]. One natural candidate atom to measure FIT is the information about $S$ that $X_{past}$ shares with $Y_{pres}$ and is unique with respect to $Y_{past}$: $SUI(S : X_{past}, Y_{pres} \setminus Y_{past})$ (Fig. 1B; Fig. S1B). This atom is defined as the difference between the shared information that the two source variables $X_{past}$ and $Y_{pres}$ carry about $S$, and the shared information that the three source variables $X_{past}$, $Y_{pres}$ and $Y_{past}$ carry about $S$. Redundancy

can only decrease when adding more sources. Hence by removing the information that is also redundant with $Y_{past}$, $SUI(S : X_{past}, Y_{pres} \setminus Y_{past})$ quantifies a non-negative component of shared information between $X_{past}$ and $Y_{pres}$ about $S$ that is unique with respect to $Y_{past}$. Importantly, using unique information to remove the feature information in $Y_{past}$ is more conservative than conditioning on $Y_{past}$ as in TE [27] . $SUI(S : X_{past}, Y_{pres} \setminus Y_{past})$ intuitively captures what we are interested in, and satisfies two desirable mathematical properties: it is upper bounded by the feature information encoded in the past of $X$ ($I(S; X_{past})$) and in the present of $Y$ ($I(S; Y_{pres})$). This is because the PID defines redundancy between source variables as sub-components of the information about the target carried by each of the sources (see SM1.3.1). However, the information atom $SUI(S : X_{past}, Y_{pres} \setminus Y_{past})$ has two undesirable properties. The first is that its value can exceed the total amount of information propagated from $X$ to $Y$ (TE). This can happen since the unique information in the PID decomposition is a component of the conditional mutual information about the target. However, the target in $SUI(S : X_{past}, Y_{pres} \setminus Y_{past})$ is the feature $S$, which means that this atom is not constrained to be smaller than the TE, which is independent of $S$ (see eq. 1 and SM1.3.4). This property is undesirable, because the overall information propagation must be an upper bound to the information transmitted about a specific feature. The second is that by construction (see SM1.3.1) this atom depends on $X_{past}, Y_{pres}, S$ only through the pairwise marginal distributions $P(X_{past}, S)$ and $P(Y_{pres}, S)$, but not through the marginal distribution $P(X_{past}, Y_{pres})$, which implies that this atom by itself cannot identify confounding scenarios where both sender and receiver encode feature information at different times with no transmission taking place (see SM1.3.1).

To address these limitations, following [42] we considered the alternative PID taking $S$, $Y_{past}$, and $X_{past}$ as source variables and $Y_{pres}$ as a target. In this second PID (Fig. S1B), the atom that intuitively relates to FIT is $SUI(Y_{pres} : X_{past}, S \setminus Y_{past})$, the information about $Y_{pres}$ that $X_{past}$ shares with $S$ that is unique with respect to $Y_{past}$. While being intuitively similar to $SUI(S : X_{past}, Y_{pres} \setminus Y_{past})$, $SUI(Y_{pres} : X_{past}, S \setminus Y_{past})$ has $Y_{pres}$ as target variable and hence is upper bounded by TE (but not by $I(S; X_{past})$) and depends on the pairwise marginal distribution $P(X_{past}, Y_{pres})$ (see SM1.3.2). Thus, this second atom has useful properties that complement those of the first atom. Importantly, Shannon information quantities impose constraints that relate PID atoms across decompositions with different targets. We [42] demonstrated that, for PID with $N = 2$ sources, these constraints reveal the existence of finer information components shared between similar atoms of different decompositions. Here, we extended this approach (see SM1.3.3) to $N = 3$ sources and demonstrated that the second atom is the only one in the second PID that has a pairwise algebraic relationship with the first atom, indicating that these atoms share a common, finer information component. Therefore we defined FIT by selecting this finer common component by taking the minimum between these two atoms:

$$FIT = \min[SUI(S : X_{past}, Y_{pres} \setminus Y_{past}), SUI(Y_{pres} : X_{past}, S \setminus Y_{past})] \qquad (4)$$

With this definition, FIT is upper bounded by $I(S; X_{past})$, by $I(S; Y_{pres})$ and by $TE(X \to Y)$. That FIT satisfies such bounds is essential to interpret it as transmitted information. If FIT could be larger than the feature information encoded by sender $X$ or receiver $Y$, or than the total information transmitted ($TE(X \to Y)$), then FIT could not be interpreted as feature information transmitted from $X$ to $Y$. Additionally, FIT depends on the joint distribution $P(S, X_{past}, Y_{pres})$ through all the pairwise marginals $P(S, X_{past})$, $P(S, Y_{pres})$, and $P(X_{past}, Y_{pres})$, implying that it can rule out, using appropriate permutation tests, false-communication scenarios in which $X$ and $Y$ encode the stimulus independently with a temporal lag, without any within-trial transmission (see SM1.3.4).

Note that the definition of FIT holds when defining present and/or past activity as multidimensional variables, potentially spanning several time points. However, use of multidimensional neural responses requires significantly more data for accurate computation of information. For this reason, following [7, 39, 38], in all computations of TE and FIT we computed the present of $Y$ at a single time point $t$ and the past of $X$ and of $Y$ at individual time points lagged by a delay $\delta$: $X_{past} = X_{t-\delta}$ and $Y_{past} = Y_{t-\delta}$. Note also that in all calculations of FIT and TE, we estimated probabilities from empirical occurrences after discretizing both features and neural activities. SM1.6 reports details of the procedure and Table S1 the number of bins used for each analysis. Simulations of accuracy of these estimates as function of the data size are reported in SM2.5 and Fig. S6.

## 3 Validation of FIT on simulated data

To test the ability of FIT to measure feature-specific information flow between brain regions, we performed simulations in scenarios of feature-related and feature-unrelated information transfer.

We performed (Fig. 2A-B) a simulation (details in SM2.1) in which the encoded and transmitted stimulus feature $S$ was a stimulus-intensity integer value (1 to 4) . The activity of the sender $X$ was a two-dimensional variable with one stimulus-feature-informative $X_{stim}$ and one stimulus-uninformative component $X_{noise}$. The stimulus-feature-informative dimension had temporally-localized stimulus-dependent activity from 200 to 250ms and had multiplicative Gaussian noise (similar results were found with additive noise, see SM2.1 and Fig. S3). The stimulus-unrelated component was, at any time point, a zero-mean Gaussian noise. The activity of the receiver $Y$ was the weighted sum of $X_{stim}$ and $X_{noise}$ with a delay $\delta$, plus Gaussian noise . The delay $\delta$ was chosen randomly in each simulation repetition in the range 40-60ms. Here and in all simulations, we averaged information values across simulation repetitions we determined their significance via non-parametric permutation tests [33, 7, 19, 29]. For TE, we permuted $X$ across all trials to test for the presence of significant within-trial transmission between $X$ and $Y$ [7, 33]. For FIT, we conducted two different permutation tests: one for the presence of feature information in $X$ and $Y$ (shuffling $S$ across trials), and another for the contribution of within-trial correlations between $X$ and $Y$ to the transmission of $S$ (shuffling $X$ across trials at fixed stimulus). We set the threshold for FIT significance as the 99th percentile of the element-wise maximum between the two permuted distributions (see SM1.7).

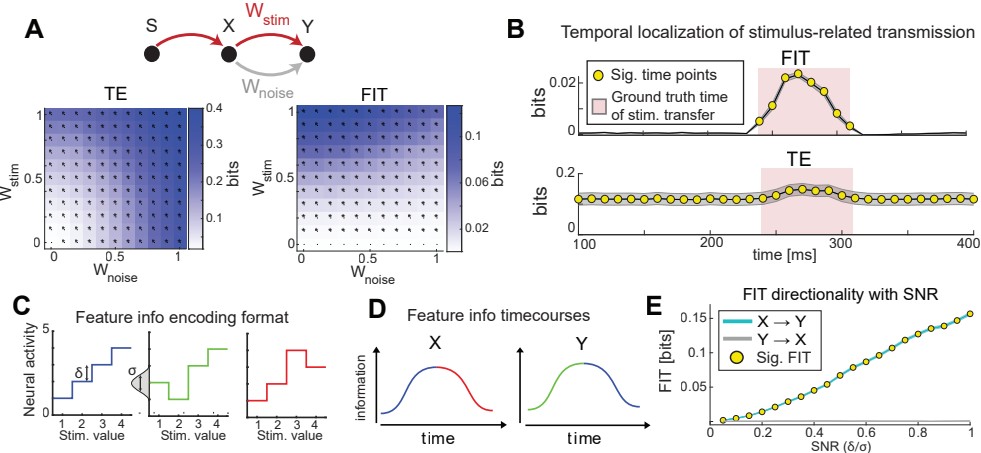

Figure 2: Testing FIT on simulated data. (A) FIT and TE as function of stimulus-feature-related ($w_{stim}$) and -unrelated ($w_{noise}$) transmission strength. * indicate significant values ($p < 0.01$, permutation test) for the considered parameter set. (B) Dynamics of FIT and TE in a simulation with time-localized stimulus-feature-information transmission. Red area shows the window of stimulus-feature-related information transfer. Results plot mean (lines) and SEM (shaded area) across 50 simulations (2000 trials each). (C) Different neural encoding functions used for the simulations in panels D-E. $\delta$: separation of responses to different features; $\sigma$: Gaussian noise SD. (D) Sketch of simultaneous stimulus feature information profiles. Different types of information content are color-coded. (E) FIT in the $X \rightarrow Y$ (cyan) and $Y \rightarrow X$ (grey) directions as a function of SNR $\delta/\sigma$ (plot in log scale). Results plot mean (lines) and SEM (shaded area) across 100 simulations (2000 trials each). Yellow dots in B, E show points with significant FIT (p < 0.01, permutation test).

We investigated how FIT and TE from $X$ to $Y$ depended on the amount of stimulus-feature-related transmission (increased by increasing $w_{stim}$) and of -unrelated transmission (increased by increasing $w_{noise}$). We report values at the first time point in which information in $Y$ was received from $X$, but similar results hold for later time points. Both FIT and TE increased when increasing $w_{stim}$ (Fig. 2A). However, TE increased with $w_{noise}$ (Fig. 2A), as expected from a measure that captures the overall information propagation. In contrast, FIT decreased when increasing $w_{noise}$, indicating that FIT specifically captures the flow of information about the considered feature.

We then investigated how well TE and FIT temporally localize the stimulus-feature-related information transmitted from $X$ to $Y$ (Fig. 2B). We simulated a case in which stimulus-feature-related information was transmitted from $X$ to $Y$ only in a specific window ($[240, 310]ms$) and computed FIT and TE at each time point (see SM2.1 for details). FIT was significant only in the time window in which $Y$ received the stimulus information from $X$. In contrast, TE was significant at any time point, reflecting that noise was transmitted from $X$ to $Y$ throughout the whole simulation time.

Importantly, FIT can detect feature-specific information flow even when information is encoded in the sender and receiver with an overlapping timecourse (see SM2.2 for details). To illustrate this, we simulated the activity of two regions $X$ and $Y$ encoding an integer stimulus feature $S$ with the same time course of stimulus-feature information (Fig. 2D), but with feature specific transmission taking place only from $X$ to $Y$. Because FIT could correctly detect that the format of information representation of $S$ in the present of $Y$ was equal to that of the past of $X$ but different to that of the past of $Y$ (Fig. 2D), it could correctly detect that feature information flows from $X$ to $Y$ (Fig. 2E).

We also performed simulations to investigate whether the non-parametric permutation test described above can correctly rule out as non-significant feature-specific transmission the scenario in which $X$ and $Y$ independently encode $S$ without actual communication occurring between them. We simulated a scenario in which feature information was encoded with a temporal lag in $X$ and $Y$, with no transmission from $X$ to $Y$. We found that the resulting values were always non-significant (see SM2.6.1 and Fig. S7C) when tested against a surrogate null-hypothesis distribution (pairing $X$ and $Y$ in randomly permuted trials with the same feature) that destroy the within-trial communication between $X$ and $Y$ without changing the feature information encoded in $X$ and $Y$ (see SM1.7). Importantly, this null hypothesis also ruled out false communication scenarios where the measured FIT and TE were only due to the presence of instantaneous mixing of sources (see SM 2.7).

Finally, we addressed how to remove the confounding effect of transmission of feature information to $Y$ not from $X$ but from a third region $Z$. In Granger causality or TE analyses, this is addressed conditioning the measures on $Z$ [1, 37]. In an analogous way, we developed a conditioned version of FIT, called cFIT (see SM1.4), which measures the feature information transmitted from $X$ to $Y$ that is unique with respect to the past activity of a third region $Z$. We tested its performance in simulations in which both $X$ and $Z$ transmitted feature information to $Y$ and found that cFIT reliably estimated the unique contribution of $X$ in transmitting feature information to $Y$, beyond what was transmitted by $Z$ (see SM2.6.2 and Fig. S7D).

# 4  Analysis of real neural data

We assessed how well FIT detects direction and specificity of information transfer in real neural data.

## 4.1  Flow of stimulus and choice information across the human visual system

We analyzed a previously published dataset ([63], see also SM3.1 for details) of source-level MEG data recorded while human participants performed a visual decision-making task. At the beginning of each trial, a reference stimulus was presented (contrast 50%), followed by a test stimulus that consisted of a sequence of 10 visual samples with variable contrasts (Fig. 3A). After the test stimulus sequence, participants reported their choice of whether the average contrast of the samples was greater or smaller than the reference contrast. The previous study on these data ([63]) analyzed the encoding of stimulus and choice signals in individual areas, but did not study information transfer. We focused on gamma-band activity (defined as the instantaneous power of the 40-75Hz frequency band), because it is the most prominent band for visual contrast information encoding [24, 45, 17] and information propagation [7, 3] in the visual system. Previous work has demonstrated that gamma-band transmission is stronger in the feedforward (from lower to higher in the visual cortical hierarchy) than in the feedback (from higher to lower in the visual cortical hierarchy) direction [55, 3, 36], suggesting that gamma is a privileged frequency channel for transmitting feedforward information. However, these previous studies did not determine the content of the information being transmitted.

To address this question, we quantified FIT in a network of three visual cortical areas (Fig. 3B) that we selected because they encoded high amounts of stimulus information and because they were sufficiently far apart ($\geq 2.8$cm) to minimize leakage in source reconstruction (see SM3.1.1 Fig. S9) [63, 23]. The areas, listed in order of position, from lower to higher, in the cortical hierarchy were: primary visual cortex (V1), area V3A, and area LO3. Because participants made errors (behavioral performance was 75% correct), in each trial the stimulus presented could differ from the participant's choice. We thus assessed the content of the information flow by computing FIT about either the sensory stimulus ($FIT_S$; using as feature the mean contrast across all 10 visual samples) or the reported choice ($FIT_C$), in each instant of time in the $[-100, 500]$ms peri-stimulus time window (because stimulus information was higher in the first 500ms post-stimulus, see SM3.1.4 and Fig. S9) and across a range of putative inter-area delays $\delta$. In Fig. 3C we show the resulting $FIT_S$

time-delay information maps for the example pair of regions V1 and V3A. A cluster-permutation analysis [34, 14] revealed significant feedforward stimulus-specific information transmission from V1 to V3A (but no significant feedback from V3A to V1) localized 200-400ms after the stimulus onset, with an inter-area communication delay between 65 and 250ms (see SM3.1.4 and Fig. S9).

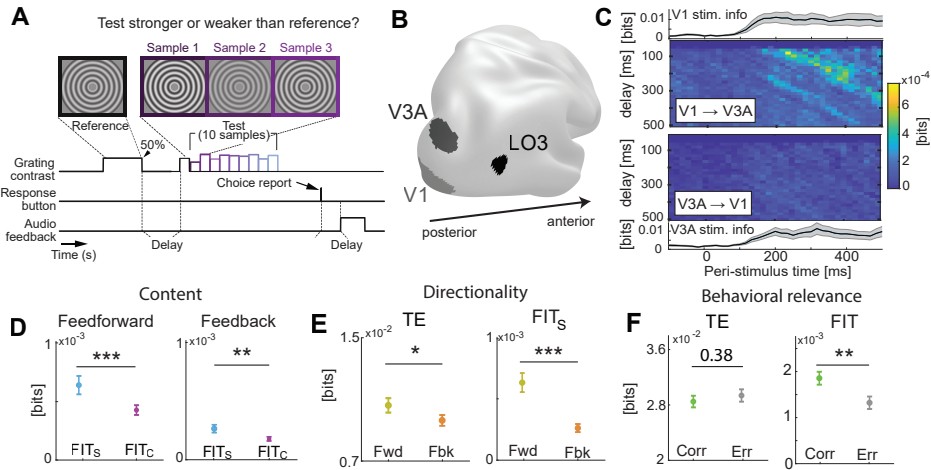

Figure 3: Information flow across the human visual hierarchy with MEG. (A) Sketch of task. (B) Cortical surface map of the location of the three considered visual regions. (C) Temporal profiles of stimulus information and time-delay stimulus FIT maps for an example regions pair (V1, V3A). Top to bottom: stimulus information in V1; time-delay FIT map in the feedforward (V1 → V3A) direction; time-delay FIT map in the feedback (V3A → V1) direction; stimulus information in V3A. (D) Comparison between FIT about stimulus ($FIT_S$) and FIT about choice ($FIT_C$) in the visual network in the feedforward (left) and feedback (right) directions. (E) Comparison between feedforward and feedback transmission in the network for TE (left) and stimulus FIT (right). (F) Same as E but for feedforward transmission on correct vs error trials. In all panels, lines and image plots show averages and errorbars SEM across participants, experimental sessions and regions pairs (in case of FIT and TE) or regions (in case of mutual information). *: p<0.05, **: p<0.01, ***: p<0.001. Information-theoretic quantities were computed from the gamma band ([40-75]Hz) power of source-level MEG, first computed separately for the left and right hemisphere and then averaged.

We compared properties of overall information propagation (computed with TE) and feature-specific information flow (computed with FIT) across all pairs of areas within the considered visual cortical network. To determine the prevalent content of information flow in the network, we compared $FIT_S$ and $FIT_C$ transmitted in the feedforward and in the feedback directions (Fig. 3D). Gamma-band transmitted more information about the stimulus than about choice (i.e. $FIT_S > FIT_C$) in both the feedforward ($p < 10^{-3}$ two-tailed paired t-test) and in the feedback ($p < 0.01$ two-tailed paired t-test) direction, with a larger difference for the feedforward direction. This result is supported by simulations where we show that, in presence of multiple simultaneously transmitted features, FIT ranks correctly the features about which most information is transmitted (see SM2.3 and Fig. S4). Thus, we focused on stimulus-specific information flow in the following FIT analyses. We then studied the leading direction of information flow. Both the total amount of information propagation (TE) and the stimulus-specific information flow ($FIT_S$) were larger in the feedforward than in the feedback direction (Fig. 3E), but with a larger effect for $FIT_S$ ($p < 10^{-6}$ two-tailed paired t-test) compared to TE ($p < 0.05$ two-tailed paired t-test). Together, these results show that gamma-band activity in the visual system carries principally information about the stimulus (rather than choice) and propagates it more feedforward than feedback.

We next assessed the behavioral relevance of the feedforward stimulus information transmitted by the gamma band. A previous study showed that the overall (feature-unspecific) strength of feedforward gamma band information propagation negatively correlates with reaction times, indicating that stronger feedforward gamma activity propagation favors faster decisions [46]. However, no study has addressed whether stimulus information transmitted forward in the gamma band promotes decision accuracy. We addressed this question by comparing how $FIT_S$ varied between trials when

participants made a correct or incorrect choice (Fig. 3F). We matched the number of correct and error trials to avoid data size confounds [40]. $FIT_S$ in the feedforward direction was significantly lower in error than in correct trials (Fig. 3F, right; $p = 0.001$ two-tailed paired t-test), while TE did not vary between correct and error trials (Fig. 3F, left; $p = 0.38$ two-tailed paired t-test). Feedback information transmission (both in terms of overall information propagation, TE, and stimulus specific information flow, $FIT_S$), did not vary between correct and incorrect trials. This indicates that the feedforward flow of stimulus information, rather than the overall information propagation, is key for forming correct choices based on sensory evidence.

These results provide the new discovery that the gamma band transmits feedfoward stimulus information of behavioral relevance, and highlight the power of FIT in revealing the content and direction of information flow between brain areas.

## 4.2 Eye-specific interhemispheric information flow during face detection

We next tested the ability of FIT to detect feature-specific information flow between brain hemispheres. We analyzed a published EEG dataset recorded from human participants detecting the presence of either a face or a random texture from an image covered by a bubble mask randomly generated in each trial ([47]; see SM3.2.1 for details). Previous analysis of these data [26] showed that eye visibility in the image (defined as the proportion of image pixels in the eye region visible through the mask) is the most relevant image feature for successful face discrimination. It then showed that eye-specific information appears first at ∼120ms post-image presentation in the Occipito-Temporal (OT) region of the hemisphere contra-lateral with respect to the position of the eye, and then appears ∼20-40ms later in the ipsi-lateral OT region (Fig. 4A). However, this study did not determine if the eye information in the ipsi-later hemisphere is received from the contra-lateral hemisphere. To address this issue, we computed FIT transmission of eye-specific information between the Left OT (LOT) and Right OT (ROT) regions (using the electrodes within these regions that had most information as in [26], see SM3.2.2). Left Eye (LE) FIT from the contra- to the ipsi-lateral OT (ROT to LOT) peaked between 150 to 190ms after image onset with transfer delays of 20-80ms (Fig. 4B). Right eye (RE) FIT from the contra- to the ipsi-lateral OT (LOT to ROT) peaked with similar times and delays (Fig. 4C). Both contra-to-ipsi-lateral LE and RE had statistically significant FIT peaks in the time-delay maps (cluster-permutation analysis, p < 0.01; see SM3.2.2 and Fig. S10). Thus, FIT determined the communication window for contra-lateral flow of eye-specific information with high precision.

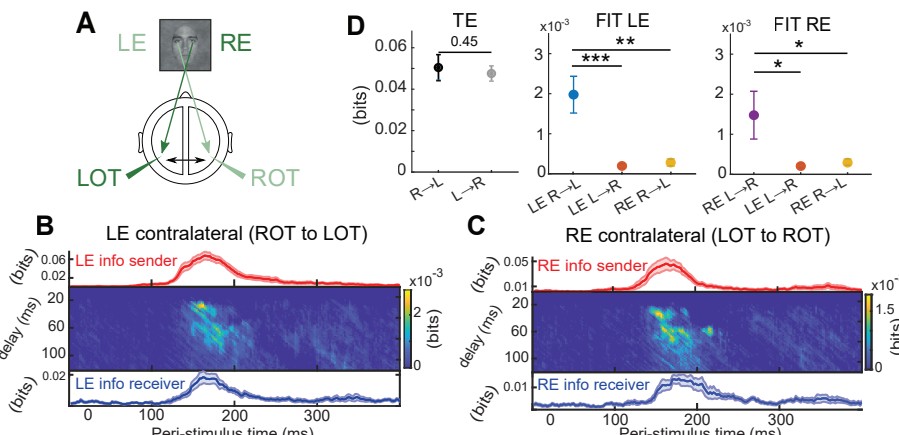

Figure 4: Inter-hemispheric eye-specific information flow during face detection using human EEG. (A) Schematic of the putative information flow. LOT (ROT) denote Left (Right) occipito-temporal regions. LE (RE) denote the Left (Right) Eye visibility feature. (B) Information (lines) carried by the EEG in each region, and FIT (image plot) about LE contra-lateral transfer. (C) Same as B for RE. (D) Contra- to ipsi-lateral vs ipsi- to contra-lateral transfers for TE and FIT for both LE and RE. Dots and images plot averages and errorbars plot SEM across participants (N=15).

To gain further insight about the directionality and feature-specificity of the information flow across hemispheres, we compared FIT and TE across transfer directions and/or eye-specific visibility

conditions (Fig. 4D, middle and right). Right-to-left LE FIT was significantly larger than left-to-right LE FIT ($p < 0.001$ two-tailed paired t-test) or right-to-left RE FIT ($p < 0.01$ two-tailed paired t-test). Left-to-right RE FIT was significantly larger than right-to-left RE FIT ($p < 0.05$ two-tailed paired t-test) or left-to-right LE FIT ($p < 0.05$ two-tailed paired t-test). In contrast, we found no significant difference between directions for the overall propagated information (TE), Fig. 4D, left). Thus, the use of FIT revealed a temporally localized flow of eye information across hemispheres that was feature-selective (i.e. about mainly the contra-lateral eye) and direction-specific (contra-to-ipsi-lateral), without direction specificity in the overall information propagation (TE) across hemispheres.

Finally, to more tightly localize the origin of eye-specific contra-lateral information flow, we asked whether the contra-lateral OT electrodes selected in our analyses were the sole senders of inter-hemispheric eye-specific information. We used the conditioned version of FIT, cFIT, to compute the amount of transfer of eye information from the contra- to the ipsi-lateral OT after removing the effect of eye-specific information possibly routed through alternative sending locations (see SM3.4). We found (Fig. S12A) that the effect we measured with FIT was robust even when conservatively removing with cFIT the information that could have been routed through other locations.

### 4.3 Stimulus-specific information flow in a thalamocortical network

We finally used FIT to measure the feature- and direction-specificity of information flow in the thalamocortical somatosensory and visual pathways. We analyzed a published dataset in which multi-unit spiking activity was simultaneously recorded in anaesthetized rats from the primary visual and somatosensory cortices, and from first-order visual and somatosensory thalamic nuclei ([8], see SM3.3.1 for details), during either unimodal visual, unimodal tactile, or bimodal (visual and tactile) stimulation. This analysis tests FIT on another major type of brain recordings (spiking activity). Moreover, due to the wealth of knowledge about the thalamocortical network [53, 16], we can validate FIT against the highly-credible predictions that information about basic sensory features flows from thalamus to cortex, and that somatosensory and visual pathways primarily transmit tactile and visual information, respectively. Using FIT, we found (see SM3.3.3 and Fig. S11) that sensory information flowed primarily from thalamus to cortex, rather than from cortex to thalamus. We also found that the feedforward somatosensory pathway transmits more information about tactile- than about visually-discriminative features, and that the feedforward visual pathway transmits more information about visually- than tactile-discriminative features. Importantly, TE was similarly strong in both directions, and when considering tactile- or visually-discriminative features. This confirms the power of FIT for uncovering stimulus-specific information transfer, and indicates a partial dissociation between overall information propagation and neural transfer of specific information.

## 5 Comparison with previously published measures

We finally examine how FIT differs from alternative methods for identifying components of the flow of information about specific features. We focus on measures that implement the Wiener-Granger discounting of the information present in the past activity of the sender. Other methods, that do not implement this (and thus just correlate past information of the sender with present information of the receiver), erroneously identify information already encoded in the past activity of the receiver as information transmitted from a sender (see SM4.3).

A possible simple proxy for identifying feature-specific information flow is quantifying how the total amount of transmitted information (TE) is modulated by the feature [7]. For the case of two feature values, this amounts to the difference of TE computed for each individual value. We show in SM4.1, using simulations, that this measure can fail in capturing feature-related information flow even in simple scenarios of feature information transmission. Additionally, when tested on MEG data, it could not assess the directionality of information transmission within brain networks (see SM4.1).

A previous study [25] defined a measure, Directed Feature Information (DFI), which computes feature-specific information redundant between the present activity of the receiver and the past activity of the sender, conditioned on the past activity of the receiver. However, DFI used a measure of redundancy that conflated the effects of redundancy and synergy (see SM1.5). Because of this, DFI is, both on real and on simulated data, often negative and thus not interpretable as measure of information flow (see SM4.2). In contrast, FIT is non-negative and uses PID to consider only redundant information between sender and receiver, as appropriate to identify transmission of information. Moreover,

because DFI discounts only past activity of the sender rather than its feature-specific information, it was less precise, conservative and sensitive in localizing direction and timing of feature-specific information flow (as shown in SM4.2 and Fig. S15 with simulated and real data).

Finally, a study defined feature-specific information using PID in the space of four variables $S$, $X_{past}$, $Y_{past}$, and $Y_{pres}$ [4]. However, this measure was not upper bounded by either feature information encoded in the past of the sending region or the total information flowing between regions.

## 6 Discussion

We developed and validated FIT, an information theoretic measure of the feature-specific information transfer between a sender $X$ and a receiver $Y$. FIT combines the PID concepts of redundancy and uniqueness of information [62] with the Wiener-Granger causality principle [11] to isolate, within the overall transmitted information (TE), the flow of information specifically related to a feature $S$.

The strengths and limitations of FIT as a neural data analysis tool stem from those of information theory for studying neural information processing. Information theory has led to major advances to neural coding because of its ability to capture linear and non-linear interactions at all orders making little assumptions [43, 15]. This is important because deviations from linearity and order of interactions vary in often unknown ways between brain areas, stimulus types and recording modalities [13, 28, 20]. Using such a general formalism avoids potentially biasing results with wrong assumptions. However, the price to pay for the fact that information theory includes full probability distributions is that it is data hungry. While our definitions of FIT and cFIT are straightforwardly valid for multivariate analyses including conditioning on the information of multiple regions [37] (as in cFIT) or obtaining more conservative estimates of information transmission on which information in the receiver $Y$ is requested to be unique with respect to the information of the sender and receiver at multiple past time points [58], for data sampling reasons in practice in real data these analyses are confined to conditioning to one region or a single past time point [7, 39, 38]. In future work, we aim to make FIT applicable to analyses of multiple regions or time points coupling it with advanced non-parametric [49] methods to robustly estimate its multivariate probability distributions.

The generality of our approach lends itself to further developments. Importantly, we defined FIT directly at the level of PID atoms. This means that, although here we implemented FIT using the original definition of redundancy in PID [62] because it has the advantage of being non-negative for all information atoms, FIT can be easily implemented also using other PID redundancy measures [22, 5, 2, 18, 30] with complementary advantages and disadvantages (see SM1.2).

We individuate two directions for improvement. First, even though the surrogate permutation test we developed to assess FIT significance provided reasonable results on real data and worked work well also with artefacts due to instantaneous mixing of sources (see SM2.7), further research is needed to generate more refined surrogate data generation techniques to rule out more conservatively false feature-specific communication scenarios [52]. Second, how best to select the time of past activity to compute FIT, an issue of significance in Wiener-Granger measures [60], remains to be determined.

To demonstrate the properties of FIT, we performed numerical simulations in different communication scenarios and compared FIT against TE (Fig. 2). These simulations confirmed that TE effectively detected the overall propagation of information, but it did not detect the flow of feature-specific information. FIT, in contrast, reliably detected feature- and direction-specific information flow with high temporal sensitivity. We confirmed the utility of FIT in applications to neural data. In three brain datasets spanning the range of electrophysiological recordings (spiking activity, MEG and EEG), FIT credibly determined the directionality and feature specificity of information flow. Importantly, in most of these datasets this happened in the absence of variations in the overall information propagation between the same brain regions (measured with TE). The partial dissociation between overall activity flow and feature-specific flow found consistently in simulations and data has important implications. First, it highlights the need of introducing a specific measure of feature information transfer such as FIT, as it resolves question unaddressed by content-unspecific measures. Second, it establishes that measuring feature-specific components of information flow between brain regions is critical to go beyond the measurement of overall neural activity propagation and uncover aspects of cross-area communication relevant for ongoing behavior. Thus, as methods to record from multiple brain regions rapidly advance, FIT is well suited to uncover fundamental principles in how brain regions communicate.

# 7 Acknowledgements

This research has received funding from the European Union's Horizon 2020 Framework Programme for Research and Innovation under the Specific Grant Agreement No. 945539 (Human Brain Project SGA3), from the NIH Brain Initiative (grants U19 NS107464, R01 NS109961, R01 NS108410), and from the Simons Foundation (SFARI Human Cognitive and Behavioral Science grant 982347). We thank G. Bondanelli and G.M. Lorenz for useful discussion and feedback on the manuscript.

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

# Supplementary Material for *An information-theoretic quantification of the content of communication between brain regions*

## SM1 Definitions, derivations, and properties of the information theoretic quantities

In this section, we first define the basic information theoretic quantities that we use in the paper. We next introduce basic concepts of the PID theory needed for our derivations. We then use these concepts to derive a mathematical definition of FIT and then prove some of its key mathematical properties.

### SM1.1 Definition of the Shannon information quantities used in this study

In the following we describe and provide analytical expression for the quantities of Shannon's information theory that we used in this paper. These are the mutual information between the stimulus and the neural activity, and the Transfer Entropy (TE), which are estimated based on the probabilities of activities of neural signals $X$ and $Y$ and of stimulus features $S$.

The mutual information $I(S; X_{pres})$ between the stimulus feature $S$ and the neural activity $X_{pres}$ of $X$ at the present time is a non-parametric measure that quantifies the full single-trial statistical relationship between $S$ and $X_{pres}$. It captures the effect of all linear and nonlinear interactions between these variables. It is defined as follows [35]:

$$I(S; X_{pres}) = \sum_{s, x_{pres}} p(s, x_{pres}) \log \frac{p(s, x_{pres})}{p(s)p(x_{pres})} \tag{S1}$$

where $p(s, x_{pres})$ is the joint probability, sampled across experimental trials, of observing in a given trial the joint occurrence of stimulus feature value $s \in S$, and activity $x_{pres} \in X_{pres}$. The sum spans all possible events. $I(S; X_{pres})$ is non-negative, and it is zero if and only if $S$ and $X_{pres}$ are independent. Similar expressions and properties hold for the information $I(S; X_{past})$, $I(S; Y_{past})$ between the stimulus and the past activity $X_{past}$, $Y_{past}$ of $X$ and $Y$, respectively.

TE [34] is an information theoretic measure that utilizes the Wiener-Granger principle to quantify the overall propagation of information by neural activity from a putative sender $X$ to a putative receiver $Y$ as the mutual information between the present neural activity of the receiver $Y_{pres}$ and the past activity of the sender $X_{past}$, conditioned upon the past activity of the receiver $Y_{past}$. The expression of TE as a function of the joint probability distribution $P(X_{past}, Y_{pres}, Y_{past})$ is as follows:

$$TE(X \to Y) = I(X_{past}; Y_{pres}|Y_{past})$$
$$= \sum_{x_{past}, y_{pres}, y_{past}} p(x_{past}, y_{pres}, y_{past}) \log \frac{p(x_{past}, y_{pres}|y_{past})}{p(x_{past}|y_{past})p(y_{pres}|y_{past})} \tag{S2}$$

where $p(x_{past}, y_{pres}, y_{past})$ is the joint probability, sampled across experimental trials, of observing the joint occurrence of $x_{past} \in X_{past}$, $y_{pres} \in Y_{pres}$, and $y_{past} \in Y_{past}$, and the sum spans all possible events. Importantly, TE does not depend on the stimulus feature $S$ and thus cannot tell how much of the overall information being transmitted from $X$ to $Y$ is about $S$ or about other factors unrelated to $S$.

37th Conference on Neural Information Processing Systems (NeurIPS 2023).

**SM1.2 Elements of the PID theory**

PID was introduced first in Ref [38] and is a very active field of research [21]. To make our paper self-standing, here we briefly summarize the basic concepts of PID that are most needed for our reasoning and derivations.

In the general case of $N$ source variables $\underline{X} = (X_1, \ldots, X_N)$, PID dissects the joint mutual information that the source variables jointly carry about a target variable $T$, $I(\underline{X}; T)$, into non-overlapping pieces of redundant, unique, and synergistic information. Let $A_1, \ldots, A_M$ be all the non-empty and potentially overlapping subsets of $\underline{X}$, that we call *sources* in the following. PID considers the collections of sources $\alpha \in P_1(P_1(\underline{X}))$, where $P_1(\underline{X})$ denotes the set of all non-empty subsets of $\underline{X}$. That is, a collection $\alpha$ corresponds to a non-empty subset of *sources*, namely to a non-empty subset of non-empty subsets of source variables. In the following, for brevity, we will call *collections* the collections of sources. Collections $\alpha$ are indicated using a bracketed notation (e.g., $\alpha = \{X_1 X_2\}\{X_1 X_3\}$ represents the collection of the two overlapping sources $\{X_1 X_2\}$ and $\{X_1 X_3\}$). Importantly, pieces of unique and synergistic information can be defined and computed algorithmically once the redundant information is identified and computed. Thus, in what follows we focus principally on defining and computing redundancies. For each $\alpha$, PID defines the amount of information about $T$ that is redundant between all sources in the collection: $I_\cap(T; \alpha)$. Conceptually, the redundancy of any collection $\alpha$ for which a source $A_i \in \alpha$ is a subset of another source $A_j \in \alpha$ ($i \neq j$) should be equal to the redundancy of the same collection after removing the superset $A_j$ [38]. Therefore, the collections of interest to compute $I_\cap(T; \alpha)$ are only those for which no source is a superset of any other, and hence removing any source $A_i \in \alpha$ could potentially reduce the redundancy. These collections form a domain called $\mathcal{A}(\underline{X})$:

$$\mathcal{A}(\underline{X}) = \{\alpha \in P_1(P_1(\underline{X})) : \forall A_i, A_j \in \alpha, A_i \nsubseteq A_j\} \tag{S3}$$

It is possible to define a partial order over the collections of $\mathcal{A}(\underline{X})$. A collection $\alpha$ precedes another collection $\beta$ if for each source B in $\beta$ it exists a source A in $\alpha$ that is a subset of B, formally:

$$\forall \alpha, \beta \in \mathcal{A}(\underline{X})(\alpha \preceq \beta \Leftrightarrow \forall B \in \beta \, \exists A \in \alpha \,|\, A \subseteq B) \tag{S4}$$

Applying the order relationship in eq. S4 to the elements of $\mathcal{A}(\underline{X})$ produces redundancy lattices, in which a collection that succeeds $\alpha$ provides at least as much redundant information about $T$ as $\alpha$ [38] (see Fig. S1A,B for the lattices for $N = 2$ and $N = 3$ source variables). PID allows quantifying the amount of redundant information $I_\partial(T; \alpha)$ that a specific collection $\alpha$ contributes to the joint mutual information about T, and that is not already redundant in any collections preceding $\alpha$ (in the following, we will call $I_\partial(T; \alpha)$ the *information atom* provided by collection $\alpha$). $I_\partial(T; \alpha)$ is implicitly defined by the following relationship [38]:

$$I_\cap(T; \alpha) = \sum_{\beta \preceq \alpha} I_\partial(T; \beta) \tag{S5}$$

Due to the so-called self-redundancy axiom of the PID theory [38], if an individual source $A_i$ appears in collection $\alpha$, the redundancy computed on collection $\alpha$ is equal to the mutual information between all source variables in $A_i$ and the target variable $T$:

$$I_\cap(T; \alpha) = I_\cap(T; \{A_i\}) = I(T; A_i) \tag{S6}$$

By combining eqs. S5 and S6 we can write Shannon information theoretic quantities as the sum of partial information atoms:

$$I(T; A_i) = \sum_{\beta \preceq A_i} I_\partial(T; \beta) \tag{S7}$$

Eq. S7 will be fundamental to provide upper bounds for FIT in terms of Shannon information quantities. When applied to the trivariate system $(S, X_1, X_2)$, taking $S$ as target and $(X_1, X_2)$ as source variables, eq. S7 provides the decomposition of the joint mutual information $I(S; X, Y)$ that we discussed in the main text:

$$I(S; X_1, X_2) = I_\partial(S; \{X_1\}\{X_2\}) + I_\partial(S; \{X_1\}) + I_\partial(S; \{X_2\}) + I_\partial(S; \{X_1 X_2\}) \tag{S8}$$

where in the main text we called $I_\partial(S; \{X_1\}\{X_2\}) = SI(S : X_1, X_2)$, $I_\partial(S; \{X_1\}) = UI(S : X_1 \setminus X_2)$, $I_\partial(S; \{X_2\}) = UI(S : X_2 \setminus X_1)$, $I_\partial(S; \{X_1 X_2\}) = CI(S : X_1, X_2)$ to improve clarity

for readers not familiar with PID. $SI$ is shorthand for Shared (that is, redundant) Information; $UI$ is short-hand for Unique information; $CI$ is shorthand for Complementary (that is, synergistic) information.

Thus far we covered elements of PID theory that hold for a generic redundancy measure $I_\cap$, but did not discuss how to compute $I_\cap(T; \alpha)$ for a specific collection $\alpha$. Several measures of redundant information have been proposed [16; 3; 11; 20], in this work we use the original measure $I_{min}$ from Williams and Beer [38], as it has the fundamental property of being non-negative for any information atoms for any number $N$ of source variables (not only for $N = 2$) (for a proof, see Appendix D of Ref [38]). The redundant information $I_{min}$ for a collection $\alpha$ is defined as follows:

$$I_{min}(T; \alpha) = \sum_{t \in T} p(t) \min_{A_i \in \alpha} I(T = t; A_i) \tag{S9}$$

where $I(T = t; A_i)$ is the specific information that source $A_i$ carries about a specific outcome of the target variable $t \in T$, and is defined as:

$$I(T = t; A) = \sum_a p(a|t) \left[ \log \frac{p(t|a)}{p(t)} \right] \tag{S10}$$

Intuitively, $I_{min}$ quantifies redundancy as the overlap between the sources in the distributions of specific information across individual values of target variable. This corresponds to quantifying the degree to which all sources in collection $\alpha$ are similarly discriminative about individual values of the target. We decided to use $I_{min}$ because of its advantages in terms of being defined for an arbitrary number of source variables $N$ (something that is needed because FIT is defined in terms of $N = 3$ source variables and cFIT in terms of $N = 4$ source variables) and being non-negative for all atoms (which is important to guarantee that FIT is interpretable as a measure of information transmission). Importantly, similarly to other redundancy measures [20], $I_{min}$ satisfies the *pairwise marginals* property, meaning that $I_{min}(T; \alpha)$ only depends on the pairwise marginals distributions $p(T, A_i)$ between the target $T$ and each source $A_i \in \alpha$.

Alternative redundancy measures proposed so far are either not straightforward to generalize beyond $N = 2$ source variables [3; 16] or can provide negative information atoms [11]. However, these alternative measures have complementary advantages with respect to $I_{min}$, such as satisfying the identity property $I_\cap(X, Y; (X, Y)) = I(X; Y)$ which guarantees that, in a system made of two independent variables, the two variables cannot carry redundant information about the whole system. Despite not satisfying this property, the $I_{min}$ measure has been applied to study information processing in simulated neural networks [2], providing insightful and interpretable results.

## SM1.3 Derivation of FIT

In this subsection, we derive the definition of FIT. In the main text, we used the notation $SUI(S : X_{past}, Y_{pres} \setminus Y_{past})$ to denote the atom of information that is shared by variables $X_{past}$ and $Y_{pres}$ about target $S$ but is unique with respect to a third variable $Y_{past}$. Using the bracketed notation introduced in Section SM1.2 to denote information atoms, $SUI(S : X_{past}, Y_{pres} \setminus Y_{past})$ corresponds to $I_\partial(S; \{X_{past}\}\{Y_{pres}\})$. From eq. S5, this atom is the difference between the information that $X_{past}$ and $Y_{pres}$ share about $S$ minus the information that $X_{past}$, $Y_{past}$ and $Y_{pres}$ share about $S$ (see also eq. S14). In the PID literature, information redundant in set of sources about a target that is not redundant with information from another set, has been termed *shared unique* information [6; 23]. Therefore, using the bracketed notation to denote the two atoms of shared unique information $SUI(S : X_{past}, Y_{pres} \setminus Y_{past})$ and $SUI(Y_{pres} : X_{past}, S \setminus Y_{past})$, we can write the definition of FIT as:

$$FIT = min[I_\partial(S; \{X_{past}\}\{Y_{pres}\}), I_\partial(Y_{pres}; \{X_{past}\}\{S\})] \tag{S11}$$

We first discuss the mathematical properties of $I_\partial(S; \{X_{past}\}\{Y_{pres}\})$, the *first atom* appearing in FIT definition. We then discuss the complementary mathematical properties met by $(I_\partial(Y_{pres}; \{X_{past}\}\{S\}))$, the *second atom* appearing in FIT definition. To keep our reasoning as general as possible, we discuss properties of the atoms that are valid when the atom is computed using any of the redundancy measures that satisfy the pairwise marginal property (which include the $I_{min}$ redundancy measure that we implemented here). We then demonstrate that a specific pairwise

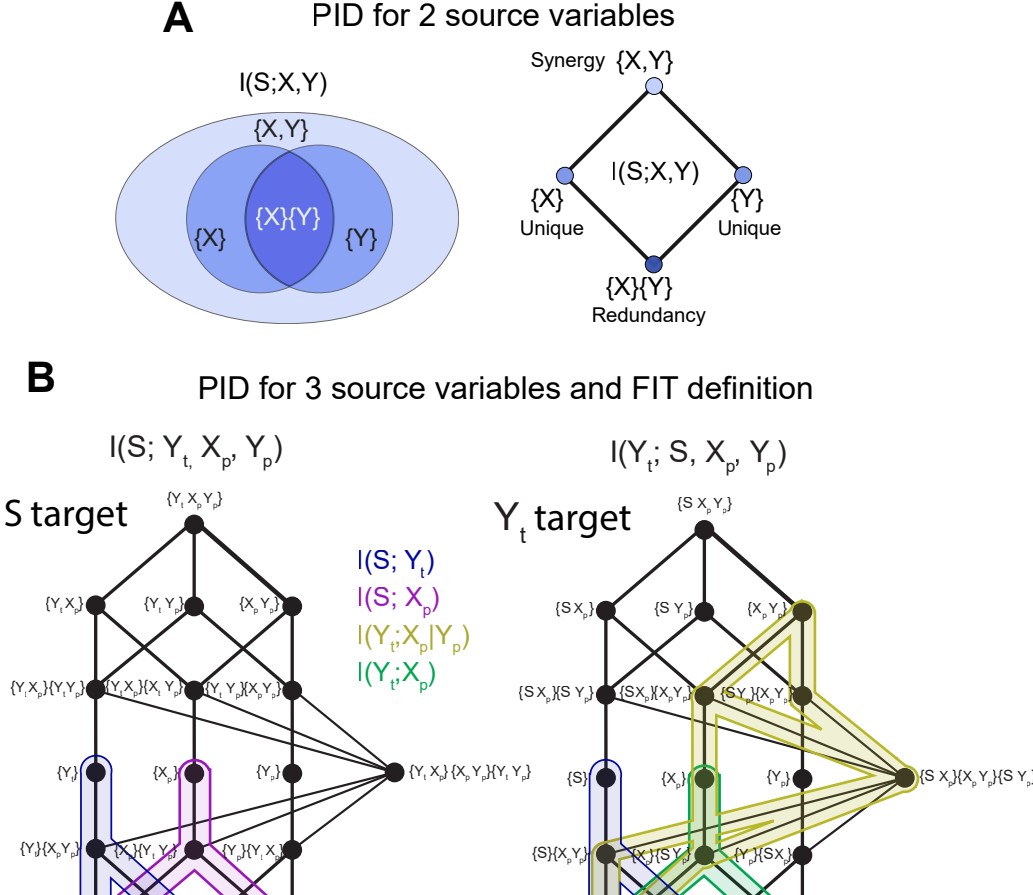

Figure S1: Schematic of the concepts of PID. (A) The information $I(S; X, Y)$ that two source variables $X, Y$ carry about a target variable $S$ can be decomposed into four PID atoms. Left: a set-theoretic diagram of the decomposition. Shared information $\{X\}\{Y\}$, darkest shade of blue; unique information $\{X\}$ and $\{Y\}$, lighter shade of blue; synergistic information $\{X, Y\}$, lightest blue. Right: the same decomposition plotted as lattice. A link between two regions symbolizes the ordering relationship of eq. S4. (B) FIT is defined on two PID lattices with three sources and one target. Left: The PID lattice with $S$ as target and $(X_{past}, Y_{past}, Y_{pres})$ as sources. Right: the PID with $Y_{pres}$ as target and $(S, X_{past}, Y_{past})$ as sources. FIT is the minimum between the two atoms highlighted in red. Classical Shannon information theoretic quantities are mapped on the two lattices with different colors (i.e. the sum of all the atoms bounded by a given color is equal to a classical information-theoretic quantity). $I(S; Y_{pres})$ is mapped using blue, $I(S; X_{past})$ using purple, $TE(X \rightarrow Y)$ using yellow, and $I(X_{past}; Y_{pres})$ using green. The $p$ and $t$ subscript in the Figure is a shorthand for *past* and *pres* respectively.

algebraic relationship exists between these two atoms. This relationship is derived from the Shannon information theoretic quantities that relate atoms in the two decompositions. Importantly, this relationship uncovers the presence of a more refined information component that is shared between the two atoms. Finally, we discuss how taking the minimum between these two atoms ensures that FIT fulfills simultaneously a series of fundamental properties, including being upper bounded at the same time by the feature information encoded in the past activity of the sender $X$, $I(S; X_{past})$, and

in the present activity of the receiver $Y$, $I(S; Y_{pres})$, and by the total information flowing from $X$ to $Y$, namely $TE(X \rightarrow Y)$.

### SM1.3.1 Properties of the first atom in the FIT definition

Our intuitive definition is that FIT should be the information shared between the past activity of a sender region $X_{past}$ and the present activity of a receiver region $Y_{pres}$ about $S$ that is unique with respect to the past activity of the receiver $Y_{past}$. Thus, within PID of the $(S, X_{past}, Y_{pres}, Y_{past})$ system the most natural candidate is the first atom in eq. S11 ($I_{\partial}(S; \{X_{past}\}\{Y_{pres}\})$) coming from the decomposition taking $(X_{past}, Y_{past}, Y_{pres})$ as source variable and $S$ as target variable. Using eq. S7, we show that the two Shannon information quantities $I(S; X_{past})$ and $I(S; Y_{pres})$ (i.e., the feature information encoded in the past values of the sender $X$ and of the receiver $Y$, respectively) set an upper bound on $I_{\partial}(S; \{X_{past}\}\{Y_{pres}\})$. Indeed, $I(S; X_{past})$ and $I(S; Y_{pres})$ can be written as the sum of information atoms appearing on the lattice having $S$ as target:

$$\begin{aligned}
I(S; X_{past}) = I_{\partial}(S; \{X_{past}\}\{Y_{pres}\}\{Y_{past}\}) + I_{\partial}(S; \{X_{past}\}\{Y_{past}\}) \\
+ I_{\partial}(S; \{X_{past}\}\{Y_{pres}\}) + I_{\partial}(S; \{X_{past}\}\{Y_{past}Y_{pres}\}) \\
+ I_{\partial}(S; \{X_{past}\}) \geq I_{\partial}(S; \{X_{past}\}\{Y_{pres}\})
\end{aligned} \tag{S12}$$

$$\begin{aligned}
I(S; Y_{pres}) = I_{\partial}(S; \{X_{past}\}\{Y_{pres}\}\{Y_{past}\}) + I_{\partial}(S; \{Y_{pres}\}\{Y_{past}\}) \\
+ I_{\partial}(S; \{X_{past}\}\{Y_{pres}\}) + I_{\partial}(S; \{Y_{pres}\}\{Y_{past}X_{past}\}) \\
+ I_{\partial}(S; \{Y_{pres}\}) \geq I_{\partial}(S; \{X_{past}\}\{Y_{pres}\})
\end{aligned} \tag{S13}$$

which proves that $I_{\partial}(S; \{X_{past}\}\{Y_{pres}\})$ is upper bounded by both quantities (see Fig. S1B for a graphical depiction of $I(S; Y_{pres})$, in blue, and $I(S; X_{past})$, in purple, upper bound the first atom). However, eq. S7 does not establish any relationship between $I_{\partial}(S; \{X_{past}\}\{Y_{pres}\})$ and Shannon information between the source variables of the decomposition, including $TE(X \rightarrow Y)$. Therefore, the value of the first atom can exceed the total amount of information transmitted from $X$ to $Y$ $TE(X \rightarrow Y)$.

Next we prove that, when computed using a redundancy measure that satisfies the *pairwise marginals* property (see Section SM1.2), $I_{\partial}(S; \{X_{past}\}\{Y_{pres}\})$ only depends on the probability distribution $P(S, X_{past}, Y_{pres})$ through the pairwise marginal distributions $P(S, X_{past})$ and $P(S, Y_{pres})$, and does not depend explicitly on $P(X_{past}, Y_{pres})$. Indeed, using eq. S5 we can express $I_{\partial}(S; \{X_{past}\}\{Y_{pres}\})$ as the difference between the redundancy about $S$ computed on collection $\{X_{past}\}\{Y_{pres}\}$ minus the redundancy computed on collection $\{X_{past}\}\{Y_{pres}\}\{Y_{past}\}$:

$$I_{\partial}(S; \{X_{past}\}\{Y_{pres}\}) = I_{\cap}(S; \{X_{past}\}\{Y_{pres}\}) - I_{\cap}(S; \{X_{past}\}\{Y_{pres}\}\{Y_{past}\}) \tag{S14}$$

If $I_{\cap}$ satisfies the pairwise marginals property, then the right-hand side of eq. S14 only depends on the full probability distribution $P(S, X_{past}, Y_{past}, Y_{pres})$ through the pairwise marginal distributions between the target $S$ and the individual sources $P(S, X_{past})$, $P(S, Y_{past})$, and $P(S, Y_{pres})$, but not through the pairwise marginals between the sources, including $P(Y_{pres}, X_{past})$. This implies that if we partially disrupt the dependency structure of our data to create surrogate data, where the individual dependencies of $X$ and of $Y$ on $S$ are preserved (i.e., the pairwise marginals $P(S, X_{past})$ and $P(S, Y_{pres})$ do not change) and the within-trial correlations at a fixed stimulus are disrupted (i.e., the conditional distribution $P(X_{pres}, Y_{past}|S)$ changes), this atom will retain the same value it had in the original data. Therefore, this atom alone cannot rule out confounding scenarios where $X$ and $Y$ encode $S$ independently with a temporal lag, with no information transfer at fixed stimulus value.

### SM1.3.2 Properties of the second atom in the FIT definition

Atoms satisfying mathematical properties that are complementary to the ones of $I_{\partial}(S; \{X_{past}\}\{Y_{pres}\})$ exist on the decomposition with $Y_t$ as target. On this decomposition one atom that intuitively captures feature-specific information flow is $I_{\partial}(Y_{pres}; \{X_{past}\}\{S\})$, i.e. the information that the past activity of the sender $X$ and the feature $S$ share about the present activity of the receiver $Y$ that is unique with respect to the past activity of the receiver $Y$. We first prove that this atom is upper bounded by the value of $TE(X \rightarrow Y)$, and then that this atom depends on $P(Y_{pres}, X_{past})$.

To prove that the value of $TE(X \rightarrow Y)$ sets an upper bound to $I_{\partial}(Y_{pres}; \{X_{past}\}\{S\})$, we first use the information-theoretic chain rule [9], to write the conditioned mutual information in eq. S2 as

the difference between the joint mutual information that $X_{past}$ and $Y_{past}$ carry about $Y_{pres}$ minus the mutual information between $Y_{pres}$ and $Y_{past}$. Then, we use eq. S7 to write the two information quantities as the sum of non-negative information atoms, including $I_\partial(Y_{pres}; \{X_{past}\}\{S\})$:

$$\begin{aligned}
TE(X \rightarrow Y) &= I(Y_{pres}; X_{past}, Y_{past}) - I(Y_{pres}; Y_{past}) \\
&= I_\partial(Y_{pres}; \{S\}\{X_{past}\}) + I_\partial(Y_{pres}; \{X_{past}\}\{SY_{past}\}) \\
&\quad + I_\partial(Y_{pres}; \{S\}\{X_{past}Y_{past}\}) + I_\partial(Y_{pres}; \{X_{past}\}) \\
&\quad + I_\partial(Y_{pres}; \{X_{past}S\}\{Y_{past}S\}\{X_{past}Y_{past}\}) \\
&\quad + I_\partial(Y_{pres}; \{X_{past}S\}\{X_{past}Y_{past}\}) + I_\partial(Y_{pres}; \{Y_{past}S\}\{X_{past}Y_{past}\}) \\
&\quad + I_\partial(Y_{pres}; \{X_{past}Y_{past}\}) \geq I_\partial(Y_{pres}; \{S\}\{X_{past}\})
\end{aligned} \tag{S15}$$

which proves that $I_\partial(Y_{pres}; \{X_{past}\}\{S\})$ is upper bounded by $TE(X \rightarrow Y)$ (see Fig. S1B for a graphical depiction of the mapping of $TE(X \rightarrow Y)$, in yellow, on the lattice to which this second atom belongs).

Similarly to the first atom, this second atom is also upper bounded by $I(S; Y_{pres})$ (not proven, but see the blue quantity in Fig. S1B for a graphical depiction of this property), however it is not upper bounded by Shannon information quantities between the source variables of the decomposition with $Y_{pres}$ as target, and in particular by $I(S; X_{past})$. This is important because it proves that neither the second atom alone satisfies all the properties that we require for a measure of feature-specific information transfer.

We then prove that the second atom depends on $P(Y_{pres}, X_{past})$, a property which makes it suited to rule out confounding scenarios where $X$ and $Y$ independently encode the $S$ but no communication occurs between the two. To do so, we use eq. S5 to write the second atom as the difference between two redundancy terms:

$$I_\partial(Y_{pres}; \{X_{past}\}\{S\}) = I_\cap(Y_{pres}; \{X_{past}\}\{S\}) - I_\cap(Y_{pres}; \{X_{past}\}\{S\}\{Y_{past}\}) \tag{S16}$$

If $I_\cap$ satisfies the pairwise marginals property, then the right-hand side. of eq. S14 depends on the full probability distribution $P(S, X_{past}, Y_{past}, Y_{pres})$ through the marginal distributions between the target $Y_{pres}$ and the individual sources $P(Y_{pres}, X_{past})$, $P(Y_{pres}, S)$, and $P(Y_{pres}, Y_{past})$. This implies that if we partially disrupt the dependency structure of our data and create surrogate data where the individual dependencies of $X$ and of $Y$ on $S$ are preserved (i.e., the pairwise marginals $P(S, X_{past})$ and $P(S, Y_{pres})$ do not change) and the within-trial correlations at a fixed stimulus are disrupted (i.e., the conditional distribution $P(X_{pres}, Y_{past}|S)$ changes), the value of the referenced atom may differ from its original value. This change occurs because this operation generally disrupts $P(X_{pres}, Y_{past})$. Therefore, this atom can rule out confounding scenarios where $X$ and $Y$ encode $S$ independently with a temporal lag, with no information transfer at fixed stimulus value.

### SM1.3.3 The two atoms in the FIT definition are related by Shannon Information theoretic quantities

This Section is structured as follows. First, we present some basic findings from Ref. [27] where, the authors showed that atoms from different decompositions are algebraically constrained by Shannon's information-theoretic quantities and used these constraints to identify, specifically for a trivariate system, a reduced set of finer information components which could describe all atoms across different decompositions. Next, we express the algebraic constraints between two decompositions as a homogeneous linear system of equations. We demonstrate that the reduced set of information components derived for two decompositions in Ref. [27] can be obtained as solutions to this homogeneous system. Finally, we derive the analogous homogeneous system in the case of four variables. One solution to this system relates specifically the two atoms appearing in the FIT definition.

Ref [27] showed that atoms belonging to different decompositions are algebraically constrained by information-theoretic quantities. These constraints derive from fundamental axioms of PID theory, specifically the fact that in the system $(X_1, \ldots, X_N)$, we can use eq. S7 to express the mutual information between two variables $X_i$ and $X_j$ (conditioned on up to $N - 2$ other variables) as the sum of information atoms from both the decomposition with $X_i$ and the decomposition with $X_j$ as target variable. For the trivariate system $(S, X, Y)$, Shannon information quantities

impose two linear constraints between the 4 atoms of information having $S$ as target and the 4 atoms of information having $Y$ as target (all atoms are: $I_\partial(S; \{X\}\{Y\})$, $I_\partial(S; \{Y\})$, $I_\partial(S; \{X\})$, $I_\partial(S; \{XY\})$, $I_\partial(Y; \{X\}\{S\})$, $I_\partial(Y; \{S\})$, $I_\partial(Y; \{X\})$, $I_\partial(Y; \{XS\})$):

$$I(S;Y) = I_\partial(S; \{X\}\{Y\}) + I_\partial(S; \{Y\}) = I_\partial(Y; \{X\}\{S\}) + I_\partial(Y; \{S\})$$
$$I(S;Y|X) = I_\partial(S; \{XY\}) + I_\partial(S; \{Y\}) = I_\partial(Y; \{XS\}) + I_\partial(Y; \{S\}) \tag{S17}$$

Combining the two equations in the system of eqs. S17 reveals an equality among the differences in the amount of information carried by pairs of similar atoms (the two redundancies $I_\partial(S; \{X\}\{Y\})$ and $I_\partial(Y; \{X\}\{S\})$, the two synergies $I_\partial(S; \{XY\})$ and $I_\partial(Y; \{XS\})$, and the two unique information $I_\partial(Y; \{S\})$ and $I_\partial(S; \{Y\})$)):

$$I_\partial(S; \{X\}\{Y\}) - I_\partial(Y; \{X\}\{S\})$$
$$= I_\partial(Y; \{S\}) - I_\partial(S; \{Y\}) = I_\partial(S; \{XY\}) - I_\partial(Y; \{XS\}) \tag{S18}$$

Therefore, 6 atoms of the 8 atoms belonging to the two decompositions (those appearing in eq S17) are not independent, while $I_\partial(S; \{X\})$ and $I_\partial(Y; \{X\})$ are independent from all other atoms. In Ref. [27] the authors showed that, due to the two constraints of eq. S17, the 8 atoms can be described by 6 finer independent information components (that they called information *subatoms*). In Ref. [27] they quantify these 6 subatoms as follows: three subatoms are the minimum between pairs of similar atoms belonging to the two decomposition (i.e., the two redundancies $I_\partial(S; \{X\}\{Y\})$ and $I_\partial(Y; \{X\}\{S\})$, the two synergies $I_\partial(S; \{XY\})$ and $I_\partial(Y; \{XS\})$ and the two unique information $I_\partial(S; \{Y\})$ and $I_\partial(Y; \{S\})$); one subatom is equal to the difference between the maximum and the minimum in each of the above pairs (which is equal for the three pairs, see eq. S18); two subatoms are equal to the unconstrained atoms not appearing in eq. S17 ($I_\partial(S; \{X\})$ and $I_\partial(Y; \{X\})$).

A novel perspective on the relationships between the amounts of information carried by specific sets of atoms from different decompositions is to conceptualize the eight atoms as forming an eight-dimensional vector space, $\mathcal{V}$. We can represent a generic column vector in $\mathcal{V}$ as $\underline{v}$ and the $2 \times 8$ matrix of constraints imposed by Shannon information quantities relating the atoms of the two decompositions as $\underline{\underline{B}}$. With these definitions, we can express the system of eqs. S17 as a homogeneous linear system:

$$\underline{\underline{B}}\underline{v} = \underline{0} \tag{S19}$$

Specifically, the coefficients of $\underline{\underline{B}}$ are obtained by taking the difference between the middle- and the right-term in the two eqs. S17. Ordering the dimensions of $\mathcal{V}$ as ($I_\partial(S; \{X\}\{Y\})$, $I_\partial(S; \{Y\})$, $I_\partial(S; \{X\})$, $I_\partial(S; \{XY\})$, $I_\partial(Y; \{X\}\{S\})$, $I_\partial(Y; \{S\})$, $I_\partial(Y; \{X\})$, $I_\partial(Y; \{XS\})$), $\underline{\underline{B}}$ has the following form:

$$\underline{\underline{B}} = \begin{bmatrix} 1 & 1 & 0 & 0 & -1 & -1 & 0 & 0 \\ 0 & 1 & 0 & 1 & 0 & -1 & 0 & -1 \end{bmatrix} \tag{S20}$$

It can easily be verified that, for instance, $I_\partial(S; \{X\}\{Y\}) = I_\partial(Y; \{X\}\{S\})$, is a solution of the homogeneous system in eq. S19 for the matrix $\underline{\underline{B}}$ defined as in eq. S20. Consider a matrix multiplication between $\underline{\underline{B}}$ and the vector $\underline{v}_{SI}$, whose only non-zero components are $I_\partial(S; \{X\}\{Y\})$ and $I_\partial(Y; \{X\}\{S\})$ (i.e., $\underline{v}_{SI} = (I_\partial(S; \{X\}\{Y\}), 0, 0, 0, I_\partial(Y; \{X\}\{S\}), 0, 0, 0)$). This matrix multiplication is equivalent to multiplying the coefficients in columns 1 and 5 of $\underline{\underline{B}}$ by the two atoms, respectively, and doing the element-wise sum of the two resulting vectors. This sum is zero if and only if $I_\partial(S; \{X\}\{Y\}) = I_\partial(Y; \{X\}\{S\})$. Put simply, columns of $\underline{\underline{B}}$ with element-wise opposite coefficients correspond to pairs (or triplets) of atoms that form a solution of eq. S19 when they have equal value. As a result, the following are all nontrivial solutions of the homogeneous system in eq. S19, or equivalently, they belong to the null space of $\underline{\underline{B}}$: $I_\partial(S; \{X\}\{Y\}) = I_\partial(Y; \{X\}\{S\})$, $I_\partial(S; \{Y\}) = I_\partial(Y; \{S\})$, $I_\partial(S; \{XY\}) = I_\partial(Y; \{XS\})$, $I_\partial(S; \{X\})$, $I_\partial(Y; \{X\})$, and $I_\partial(S; \{X\}\{Y\}) = I_\partial(S; \{XY\}) = I_\partial(Y; \{S\})$. These solutions uncover specific relationships between pairs or triplets of atoms across different decompositions. Importantly, considering that the eight atoms are not independent and can be represented by six finer, independent quantities, these solutions lend support to the notion that these finer components of information are shared among the atoms linked by a single solution. Remarkably, the atoms identified as related by a solution precisely correspond to the six subatoms previously defined in Ref [27].

Similar to the case of $N = 2$ source variables, for $N = 3$ source variables, there are 36 information atoms (18 per lattice) that belong to two decompositions with different targets. Shannon information

quantities impose constraints relating these 36 atoms, implying the existence of finer information components (or subatoms) that can describe the two decompositions even when there are $N = 3$ source variables. Our goal here is not to uncover the complete set of components that describe all atoms belonging to the two decompositions. Rather, we aim to demonstrate that a specific algebraic relationship, similar to the ones discussed above, exists between the two atoms present in the FIT definition. To do this, we generalize the homogeneous linear system in eq. S19 to the four-variable case $(S, X_{past}, Y_{past}, Y_{pres})$. In this scenario, the two decompositions that have $S$ and $Y_{pres}$ as their respective targets (represented by the two lattices in Fig. S1B) are constrained by the following four Shannon information quantities:

$$
\begin{aligned}
&I(S; Y_{pres}) \\
&I(S; Y_{pres}|X_{past}) \\
&I(S; Y_{pres}|Y_{past}) \\
&I(S; Y_{pres}|Y_{past}, X_{past})
\end{aligned}
\tag{S21}
$$

Similarly to eq. S17, we can use eq. S7 to express the 4 quantities in eq. S21 as sums of atoms either belonging to the decomposition with $S$ as target, or the one with $Y_{pres}$ as target. As an example, in Fig. S1B, we demonstrate that $I(S; Y_{pres})$ is the sum of atoms belonging to both decompositions, which together consist of 36 atoms (18 per decomposition). In general, each quantity in eq. S21 imposes constraints between two sets of many atoms from the two decompositions.

To numerically study the solutions of S19 for these 36 atoms, we wrote a MATLAB script named *FIT_nullB.m*. This script computes the four quantities in eq. S21 as the sum of atoms from either the decomposition with $S$ or $Y_{pres}$ as target. It then constructs the $4 \times 36$ matrix $\underline{\underline{B}}$ (Fig. S2) in a similar way to how we derived the $2 \times 8$ matrix in eq. S20 from the eqs. in S17. From Fig. S2, it is clear that in this four-variable case, some atoms, such as $I_{\partial}(Y_{pres}; \{X_{past}\}\{Y_{past}\})$, are not constrained by $\underline{\underline{B}}$ and can vary independently (analogously to how for $N = 2$ the two terms $I_{\partial}(S; \{X\})$ and $I_{\partial}(Y; \{X\})$) were unconstrained). However, there are also pairs of atoms that are not independent when considered individually, but that are specifically related by eq. S19 (i.e. the equality between the two atoms in the pair is a solution of eq. S19). A notable example of these pairwise solutions is made by the pair of atoms appearing in the FIT definition: $I_{\partial}(S; \{X_{past}\}\{Y_{pres}\}) = I_{\partial}(Y_{pres}; \{X_{past}\}\{S\})$. This relationship can be easily verified from Fig. S2, where the first and the second atom are highlighted in red and white, respectively. Indeed, drawing from the intuition developed in the $N = 2$ source variables case, these two atoms belong to columns of $\underline{\underline{B}}$ with element-wise opposite coefficients. This solution (Fig. S2) reveals a specific pairwise relationship between the two atoms appearing in the FIT definition and supports the existence of a finer component of information shared by these two atoms. It is actually apparent from the plot in Fig. S2 that this is the only pairwise relationship involving any of the two atoms. We quantify this finer component of information by taking the minimum between the two related atoms.

**SM1.3.4    Proofs and summary of the main mathematical properties of FIT**

Here we prove that FIT defined as in eq. S11 satisfies the two following properties:

1. FIT is simultaneously upper bounded by $I(S; X_{past})$, $I(S; Y_{pres})$, and $TE(X \to Y)$.

2. FIT depends on $P(S, X_{past}, Y_{pres})$ through all the pairwise marginal distributions $P(S, X_{past})$, $P(S, Y_{pres})$, and $P(X_{past}, Y_{pres})$. Thus, FIT can rule out confounding scenarios where $X$ and $Y$ independently encode $S$ with a temporal lag in absence of within-trial correlations between $X$ and $Y$ at fixed stimulus.

To prove that FIT is simultaneously upper bounded by $I(S; X_{past})$, $I(S; Y_{pres})$, and $TE(X \to Y)$, it is sufficient to note that FIT is simultaneously upper bounded by all quantities that set an upper bound to the two atoms appearing in its definition. This can be seen from eqs. S12, S13, and S15, which show:

$$
\begin{aligned}
FIT &\leq I_{\partial}(S; \{X_{past}\}\{Y_{pres}\}) \leq I(S; X_{past}) \\
FIT &\leq I_{\partial}(S; \{X_{past}\}\{Y_{pres}\}) \leq I(S; Y_{pres}) \\
FIT &\leq I_{\partial}(Y_{pres}; \{X_{past}\}\{S\}) \leq TE(X \to Y)
\end{aligned}
\tag{S22}
$$

A particularly important consequence of the upper bound set by $TE(X \to Y)$ is that if $X$ and $Y$ are independent, then $FIT = 0$. Indeed, if $X$ is independent of $Y$, then $X_{past}$ is independent of

**Matrix of constraints between the two decompositions having S and Yt as targets**

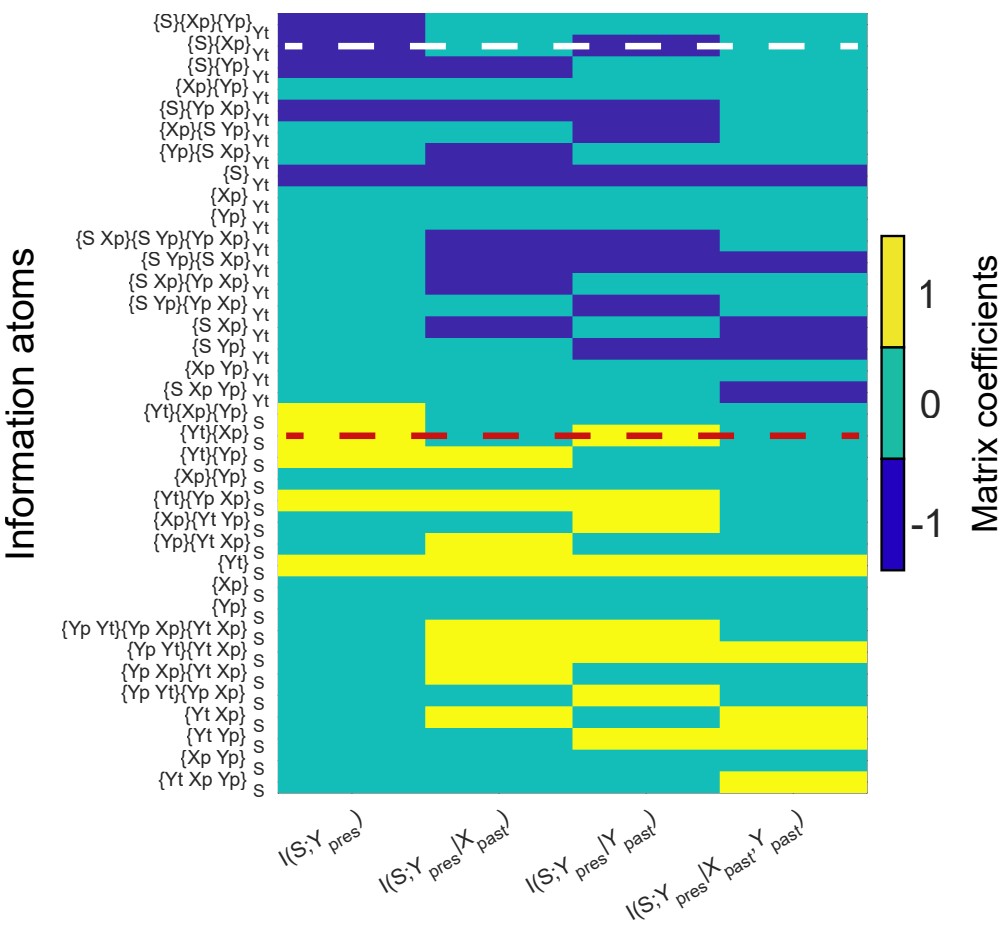

Figure S2: Matrix of constraints imposed by Shannon information-theoretic quantities relating the PID having $(Y_{pres}, X_{past}, Y_{past})$ as source variables and $S$ as target variable to the PID having $(S, X_{past}, Y_{past})$ as source variables and $Y_{pres}$ as target variable. For brevity, we used the notation $Y_t = Y_{pres}, X_p = X_{past}, Y_p = Y_{past}$ and denoted each atom (y axis) directly with the collection it is computed on, with a subscript indicating the target variable of the decomposition (e.g. $\{X_p\}\{Y_t\}_S = I_\partial(S; \{X_{past}\}\{Y_{pres}\})$). For better visibility, we plotted the transpose of the $4 \times 36$ matrix appearing in eq. S19. The red dashed line highlights the first atom appearing in FIT definition $I_\partial(S; \{X_{past}\}\{Y_{pres}\})$, the white line highlights the second atom in FIT definition $I_\partial(Y_{pres}; \{X_{past}\}\{S\})$. Importantly, only atom highlighted in red has coefficients that are opposite to the ones of the atom highlighted in white.

$(Y_{past}, Y_{pres})$, and therefore $I(X_{past}; Y_{past}, Y_{pres}) = 0$. By applying the information theoretic chain rule [9] to $I(X_{past}; Y_{past}, Y_{pres})$, we obtain:

$$I(X_{past}; Y_{past}, Y_{pres}) = I(X_{past}; Y_{pres}|Y_{past}) + I(X_{past}; Y_{past})$$
$$\geq I(X_{past}; Y_{pres}|Y_{past}) \geq FIT \tag{S23}$$

Proving that, if $X$ and $Y$ are independent, then $FIT = 0$. Another important point is that none of the 36 atoms belonging to either decomposition with $S$ or decomposition with $Y_{pres}$ as target satisfies the first property. Indeed eq. S7 does not establish any relationship between atoms of the decomposition with $S$ as target and Shannon information between the sources of the decomposition, including $TE(X \to Y)$ (see Fig. S1B, left), nor between atoms of the decomposition with $Y_{pres}$ as

target and Shannon information between the sources of the decomposition, including $I(S; X_{past})$ (see Fig. S1B, right). Therefore it is necessary to simultaneously consider atoms belonging to different decompositions to obtain a quantity that satisfies the first property.

To prove that FIT depends on $P(S, X_{past}, Y_{pres})$ through all the pairwise marginal distributions $P(S, X_{past})$, $P(S, Y_{pres})$, and $P(X_{past}, Y_{pres})$ we also leveraged on the simultaneous dependence of FIT on $I_{\partial}(S; \{X_{past}\}\{Y_{pres}\})$ and on $I_{\partial}(Y_{pres}; \{X_{past}\}\{S\})$. $I_{\partial}(S; \{X_{past}\}\{Y_{pres}\})$ and $I_{\partial}(Y_{pres}; \{X_{past}\}\{S\})$ depend of $P(S, X_{pres}, Y_{past})$ through the marginals $P(S, X_{pres})$ and $P(S, Y_{past})$, and $P(Y_{pres}, X_{pres})$ and $P(S, Y_{past})$, respectively (see SM1.3.1 and SM1.3.2). Therefore:

$$FIT = f(P(S, X_{pres}, Y_{past})) = f(P(S, X_{past}), P(S, Y_{pres}), P(X_{past}, Y_{pres})) \tag{S24}$$

This implies that if we partially disrupt the dependency structure of our data and create surrogate data where the individual dependencies of $X$ and of $Y$ on $S$ are preserved (i.e., the pairwise marginals $P(S, X_{past})$ and $P(S, Y_{pres})$ do not change) and the within-trial correlations at a fixed stimulus are disrupted (i.e., the conditional distribution $P(X_{pres}, Y_{past}|S)$ changes), the value of the FIT can differ from its original value. This change occurs because this operation generally disrupts $P(X_{pres}, Y_{past})$. Therefore, FIT can rule out confounding scenarios where $X$ and $Y$ encode $S$ independently with a temporal lag, with no information transfer at fixed stimulus value.

### SM1.4 The conditional feature specific information cFIT

Here we discuss the definition and the properties of the conditioned version of FIT, termed cFIT.

### SM1.4.1 Definition and derivation of cFIT

We defined a conditioned version of FIT, to remove from the feature information transmitted from $X$ to $Y$ (that in this Section we term $FIT_X$) the information potentially routed through the past activity of a third region $Z$ ($Z$ can, in principle, also be the multivariate activity of a set of regions). To do so, we identified subcomponents of the two atoms in the FIT definition that quantified pieces of information that were also shared with the past of $Z$, and removed them from FIT.

In this subsection, we will be working with atoms computed on collections belonging to decompositions with $N = 3$ and $N = 4$ source variables. To avoid any confusion, we will explicitly denote the number of source variables of each collection in the following discussion. For example, we will use the notation $\{X_{past}\}\{Y_{pres}\}_{(3)}$ to indicate the collection on which the first atom in the FIT definition is computed. This collection refers to the PID with $N = 3$ source variables $(X_{past}, Y_{pres}, Y_{past})$ and target variable $S$.

Previous studies showed that, using eq. S7 , atoms on the PID with target $T$ and $N$ source variables $\underline{X}_N = (X_1, \dots, X_N)$ can be written as the sum of finer atoms belonging to the PID with same target and an additional source variable $\underline{X}_{N+1} = (X_1, \dots, X_N, X_{N+1})$[7]. Importantly, collections $\alpha$ present in the PID with $N$ source variables, denoted as $\alpha_{(N)}$, also exist in the PID with $N + 1$ source variables, denoted as $\alpha_{(N+1)}$, since $\underline{X}_N \subset \underline{X}_{N+1}$. However, the opposite does not necessarily hold.

The atom $I_{\partial}(T; \alpha_{(N)})$ is the sum of atoms $I_{\partial}(T; \beta_{(N+1)})$, where $\beta_{(N+1)}$ simultaneously precedes $\alpha_{(N+1)}$ in the PID with $N + 1$ source variables (as per the ordering relationship of eq. S4, in which precedence includes equality), but does not precede any collections $\gamma_{(N+1)}$ such that $\gamma_{(N)}$ precedes $\alpha_{(N)}$ [7]. For example, the collection $\{X_{past}\}\{Y_{pres}\}_{(3)}$, present in the PID with $N = 3$ source variables $(X_{past}, Y_{past}, Y_{pres})$ and target $S$, is preceded only by the collection $\{X_{past}\}\{Y_{pres}\}\{Y_{past}\}_{(3)}$ (i.e., the information that all source variables $(X_{past}, Y_{past}, Y_{pres})$ share about $S$) and by itself (see Fig.S1B, left). When adding $Z_{past}$ to the set of source variables, the collection $\{X_{past}\}\{Y_{pres}\}_{(4)}$ is preceded by four collections (additionally to itself): $\{X_{past}\}\{Y_{pres}\}\{Y_{past}\}_{(4)}$, $\{X_{past}\}\{Y_{pres}\}\{Y_{past}\}\{Z_{past}\}_{(4)}$, $\{X_{past}\}\{Y_{pres}\}\{Z_{past}\}_{(4)}$, and $\{X_{past}\}\{Y_{pres}\}\{Z_{past}Y_{past}\}_{(4)}$ (see Fig.S7A, right). Collection $\{X_{past}\}\{Y_{pres}\}\{Y_{past}\}_{(4)}$), which was not preceded by any collection apart from itself in the PID with $N = 3$ variables, is preceded also by $\{X_{past}\}\{Y_{pres}\}\{Y_{past}\}\{Z_{past}\}_{(4)}$). Therefore:

$$I_{\partial}(S; \{X_{past}\}\{Y_{pres}\}\{Y_{past}\}_{(3)})$$
$$= I_{\partial}(S; \{X_{past}\}\{Y_{pres}\}\{Y_{past}\}_{(4)}) + I_{\partial}(S; \{X_{past}\}\{Y_{pres}\}\{Y_{past}\}\{Z_{past}\}_{(4)}) \tag{S25}$$

which intuitively means that, when considering also the past of a third region $Z$, the information that $(X_{past}, Y_{past}, Y_{pres})$ share about $S$ breaks down into a component that is also shared with $Z_{past}$ and a component that is unique with respect to $Z_{past}$.

The other two collections $\{X_{past}\}\{Y_{pres}\}\{Z_{past}\}_{(4)})$ and $\{X_{past}\}\{Y_{pres}\}\{Z_{past}Y_{past}\}_{(4)})$ precede $\{X_{past}\}\{Y_{pres}\}_{(4)})$ but do not precede $\{X_{past}\}\{Y_{pres}\}\{Y_{past}\}\}_{(4)})$ (Fig.S7A). Therefore:

$$I_\partial(S; \{X_{past}\}\{Y_{pres}\}_{(3)}) = I_\partial(S; \{X_{past}\}\{Y_{pres}\}_{(4)})$$
$$+I_\partial(S; \{X_{past}\}\{Y_{pres}\}\{Y_{past}Z_{past}\}_{(4)}) + I_\partial(S; \{X_{past}\}\{Y_{pres}\}\{Z_{past}\}_{(4)}) \tag{S26}$$

which shows how the first atom in FIT definition (eq. S11) breaks down into three components in the PID with $(X_{past}, Y_{pres}, Y_{past}, Z_{past})$ as source variables and $S$ as target variable (one component that is unique with respect to $Y_{past}$ but shared with $Z_{past}$, one that is unique with respect to both $Y_{past}$ and $Z_{past}$ but shared with $\{Y_{past}Z_{past}\}$, and one that is also unique with respect to $\{Y_{past}Z_{past}\}$). One of these atoms is the information that $X_{past}$, $Y_{pres}$, and $Z_{past}$ share about $S$, i.e. the component of the first FIT atom that is also shared with $Z_{past}$: $I_\partial(S; \{X_{past}\}\{Y_{pres}\}\{Z_{past}\}_{(4)})$.

Similarly, the second atom appearing in the FIT definition, is the sum of finer atoms belonging to the PID with $(X_{past}, S, Y_{past}, Z_{past})$ as source variables and $Y_{pres}$ as target variable:

$$I_\partial(Y_{pres}; \{X_{past}\}\{S\}_{(3)}) = I_\partial(Y_{pres}; \{X_{past}\}\{S\}_{(4)})+$$
$$+I_\partial(Y_{pres}; \{X_{past}\}\{S\}\{Y_{past}Z_{past}\}_{(4)}) + I_\partial(Y_{pres}; \{X_{past}\}\{S\}\{Z_{past}\}_{(4)}) \tag{S27}$$

One of these atoms is the information that $X_{past}$, $S$, and $Z_{past}$ share about $Y_{pres}$, i.e. the component of the second FIT atom that is also shared with $Z_{past}$: $I_\partial(Y_{pres}; \{X_{past}\}\{S\}\{Z_{past}\}_{(4)})$.

To remove from FIT the information that is also shared with $Z_{past}$ we defined the conditional FIT (cFIT) from $X$ to $Y$ conditioned to $Z$ as:

$$cFIT_{X|Z} = min[I_\partial(S; \{X_{past}\}\{Y_{pres}\}_{(3)}), I_\partial(Y_{pres}; \{X_{past}\}\{S\}_{(3)})]+$$
$$-min[I_\partial(S; \{X_{past}\}\{Y_{pres}\}\{Z_{past}\}_{(4)}), I_\partial(Y_{pres}; \{X_{past}\}\{S\}\{Z_{past}\}_{(4)})] \tag{S28}$$

Therefore $cFIT_{X|Z}$ is equal to FIT from $X$ to $Y$ (cf. eq. S11) minus a term that is the minimum between two similar information atoms (both quantifying intuitively the feature information about $S$ that both the past of $X$ and the past of $Z$ share with the present of $Y$, but is unique with respect to the past of $Y$) on the two PID having $S$ and having $Y_{pres}$ as target variables, respectively.

### SM1.4.2  Properties of cFIT

In this Section we prove two properties of cFIT, under the assumption that we compute PID atoms using a redundancy measure (such as $I_{min}$) that is non-negative for each atom. The first property we prove (i) is that $cFIT_{X|Z}$ is upper bounded by $FIT_X$ and is lower bounded by the maximum between 0 and $FIT_X - FIT_Z$ (where we denote as $FIT_X$ the feature information transmitted from $X$ to $Y$ and $FIT_Z$ the one transmitted from $Z$ to $Y$). The second property that we prove (ii) is that if $S \to Z_{past} \to Y_{pres}$ is a Markov chain (i.e. $P(S; Y_{pres}|Z_{past}) = P(S|Z_{past})P(Y_{pres}|Z_{past})$) then $cFIT_{X|Z} = 0$. This second property is important because it means that if the present of $Y$ received all its feature information from the past of a recorded region $Z$, then there is no residual FIT through $X$ once any contribution from $Z$ is eliminated.

We start by proving property (i). From eq. S28, since we subtract from FIT the minimum between two non-negative quantities, it immediately follows that $cFIT_{X|Z} \leq FIT_X$. This proves that $cFIT_{X|Z}$ is upper bounded by $FIT_X$. Then, since from eqs. S26 and S27 we have that

$$I_\partial(S; \{X_{past}\}\{Y_{pres}\}_{(3)}) \geq I_\partial(S; \{X_{past}\}\{Z_{past}\}\{S\}_{(4)})$$
$$I_\partial(Y_{pres}; \{X_{past}\}\{Y_{pres}\}_{(3)}) \geq I_\partial(Y_{pres}; \{X_{past}\}\{Z_{past}\}\{S\}_{(4)}) \tag{S29}$$

from which it follows that:

$$min[I_\partial(S; \{X_{past}\}\{Z_{past}\}\{Y_{pres}\}_{(4)}), I_\partial(Y_{pres}; \{X_{past}\}\{Z_{past}\}\{S\}_{(4)})]$$
$$\leq min[I_\partial(S; \{X_{past}\}\{Y_{pres}\}_{(3)}), I_\partial(Y_{pres}; \{X_{past}\}\{S\}_{(3)})] \tag{S30}$$

Eq. S30 shows that the term that we subtract from the right-hand side in eq. S28 is lower or equal to the first one, proving that $cFIT_{X|Z} \geq 0$.

Finally, we prove that $cFIT_{X|Z} \geq FIT_X - FIT_Z$. We do so by proving that the term we subtract from $FIT_X$ in the definition of $cFIT_{X|Z}$ (eq. S28) is smaller than $FIT_Z$. $FIT_Z$ is defined on the two decompositions having $(Y_{pres}, Y_{past}, Z_{past})$ as sources and $S$ as target variable, and the one having $(S, Y_{past}, Z_{past})$ as sources and $Y_{pres}$ as target variable:

$$FIT_Z = min[I_\partial(S; \{Z_{past}\}\{Y_{pres}\}_{(3)}), I_\partial(Y_{pres}; \{Z_{past}\}\{S\}_{(3)})] \tag{S31}$$

Similarly to eqs. S26 and S27, the two atoms in S31 break down into the sum of finer information atoms - when adding variable $X_{past}$ to the respective sets of source variables (in Fig.S7A we show a graphical depiction of the decomposition of the first atom in $FIT_Z$ definition, depicted in light blue):

$$I_\partial(S; \{Z_{past}\}\{Y_{pres}\}_{(3)}) = I_\partial(S; \{Z_{past}\}\{Y_{pres}\}_{(4)})$$
$$+I_\partial(S; \{Z_{past}\}\{Y_{pres}\}\{Y_{past}X_{past}\}_{(4)}) + I_\partial(S; \{Z_{past}\}\{Y_{pres}\}\{X_{past}\}_{(4)}) \tag{S32}$$

$$I_\partial(Y_{pres}; \{X_{past}\}\{S\}_{(3)}) = I_\partial(Y_{pres}; \{X_{past}\}\{S\}_{(4)})$$
$$+I_\partial(Y_{pres}; \{X_{past}\}\{S\}\{Y_{past}Z_{past}\}_{(4)}) + I_\partial(Y_{pres}; \{X_{past}\}\{S\}\{Z_{past}\}_{(4)}) \tag{S33}$$

From eqs. S32 and S33 it follows that $I_\partial(S; \{Z_{past}\}\{Y_{pres}\}_{(3)}) \geq I_\partial(S; \{Z_{past}\}\{X_{past}\}\{Y_{pres}\}_{(4)})$ (i.e. the information the past of $Z$ and the present of $Y$ share about $S$ is larger than the information that they both share also with the past of $X$ about $S$) and $I_\partial(Y_{pres}; \{Z_{past}\}\{S\}\}_{(3)}) \geq I_\partial(Y_{pres}; \{Z_{past}\}\{X_{past}\}\{S\}\}_{(4)})$. Thus:

$$FIT_Z \geq min[I_\partial(S; \{Z_{past}\}\{Y_{pres}\}\{X_{past}\}_{(4)}), I_\partial(Y_{pres}; \{X_{past}\}\{S\}\{Z_{past}\}_{(4)})] \tag{S34}$$

The above proves that $cFIT_{X|Z} \geq FIT_X - FIT_Z$. This is important because it assures that the component that we subtract from $FIT_X$ when removing from it any contribution potentially due to $Z_{past}$ cannot exceed the feature information transmitted from $Z$ to $Y$ (if we only remove the $FIT_Z$ that is shared with $FIT_X$). To summarize, we proved that $cFIT_{X|Z} \leq FIT_X$, that $cFIT_{X|Z} \geq 0$ and that $cFIT_{X|Z} \geq FIT_X - FIT_Z$, meaning that $cFIT_{X|Z}$ is upper bounded by $FIT_X$ and is lower bounded by $max[0, FIT_X - FIT_Z]$.

We now prove property (ii): if $S \to Z_{past} \to Y_{pres}$ is a Markov chain (i.e. $I(S; Y_{pres}|Z_{past}) = 0$) then $cFIT_{X|Z} = 0$). If $S \to Z_{past} \to Y_{pres}$ is a Markov chain, that is $P(S; Y_{pres}|Z_{past}) = P(S|Z_{past})P(Y_{pres}|Z_{past})$, then $I(S; Y_{pres}|Z_{past}) = 0$ [9]. Using the information-theoretic chain rule [9] we can write:

$$I(S; Y_{pres}|Z_{past}) = I(S; Y_{pres}, Z_{past}) - I(S; Z_{past}) =$$
$$= I(Y_{pres}; S, Z_{past}) - I(Y_{pres}; Z_{past}) \tag{S35}$$

Therefore $I(S; Y_{pres}|Z_{past}) = 0$ implies $I(S; Y_{pres}, Z_{past}) = I(S; Z_{past})$ (meaning that all PID atoms that are a subpart of $I(S; Y_{pres}, Z_{past})$, but not of $I(S; Z_{past})$, are zero) and also $I(Y_{pres}; S, Z_{past}) = I(Y_{pres}; Z_{past})$ (meaning that all PID atoms that are a subpart of $I(Y_{pres}; S, Z_{past})$, but not of $I(Y_{pres}; Z_{past})$, are zero). In particular, in eqs. S26 all atoms are computed on collections preceding (according to eq. S4) collection $\{Y_{pres}Z_{past}\}$, meaning that, due to eq. S7, they are all a subcomponent of $I(S; Y_{pres}, Z_{past})$. However, among these atoms, only the collection in $I_\partial(S; \{X_{past}\}\{Y_{pres}\}\{Z_{past}\}_{(4)})$ precedes collection $\{Z_{past}\}$ on this decomposition and, therefore, is a subcomponent of $I(S; Z_{past})$. Since in our case $I(S; Y_{pres}, Z_{past}) = I(S; Z_{past})$, the other two atoms on the right-hand side of S26 are zero. Thus, if $S \to Z_{past} \to Y_{pres}$ is a Markov chain, the following identity holds for the first atom in FIT definition $I_\partial(S; \{X_{past}\}\{Y_{pres}\}_{(3)}) = I_\partial(S; \{X_{past}\}\{Y_{pres}\}\{Z_{past}\}_{(4)})$. Similarly, in eqs. S27 all atoms are computed on collections preceding collection $\{SZ_{past}\}$ (meaning that, they are a subcomponent of $I(Y_{pres}; S, Z_{past})$). However, among these atoms, only $I_\partial(Y_{pres}; \{X_{past}\}\{S\}\{Z_{past}\}_{(4)})$ precedes collection $\{Z_{past}\}$ on this decomposition and, therefore, is a subcomponent of $I(Y_{pres}; Z_{past})$. Since in our case $I(Y_{pres}; S, Z_{past}) = I(Y_{pres}; Z_{past})$, the other two atoms on the r.h.s. of S26 are zero. Thus, if $S \to Z_{past} \to Y_{pres}$ is a Markov chain, the following identity holds for the second atom in FIT definition $I_\partial(Y_{pres}; \{X_{past}\}\{S\}_{(3)}) = I_\partial(Y_{pres}; \{X_{past}\}\{S\}\{Z_{past}\}_{(4)})$. Altogether, we found that, if $S \to Z_{past} \to Y_{pres}$ is a Markov chain, the two atoms appearing in FIT definition (eq. S11) are exactly equal to the two atoms between which we minimize to remove the effect of $Z$ from $FIT_X$ in eq. S28, proving that in this scenario $cFIT_{X|Z} = 0$.

**SM1.5   PID decomposition of DFI**

We next use PID to examine a previously introduced measure of the information about a specific stimulus feature $S$ flowing from $X$ to $Y$, called Directed Feature Information (DFI) [17].

This measure was defined by reasoning to first consider TE between $X$ and $Y$ as a measure of the overall information transmitted from $X$ to $Y$ and then to subtract out from it the information that is not due to changes in the value of the stimulus feature. The latter was estimated as $TE(X \rightarrow Y|S)$, the value of TE conditioned on the stimulus feature, that is the expected value of the TE when it is conditioned on the value of a particular stimulus feature. The reasoning of [17] is that the conditioning removes information not related to variations of the stimulus feature, and that thus $TE(X \rightarrow Y|S)$ quantifies the amount of information transferred from $X$ to $Y$ that is not related to the variations in the stimulus feature. With this reasoning, the authors of [17] defined the DFI to measure stimulus-feature specific information transfer by subtracting out from the total information their estimate of the one that is not related to variations in stimulus features [17]:

$$DFI(X \rightarrow Y) = TE(X \rightarrow Y) - TE(X \rightarrow Y|S) \tag{S36}$$

The authors of Ref [17] showed that DFI is equivalent to the difference between the sum of the information about $S$ that each of $X_{past}$ and $Y_{pres}$ individually carry, minus the information about $S$ jointly carried by $X_{past}$ and $Y_{pres}$, with all the information quantities conditioned on $Y_{past}$:

$$DFI(X \rightarrow Y) = I(S; X_{past}|Y_{past}) + I(S; Y_{pres}|Y_{past}) - I(S; X_{past}, Y_{pres}|Y_{past}) \tag{S37}$$

The difference between information individually carried and information jointly carried is often referred to as co-information [38; 3]. This measure of co-information has been used in the literature as a measure of the net effect of redundancy and synergy and it indicates prevalent redundancy when positive and prevalent synergy when negative [33; 30]. In this rewriting, DFI has some similarities with FIT, in that it uses a measure of redundancy (although conflating synergy and redundancy) between stimulus information in the past of $X$ and in the present of $Y$, as well as a discounting, by conditioning, of the past activity of $Y$.

Previous work on PID has shown that co-information can be expressed as the difference between two non-negative pieces of information which properly quantify synergy and redundancy [38; 3]. Therefore a simple difference between DFI and FIT is that DFI possibly also includes terms of synergy between $X_{past}$ and $Y_{pres}$ than should not be included in a definition of transmission of feature information from $X$ to $Y$. Moreover, given that DFI conditions on the past activity of $Y$ rather requiring uniqueness with respect to the past feature information of $Y$ (as in FIT), it does not isolate information in the present activity of $Y$ that has not been present before in $Y$. To understand better the consequences of these facts in terms of the difference between DFI and FIT, we reformulated DFI as a sum of the partial information terms from the PID, as follows:

$$\begin{aligned}
DFI_{X \rightarrow Y} &= I(S; X_{past}|Y_{past}) + I(S; Y_{pres}|Y_{past}) \\
&\quad - I(S; Y_{pres}, Y_{past}, X_{past}) + I(S; Y_{past}) \\
&= I_{\partial}(S; \{X_{past}, Y_{past}\}\{Y_{pres}, Y_{past}\}) \\
&\quad + I_{\partial}(S; \{X_{past}, Y_{past}\}\{Y_{pres}, Y_{past}\}\{X_{past}, Y_{pres}\}) + \\
&\quad + I_{\partial}(S; \{X_{past}\}\{Y_{pres}, Y_{past}\}) + I_{\partial}(S; \{Y_{pres}\}\{X_{past}, Y_{past}\}) \\
&\quad + I_{\partial}(S; \{X_{past}\}\{Y_{pres}\}) - I_{\partial}(S; \{X_{past}, Y_{pres}\}) - I_{\partial}(S; \{X_{past}, Y_{past}, Y_{pres}\})
\end{aligned} \tag{S38}$$

Note that in the above expression all terms involving pieces of redundant information are positive and those only involving synergistic information are negative. Thus this decomposition of DFI demonstrates that it is the linear combination of (mostly) redundant information terms appearing with a positive sign and synergistic information terms appearing with a negative sign. This explains why, as a result of not separating redundancy from synergy, DFI can be negative and difficult to interpret as information about a stimulus feature transmitted from $X$ to $Y$.

The fact that DFI can become negative also shows that using $TE(X \rightarrow Y|S)$ to remove from the total transmitted information the one not about the stimulus feature $S$ (as done in DFI, see [17]) is incorrect. This is because $TE(X \rightarrow Y|S)$ does not really quantify the information from $X$ to $Y$ which is not about $S$, as conceptualized in Ref [17]. It actually quantifies the information transmitted on average within each feature condition. This can overestimate the information from $X$

to $Y$ which is not about $S$. In simple terms, when using data from the same feature conditions, some information sent from $X$ to $Y$ in this subset of data could be about the specific value of the feature in the considered set of trials. When the strength of communication about $S$ between $X$ and $Y$ varies from one feature value to another, this overestimation may become even more severe, because in this case additional information about $S$ is encoded synergistically within the network of $X$ ad $Y$ by their feature-dependent relationship [30].

## SM1.6 Numerical computation of FIT and other information quantities

FIT and all other information theoretic quantities were computed from both simulated and real data by plugging into the corresponding equations the numerical evaluation of the response probabilities from the data. We computed the response probabilities by discretizing neural activity into a number $R$ of equipopulated bins [22] and then computing empirically the frequency of occurrence of each binned response across all available trials.

In Table S1 we summarize the number of bins we used to discretize neural activity for each figure in the paper. In Section SM2.5 we study the accuracy of the FIT and TE estimates with the number of available trials and we show that the estimates of FIT and TE are accurate and unbiased for the number of bins and number of trials used for all analyses. However, in the code we provide to compute FIT and TE, we also implemented limited-sampling bias correction routines that can be used to obtain more accurate estimates when data are more scarce (see Section SM2.5).

| Number of bins | 2 | 3 | 4 |
|---|---|---|---|
| **Figures** | Fig.3 Fig.S8 Fig.S9 Fig.S11 Fig.S12B Fig.S13B Fig.S14B Fig.S15D-E | Fig.2A,B,E Fig.S3 Fig.S4 Fig.S5 Fig.S7 Fig.S13A Fig.S14A Fig.S15A-B | Fig.2C Fig.4 Fig.S10 Fig.S12A Fig.S15C |

Table S1: Number of bins used to discretize neural activity for information-theoretic analyses of simulated and real data, for each main text and SM figure

## SM1.7 Permutation-based non-parametric null hypotheses for FIT and TE

To test for significance of the information theoretic quantities, we used non-parametric permutation tests, described below.

To test for the significance of mutual information values about the feature of interest, we used established non-paramametric procedures [8; 12; 19]. We constructed surrogate datasets in which we destroyed any feature information by randomly permuting across trials the values of $S$, and then we recomputed information on the surrogate data to obtain a null-hypothesis distribution of null information values.

To test for the significance of FIT, we developed a permutation test in which we created surrogate data in which we preserve the feature information in the past of $X$ and the present of $Y$ while destroying the communication of this information between $X$ and $Y$. We shuffled $X$ within trials with the same value of the feature $S$, destroying any within-trial statistical relationship between the activity of $X$ and the activity of $Y$ at fixed values of $S$, and recomputed FIT on the surrogate data. This data shuffling preserves the marginal distributions between the feature and the past activity of $X$ and between the stimulus and the present activity of $Y$, thereby preserving the information about the stimulus that each carries. However, it destroys the within-trial statistical relationship at fixed stimulus between $X$ and $Y$ that would be present if $X$ sends stimulus information to $Y$. Because FIT depends on $P(X_{past}, Y_{pres})$ (see mathematical proof in Section SM1.3.4), the values of FIT on the permuted data will be smaller than the ones on the original data whenever there is direct within-trial communication of stimulus information between $X$ and $Y$, but will be similar to the value of the original data when

there is no such direct within-trial communication. As shown by numerical simulations (see Section SM2.6) the so generated null hypothesis distribution of FIT values when $X$ and $Y$ encode but do not communicate stimulus feature information is more conservative (see Fig. S7C) than the simpler one that would be obtained by a permutation test destroying all information about $S$ in $X$ and $Y$ by randomly permuting $S$ across all trials, as for the mutual information quantities above. (This permutation test would implement the idea that if no stimulus-feature information is present, it cannot be transmitted). However, in limiting cases in which the stimulus information in the neural data is absent, we found it numerically better to perform this second random permutation of the label of $S$ across trials (because more possible independent data permutation are available in this second permutation, which therefore may have some advantages in the case of zero or negligible stimulus information, see Fig. S7E). To reduce the probability of false positives in such cases of no information present in the network, we computed and then intersected the two above describe possible permuted distributions by taking the element-wise maximum between the two distributions, and obtained a null distribution for FIT. (In practice, in real data and simulations with stimulus information present, the maximum of the two permuted values coincided in all simulations with the maximum of the first permutation. This is exemplified in Fig. S7C, in which for higher value of the parameter $W_{ZY}$ some information about the stimulus is created in both $X$ and $Y$ in absence of communication between $X$ and $Y$, the null hypothesis for FIT taking the maximum between the first and second permutation has values not only larger than the FIT value measures in the simulation, but also much larger values then the ones based on only shuffling $S$. In simulations with null information, the maximum value of the permuted data in each simulation could instead belong to either permutation.)

An identical procedure was applied to test for the significance of DFI. For TE, since by design the measure captures the total amount of information flowing from $X$ to $Y$, we permuted the neural activity of the sender $X$ across all trials.

Since in all simulations and real data analyses we wanted to test for the significance of the information values averaged either across simulations or participants, we computed the average over simulations and participants of the information values in each realization of the random permutation and we used this distribution of null hypothesis of averaged values for testing the significance of the averaged information value. (To compute the null-hypothesis distribution, we generated 500 different realizations of the permuted average information for each test we conducted.)

In some analyses (e.g. Fig. 3C and 4B,C) we had to identify the cluster of post-stimulus times and transmission delays for which FIT or TE were significantly different from zero (shown, e.g., in Figs. S9G, S10A,B). We individuated these clusters of points in the time-delay space using a cluster-based permutation test [24; 8] using as null hypothesis values those obtained from the permutation test described above. We computed the cluster forming threshold as the 99th percentile of information values in the surrogate data. We created information clusters in the original and shuffled datasets by summing together all adjacent information values above the cluster forming threshold. We then determined a null distribution for information clusters using the maximum cluster value from each shuffled dataset. Finally, we assigned significance to clusters in the original dataset if their value was larger than the 99th percentile of the clusters null distribution (p < 0.01).

## SM2 Details of simulations and and additional analyses of simulated data

### SM2.1 Simulations of FIT and TE as a function of signal and noise transmission

This section pertains to the description of Fig. 2A-B of the main text.

The goal of the first simulation (whose results are reported in Fig. 2A) was to evaluate the dependence of FIT and TE on stimulus-feature-related and -unrelated transmission. The goal of the second simulation (whose results are reported Fig. 2B) was to test the ability of FIT and TE to localize in time the stimulus-feature-related information transmission. The setting of both simulations was identical and is described in the following.

We simulated 500ms of activity of activity of $X$ and $Y$, in time steps of 10ms. The sending region $X$ encoded a stimulus $S$ over time and transmitted stimulus-feature-related and -unrelated activity to the receiver $Y$ with a given temporal lag $\delta$. The stimulus feature $S$ being encoded by $X$ and transmitted to $Y$ was an integer (between 1 and 4) drawn independently and uniformly in each trial (500 trials per stimulus for each of the 50 simulations). The activity of the sender was a two-dimensional

variable with one feature-informative $X_{stim}$ and one feature-uninformative component $X_{noise}$. The stimulus-feature-informative dimension had a temporally-localized feature-dependent bump in the activity (from 200 to 250ms) and multiplicative Gaussian noise

$$X(t)_{stim} = S(t)(1 + \mathcal{N}(0, \sigma_{stim})) \tag{S39}$$

where $S(t)$ was a function equal to the value of the stimulus $s \in [1, 4]$ during the time window $[200, 250]ms$ and was zero outside of this window. The presence of noise in $X(t)_{stim}$ was needed to test for the impact of within-trial encoding of $S$ in $X$ on the within-trial encoding of $S$ in $Y$, at fixed values of $S$ (i.e., when $X$ encodes $S$ incorrectly, also $Y$ encodes $S$ in a similar way). If $X(t)_{stim}$ encoded the stimulus perfectly (no noise in $X(t)_{stim}$, therefore $X(t)_{stim} = S$ for $t \in [200, 250]ms$), it would be impossible to determine whether $Y$ is receiving stimulus information from $X$ or directly from $S$. We choose the noise in the stimulus-feature-informative dimension to be multiplicative because it made it a more challenging scenario for FIT. In fact using multiplicative noise $X$ developed a stimulus-dependent noise in the encoding of $S$. The stimulus-dependent noise in the encoding of $S$ leads to stimulus-dependent within-trial correlations between $X$ and $Y$, which potentially induces synergies in the encoding of $S$ in $X$ and $Y$ [30]. Since FIT computes information transmission by identifying a component of redundant information between the past of $X$ and the present of $Y$, using simulations that have both redundancy between $X$ and $Y$ induced by information transmission and synergy between $X$ and $Y$ induced by stimulus-dependent amount of noise encoded and transmitted (using this kind of multiplicative noise), makes it potentially harder for a measure of feature-specific information transmission to separate out the redundant information that was transmitted. In fact, we will see that measures that do not separate well redundancy and synergy, such as DFI, will suffer under such conditions (leading to negative values of transmitted information (Fig. S15A), whereas FIT seems to work well even under this condition because it uses PID to only include redundant time-lagged information about $S$ in $X$ and $Y$, discarding synergy. (However, we found similar results for TE and FIT by replacing the multiplicative noise with an additive noise in Eq. S39 (Fig. S3)). The stimulus-feature-unrelated component was, at any time point, a zero-mean Gaussian noise $X(t)_{noise} = \mathcal{N}(0, \sigma)$. The activity of the receiver $Y$ was the weighted sum of $X_{stim}$ and $X_{noise}$ with a delay $\delta$, plus a Gaussian noise: $Y(t) = W_{stim}X_{stim}(t-\delta) + W_{noise}X_{noise}(t-\delta) + \mathcal{N}(0, \sigma)$. The delay $\delta$ was chosen randomly in each repetition from a uniform distribution in the range between $40ms$ and $60ms$, in steps of $10ms$. Therefore, across repetitions of the simulation, $Y$ received information from $X$ only in the time window $[240, 310]ms$. In all simulations we set a standard deviation $\sigma = 2$ for the additive Gaussian noise in $X_{noise}$ and $Y$, and a standard deviation $\sigma_{stim} = \sigma/5 = 0.4$ for the multiplicative Gaussian noise in $X_{stim}$.

In the first simulation, we computed FIT and TE at the first time instant in which $Y$ received information from $X$ ($t = 200ms + \delta$), and at the ground truth delay $\delta$, for all combinations of $W_{stim}$ and $W_{noise}$ in the range between 0 and 1, in steps of 0.1. In the second simulation we set $W_{stim} = 0.5$ and $W_{noise} = 1$ and computed FIT and TE at all time points, in a rage of communication delays between 0 and 100ms, and averaged their values over delays to obtain temporal profiles of transmitted information.

### SM2.2  FIT can detect feature specific information flow even with overlapping time courses of stimulus information

This section pertains to the description of Fig. 2C-E of the main text.

One often used method to infer hierarchical flow of information across ares is to consider the timing of neural activation or of stimulus selectivity of activity across brain regions [32]. However, this method is neither necessary nor sufficient to determine real communication. On the one hand time lagged information selectivity between two regions may arise in absence of communication for example if the two regions received a partly shared input signal with a different delay (see also Section SM4.3). On the other hand, as we will exemplify in this section, real features-specific communication between two brain regions could take place even without detectable differences in timing of information across the considered regions. The purpose of this subsection is to illustrate that this can happen and also to show that in such case FIT has power to discriminate between cases in which feature-specific information flow does or does not take place. We will show that this is because FIT can assess that in cases of real communication the format of information encoding is the same in the past activity of the sender and in the present activity of the receiver.

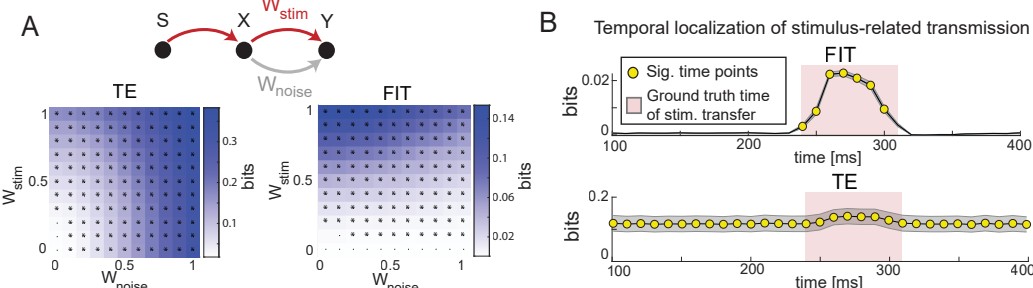

Figure S3: Further tests of FIT on simulated data with additive noise in $X$. This figure is similar to Fig. 2 of the main text except that the results are now obtained with additive (rather than multiplicative) noise in $X$. (A) FIT and TE as function of stimulus-feature-related ($W_{stim}$) and -unrelated ($W_{noise}$) transmission strength. * indicate significant values (p < 0.01, permutation test) for the considered parameter set. (B) Dynamics of FIT and TE in a simulation with time-localized stimulus-feature-information transmission. The red area shows the window of stimulus-feature-related information transfer. Yellow dots show time points with significant information (p < 0.01, permutation test). Results plot mean (lines) and SEM (shaded area) across 50 simulations (2000 trials each).

We simulated a scenario where a sending regions $X$ encodes and transmits to a receiving region $Y$ information about an integer stimulus-feature $S$ (ranging between 1 and 4). Importantly, the past of $Y$ and the present of $X$ carry the same amount of feature information of the past of $X$ and the present of $Y$, but encode the information with different formats.

The *feature encoding format* of each region is determined by the encoding function $f(S)$ controlling the average response of each region to individual stimulus values $s = [1, 2, 3, 4]$. We simulated responses with three different encoding functions:

$$
\begin{aligned}
f_1(S) &= 1 + \delta[0, 1, 2, 3] \\
f_2(S) &= 1 + \delta[1, 0, 2, 3] \\
f_3(S) &= 1 + \delta[0, 1, 3, 2]
\end{aligned}
\tag{S40}
$$

where $\delta$ is a parameter controlling the separation of the average responses to different stimuli, and therefore the amount of feature information carried by each region at a specific time point. We set $\delta = 1$ in all simulations. The three encoding functions $f_1(S)$, $f_2(S)$, $f_3(S)$ are depicted in Fig.2C in blue, green and red, respectively. The encoding function determined the feature values that each region preferentially encodes at each specific time point. Specifically, due to the presence of additive Gaussian noise, regions were most informative (according to Eq. S10) about stimulus values for which the response was either minimum (i.e. equal to 1 in eq. S40) or maximum (i.e. equal to $1 + 3\delta$ in eq. S40). Indeed, activity distributions in response to these stimulus values were less overlapped with activity in response to other stimuli. For example, regions encoding the stimulus as $f_1(S)$ would carry high specific information about stimulus values 1 and 4, and low specific information about stimulus values 2 and 3. On the other hand, regions encoding the stimulus as $f_2(S)$ would carry high specific information about stimulus values 2 and 4, and low specific information about stimulus values 1 and 3. Therefore, since the $I_{min}$ measure quantifies redundancy as the overlap in the distributions of specific information across individual values of target variable, the responses of two regions $X$ and $Y$ would be maximally redundant if they encoded the feature with the same encoding format (e.g., $f_X(S) = f_Y(S) == f_1(S)$), partially redundant if they both carried high specific information about one stimulus value (e.g., $f_X(S) == f_1(S)$ and $f_Y(S) == f_2(S)$) or minimally redundant if they carried high specific information about different pairs of stimulus values (e.g., $f_X(S) == f_2(S)$ and $f_Y(S) == f_3(S)$).

In our simulation, $X$ and $Y$ activity at different time points was described by the following set of equations:

$$
\begin{aligned}
X_{past} &= f_1(S) + E_1 \\
Y_{pres} &= X_{past} \\
Y_{past} &= f_2(S) + E_2 \\
X_{pres} &= f_3(S) + E_3
\end{aligned}
\tag{S41}
$$

where $E_1$, $E_2$, and $E_3$ are additive Gaussian noise with standard deviation equal to $\sigma$. Importantly, real feature transfer only occurs in the $X \to Y$ direction, causing the past of the sender and the present of the receiver to encode the feature with the same format. Since $\sigma$ was equal for all noise terms in eq. S41, both $X$ and $Y$ carried the same amount of stimulus-feature information in the past and in the present, removing any contribution to FIT due to time-lagged information levels in the sender and the receiver region. We measured FIT in the two directions ($X \to Y$ and $Y \to X$) for different levels of noise $\sigma$. By changing $\sigma$, we controlled the $SNR = \delta/\sigma$ in both past and present activity of $X$ and $Y$. We repeated the simulation 100 times for each $SNR$ value ranging between 0.05 and 1 with a precision of 0.05. We measured the FIT significance in the two directions using the permutation test described in section SM1.7.

We found that, because FIT could correctly detect that the format of information representation of $S$ in the present of $Y$ was equal to that of the past of $X$ but different to that of the past of $Y$ (Fig.2D), and that feature information flowed from $X$ to $Y$ (Fig.2E).

### SM2.3   Simultaneous transfer of information about more than one feature

We performed a simulated study of how FIT performs when studying neural system that encode and transmit more than one feature (Fig. S4).

We simulated two independent features (e.g. of a sensory stimulus) S1,S2 simultaneously encoded in a brain region $X$ and transmitted to a brain region $Y$. In the simulation, $S_1$ is more strongly and encoded and transmitted than $S_2$. The equation for simulating the data are as follows:

$$
X = S_1 + DS_2 + E_x
\tag{S42}
$$

where $S_1, S_2$ are independent binary variables (values equal to $\pm 1$), $E_x$ is Gaussian noise with standard deviation equal to 1, and $Y$ equals $X$ with a time lag, plus independent Gaussian noise with standard deviation equal to 1.

We simulated the system with different values of $D$ (the strength of encoding and transmission of $S2$ relative to $S1$). We found (Fig. S4) that FIT identifies correctly that both features are transmitted, and ranks correctly the features about which most information is transmitted. FIT also identified correctly the limiting case ($D = 1$) in which both features are encoded and transmitted with equal strength.

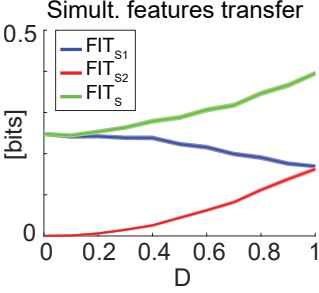

Figure S4: Simulation of a system that encodes independently two features $S1$ and $S2$ in the activity of a brain region $X$ and transmits them to another brain region $Y$. We compute FIT about each of the two stimulus features $S1$, $S2$ or their combination $S = (S_1, S_2)$ as a function of the parameter $D$ describing the strength of encoding and transmission of $S_2$ with respect to $S_1$. We plot mean $\pm$ SEM over 50 simulations

**SM2.4 Simulations of bidirectional transmission between $X$ and $Y$**

Here we describe the simulations whose results are presented in Fig. S5.

To further investigate the ability of FIT to determine the direction of stimulus-feature information flow, we simulated a scenario with bidirectional (back and forth) communication between $X$ and $Y$ with stimulus-feature-related transfer from $X$ and stimulus-feature-unrelated transfer from $Y$ to $X$ (Fig. S5A).

In brief, both $X$ and $Y$ received information directly from a feature-information-sending region $S$. The region $X$ received stimulus information from $S$ early on (between 50 and 90 ms) and $Y$ received stimulus information from $S$ at a later time (between 110 and 150 ms). $X$ sent its entire activity to $Y$ (therefore communicating its stimulus information when it became available). $Y$ instead only sent to $Y$ a part of its activity that did not carry stimulus information. The details of how this was achieved are reported below.

We simulated 180ms of activity of $X$ and $Y$, in steps of 1ms. The stimulus feature $S$ being encoded and transmitted from a stimulus region $S$ to $X$ and $Y$ was an integer (between 0 and 3) drawn independently and uniformly in each trial. The activity of $X$ was one-dimensional. The activity of $Y$ was two-dimensional. Both dimensions of $Y$ ($Y_+$ and $Y_-$) were generated with a Poisson process whose mean was modulated over time by a Gaussian bump (whose amplitude was equal to the stimulus-feature value $S$) in the time window $[110, 150]$ms, plus an additive Gaussian noise and time-lagged readout of $X$ activity (with a $X$ to $Y$ transmission delay $\delta_{xy} = 10$ms). Importantly, $Y_+$ encoded the stimulus as a positive Gaussian bump and $Y_-$ encoded the stimulus as a negative Gaussian bump. Therefore, the entire activity of $Y$, i.e. the sum of the two components, $Y_{noise} = Y_+ + Y_-$ carried no information about the stimulus, and the difference of the two components $Y_{stim} = Y_+ - Y_-$ carried all the stimulus information in $Y$. $X$ was a Poisson process whose mean was positively modulated over time by a Gaussian bump - whose amplitude was modulated by the stimulus - in the time window $[50, 90]$ms, plus an additive Gaussian noise and time-lagged readout of the entire activity of $Y$ $Y_{noise}$ (with a $Y$ to $X$ transmission delay $\delta_{yx} = 15$ms).

We measured FIT at each time step of the simulation over a range of communication delays and averaged the resulting information values over delays to obtain temporal profiles of information transmission. We found that FIT correctly captured the flow of information about $S$ between $X$ and $Y$ that we put by design into these simulated data. FIT revealed that there was a significant stimulus-feature-related information transmission from $X$ to $Y$, that was temporally localized in the actual "ground-truth" $[60, 100]ms$ window in which $Y$ received stimulus information, and no significant stimulus-feature-information transmitted from $Y$ to $X$ (Fig. S5B).

We also used these simulations to test the performance of the Directed Feature Information, DFI [17], using the same analysis pipeline described here for FIT. Results are discussed in Section SM4.2.

**SM2.5 Limited-sampling bias of FIT and TE**

Information-theoretic quantities are know to suffer from a systematic error (called limited sampling bias) when the probabilities used to compute them are estimated from a limited number of experimental trials [26]. While the limited bias of Shannon information quantities such as TE have been studied well and has been shown to be inversely proportional to the number of available trials and directly proportional to the number of bins used to discretize the data [**?** ], those of FIT remain to be investigated.

Therefore here we simulated a simple scenario to study how FIT and TE scale with the number of trials available in the dataset and the number of bins of the discretized activity. In these simulations $X$ encoded a binary feature $S$ with additive Gaussian noise. $Y$ was equal to $X$ with a time lag of 1 plus independent Gaussian noise (standard deviation of noise = 0.5).

We found (Fig. S6) that FIT behaved much better than TE with the data size and number of bins. The correct value of FIT, which can be estimated from large numbers of trials, was achieved already with smaller number of trials than for TE. We found that accurate calculations of FIT are possible with the number of trials available in empirical datasets (Fig. S6); for comparison FIT calculations with real and simulated data in this paper were done with 2-4 discretization bins, see Table S1). Our understanding is that the better scaling and sampling properties of FIT with respect to Shannon Information quantities arise because FIT considers a PID part of the total information which has

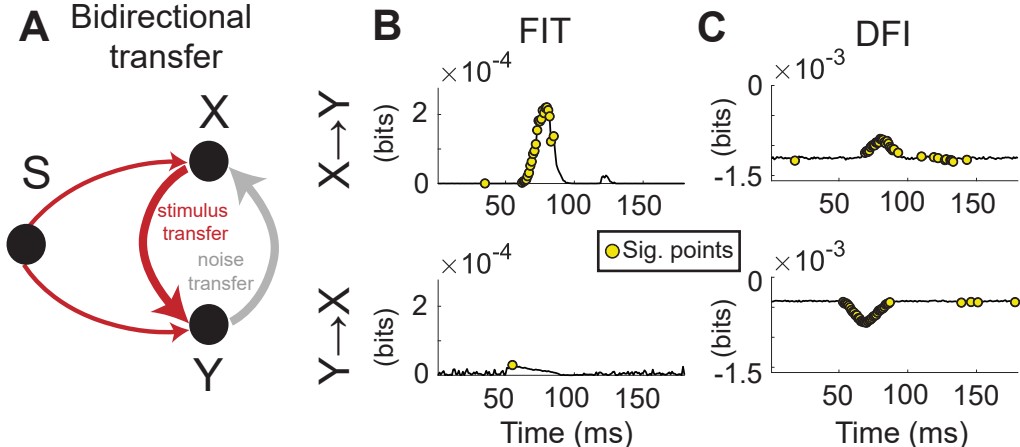

Figure S5: Simulations of performance of FIT and other measures in a case of transfer of stimulus-feature-related and stimulus-unrelated information in different directions. (A) Schematic of the simulation. A stimulus node $S$ provides partly complementary information about a stimulus feature to $X$ (in the 50-90ms interval) and to $Y$ (in the 110-150ms interval). $X$ transmits stimulus information to $Y$. $Y$ instead has different components of activity and it projects to $X$ only the component of its activity that is stimulus-unrelated. In other words, we have stimulus-feature-information transfer from $X$ to $Y$ and noise transfer from $Y$ to $X$). (B) Results of the analyses of this simulated activity using FIT (left panels) and DFI (right panels). Gray lines plot the value of these quantities Yellow dots plot time points in which the measure was significantly different from null (permutation test of Section SM1.7; p<0.01)

.

lesser bias compared to other parts of the total information. Given that the PID atoms of FIT do not contain synergistic terms, this is in line with previous work [25] showing that synergistic components of information have much larger limited sampling bias, and that information quantities that do not include synergistic components have much better sampling properties than full multivariate Shannon information quantities. Thus, FIT can be computed from the datasets in which Shannon information measures typically applied to neural data. We applied a widely used bias correction technique, called the Quadratic Extrapolation [37; 26]. This method is based on subtracting the bias estimated from a second-order polynomial fitting of the dependence of the estimated quantity on sub-samples of the available data. We found (Fig. S6) that this bias subtraction technique was helpful in further improving the estimate of information (reducing the limited sampling bias) in cases of very low numbers of trials available. This bias correction technique is made available in the software we provide for both FIT and TE.

### SM2.6   Simulation tests of the significance of FIT and cFIT

In this Section we describe the simulations and results presented in Fig. S7.

In this set of simulations, we first evaluated the effectiveness of the permutation-based non-parametric tests for FIT in a difficult scenario in which $X$ and $Y$ independently encode stimulus-feature information with a temporal lag, but no actual communication occurs between them. We then evaluated the performance of cFIT in measuring the unique contribution of $X$ in sending feature information to $Y$ in presence of an alternative information route through a third node $Z$ sending information to $Y$.

We addressed both questions using a the following simulation setup. We performed a simulation in which two senders ($X$ and $Z$) both transmitted stimulus-feature information to $Y$ (Fig. S7B). $X$ encoded the stimulus-feature linearly and $Z$ non-linearly so that they carried partially different information about $S$. This is important because a good measure quantifying the unique contribution of $X$ (but not $Z$) in the transmission of information to $Y$ should capture that, even if the total amount of feature information transmitted from $Z$ to $Y$ is stronger than the one transmitted from $X$ to

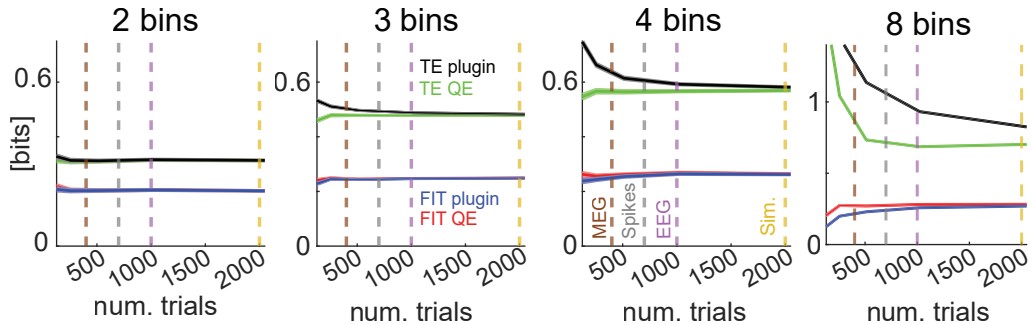

Figure S6: FIT and TE as a function of the number of simulated trials used to compute them. Vertical lines: number of trials used for the analyses of real and simulated brain data in this paper. "Plugin": plugging empirical probability histograms into FIT, TE eqs. "QE": adding Quadratic Extrapolation bias correction. For all analyses in the rest of the paper we used 2-4 discretization bins (see Table S1). Results in each panel are ploted as mean $\pm$ SEM over 100 simulations

.

$Y$, there can still be a different component of information that is uniquely transmitted by $X$ and not by $Z$. We simulated 500ms of activity, in time steps of 10ms. The encoded and transmitted stimulus feature $S$ was a stimulus-intensity integer value (0 to 3) drawn independently and uniformly in each trial (500 trials per stimulus). The activity of $X$ had a temporally-localized square bump between 200 to 250ms whose amplitude depended linearly on $S$, with multiplicative Gaussian noise. Activity of $Z$ had a temporally-localized square bump (from 200 to 250ms) encoded with a different format with respect to $X$, and multiplicative Gaussian noise. Specifically, $X$ encoded the stimulus feature $S = (0, 1, 2, 3)$ with the encoding function while $Z$ encoded $S$ with the encoding function $S = (1, 0, 3, 2)$. In this way, $X$ carries more specific information than $Z$ (see eq S10) about $S = 0, 3$ and $Z$ carries more specific information than $X$ about $S = 1, 2$, therefore both $X$ and $Z$ carry some unique information about $S$. Activity of $Y$ was the weighted sum of $X$ and $Z$ with a temporal lag, plus additive Gaussian noise: $Y(t) = W_{xy}X(t - \delta) + W_{zy}Z(t - \delta) + \mathcal{N}(0, \sigma)$. The delay $\delta$ in the transmission of information from $X$ to $Y$ was chosen in each repetition of the simulation randomly from a uniform distribution in the range between $40ms$ and $60ms$ . We computed FIT and cFIT at the first time instant in which information in $Y$ was received from $X$ and $Z$ ($t = 200ms + \delta$) using to define past activity the ground-truth delay $\delta$ actually used in that simulation. We set a standard deviation $\sigma = 2$ for the additive Gaussian noise in $Y$, and a standard deviation $\sigma_{stim} = \sigma/5 = 0.4$ for the multiplicative Gaussian noise in $X$ and $Z$.

### SM2.6.1 Tests of significance of FIT accounting for the possible existence of encoded feature information in the absence of transfer of it across regions

We first addressed the first question, that is how to deal with confounding scenarios where $X$ and $Y$ independently encode feature information with a temporal lag, but no actual communication occurs between them.

We studied how the FIT from $X$ to $Y$ depended on the strength of feature-related transmission from $Z$ to $Y$ $W_{zy}$ when no stimulus-feature-information was transmitted from $X$ to $Y$ ($W_{xy} = 0$). We found that FIT from $X$ to $Y$ increased with $W_{zy}$, since $X$ and $Y$ carried redundant information about $S$ with a temporal lag. However, FIT was always non-significant (Fig. S7C) using the permutation test described in Section SM1.7, since there were no within-trial correlations between the encoding of $S$ in $X$ and the time-lagged encoding of $S$ in $Y$. This proves that, even if $Z$ was not measured, the permutation test we provided for FIT can correctly rule out confounding scenarios where $X$ and $Y$ encode $S$ with a temporal lag but with no actual communication occurring between $X$ and $Y$ (see Section SM1.7 and SM2.6).

**SM2.6.2  Simulations testing cFIT in the presence of information transfer through an alternative route involving a third region $Z$**

We next addressed the second question, that is how to evaluate the unique contribution of $X$ in sending feature information to $Y$ in presence of an alternative information route through a third node $Z$ sending information to $Y$. We studied how FIT from $X$ to $Y$ and cFIT from $X$ to $Y$ conditioned on the feature information in $Z$ depended on the simultaneous transmission of feature information from $X$ to $Y$ and from $Z$ to $Y$. To do this, we computed FIT and cFIT for all combinations of $W_{xy}$ and $W_{zy}$ in the range between 0 and 1, in steps of 0.1. We found that FIT grew both as a function of $W_{xy}$ and of $W_{zy}$ and was significant as soon as some information was transmitted from $X$ to $Y$ ($W_{xy} > 0$; Fig. S7D, left). On the contrary, cFIT increased only as a function of $W_{xy}$ and decreased with $W_{zy}$, correctly removing from the FIT from $X$ to $Y$ the feature information that was routed to $Y$ through $Z$ (Fig. S7D, right). Crucially, cFIT did not simply subtract from FIT through $X$ the FIT through $Z$, but it only removed the amount of feature information that was redundantly transmitted by $X$ and by $Z$ to $Y$. Indeed, since $X$ and $Z$ transmitted partially different feature information to $Y$, we have that cFIT was still significant for many combinations of parameters where $W_{zy} > W_{xy}$ (Fig. S7D, right) and, therefore, FIT through $X$ was larger than FIT through $Z$ (not shown).

**SM2.7  Simulation studies of how FIT and TE are affected by the mixing of sources**

In real electrophysiological recordings, it is possible that that separation and reconstruction of the underlying neural sources is imperfect, due to issues such as for example field spread or common referencing. As a result, electrophysiological recordings from different brain regions may contain, with different weights, a mixture of sources. It has been proposed that such source mixing may affect measures of communication between brain areas [1]. Here, we examine the effect of this source mixing in FIT and TE measures.

We simulated source mixing in different proportions in the sender $X$ and the receiver $Y$. In our simulations we assumed that (as it is expected to be the case in real brain data) the mixing is instantaneous (i.e. sources are mixed with zero lag) and with a proportion of source sharing in $X,Y$ that is stable across time.

We first simulated a "null model" scenario in which a source $Z$ (informative about a stimulus feature $S$) is shared between $X$ and $Y$ with a different proportion $A$:

$$
\begin{aligned}
X &= Z(s) + E_x \\
Y &= AZ(s) + E_y
\end{aligned}
\tag{S43}
$$

with $E_x$, $E_y$ being independent Gaussian noise. The stimulus feature $S$ was a binary value extracted independently in each trial. The source $Z$ encoded the feature with additive Gaussian noise. The amount of stimulus-feature information encoded in $Z$ increased linearly over time. We controlled the SNR of $X$ and $Y$ by changing A (which sets the relative level of stimulus-feature signal in $X,Y$) and fixing noise standard deviation to 1. On this model FIT and TE had spurious positive values (Fig. S8B). We used the permutation test introduced in Section SM1.7, testing for spurious values induced by $X,Y$ covariations due to feature-signal sharing. We found that this test correctly ruled out as non-significant FIT and TE values generated only by source sharing with no real transmission (Fig. S8B). Importantly, analysis of this model also showed that with instantaneous source mixing (and notably under the assumption that recording noise is constant over the time of the trial) the ratio between stimulus-feature info in $X$ and $Y$ is constant in time (Fig. S8A). This gives a useful heuristic: while the finding that the feature information time courses of two individual regions that overlap in time cannot be used to rule in or out communication of information about the feature between the two areas (see SM2.2 and Fig 2D-E), different timecourses of stimulus-feature info in $X$ vs $Y$ cannot be easily explained by instantaneous source mixing. We measured all real-data FIT in cases with a delay in stimulus-feature info latencies between $X$ and $Y$ ($X$ to $Y$ info latencies: MEG: 17-35ms between V1 and higher areas, Fig. S9B. EEG: 25ms across hemispheres (Fig 4B). Spike data: 20ms from thalamus to cortex, Fig. S11). Overall, these findings speaks against dominant mixing of a feature-informative source in our analyses.

Finally, we simulated the case with real FIT between two "pure" signals $Z_1$ and $Z_2$ that are unevenly mixed in the measured $X,Y$:

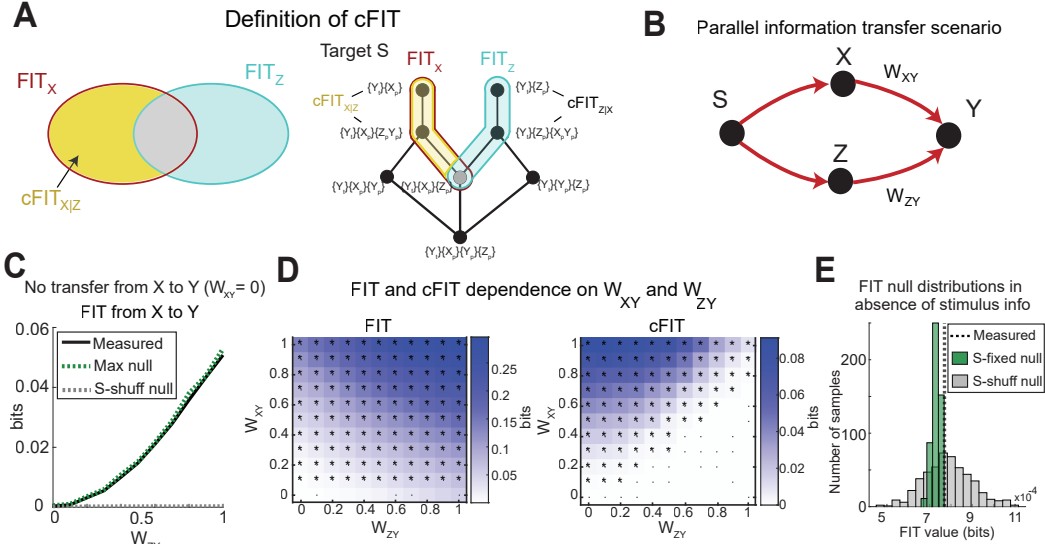

Figure S7: Simulation tests of the significance of FIT and cFIT. (A) Schematic of the cFIT definition. Left: intuitive definition with set-theoretic diagrams. Right: breakdown of $FIT_X$ (in red) and $FIT_Z$ (in light blue) into finer information atoms considered in the definition of $cFIT_{X|Z}$ (atom that can be part of $cFIT_{X|Z}$ are indicated in yellow). Only atoms of $FIT_X$, $FIT_Z$ and $cFIT_{X|Z}$ belonging to the PID having $S$ as target variable are shown. $FIT_X$ is the feature information transmitted from $X$ to $Y$, $FIT_Z$ is the one transmitted from $Z$ to $Y$, $cFIT_{X|Z}$ is the cFIT from $X$ to $Y$ conditioned on the stimulus-feature information of $Z$. (B) Schematic of the scenario implemented in the simulations: both $X$ and $Z$ transmit feature information to $Y$. (C) FIT dependence on the amount of stimulus-feature information transmitted from $Z$ to $Y$ ($W_{ZY}$) even when simulating a case ($W_{XY} = 0$) in which there was no within-trial transmission from $X$ to $Y$. FIT grows with $W_{ZY}$, but its value is always non significant using the permutation null hypothesis described in Section SM1.7. The dashed green line shows the 99th percentile of the FIT null hypothesis distribution described in Section SM1.7. For comparison, the dashed gray line shows the 99th percentile of the null-hypothesis distribution that would have been obtained simply shuffling $S$ across all trials. The fact that the latter remains so low across all values of $W_{zy}$ highlights the need of using a shuffling procedure that preserves the stimulus-feature information in the individual nodes, as we did in this paper and described in Section SM1.7 (D) FIT and cFIT as function of feature-related transmission from $X$ to $Y$ ($W_{XY}$) and from $Z$ to $Y$ ($W_{ZY}$). * indicate significant values ($p < 0.01$, permutation test) for the considered parameter set. In Panels C,D, results plot mean across 50 simulations (2000 trials each). (E) Example of shuffled distributions in a case in which there is not stimulus-feature information in $Y$. While the null hypothesis values of permuting $X$ at fixed $S$ give much more conservative and effective null hypothesis values when the analyzed network has stimulus-feature information across the nodes (see panel C), in specific cases of no feature information in parts of the network it may be safer to consider also the permutation of $S$ across all trials, as this has more available independent permutations form the data and thus gives wider distributions. The example is with the simulations performed in Fig. 2A, for the set of parameters ($W_{stim} = 0, W_{noise} = 0.6$).

$$X = Z_1 + AZ_2 + E_x$$
$$Y = Z_2 + BZ_1 + E_y$$
(S44)

Since adding a new feature-informative channel ($Z_1$ to $Y$ and $Z_2$ to $X$) increases the stimulus-feature information in $X,Y$, we set the standard deviation of independent Gaussian noise $E_x,E_y$ to equalize SNR of $X$ and $Y$ across the simulated parameters space. We found (Fig. S8C) that mixing ($A, B > 0$) reduced FIT and TE compared to the pure case ($A = B = 0$). However, the correct direction of information transfer was always detected for all mixtures. Thus, FIT is reasonably conservative and robust to this mixing.

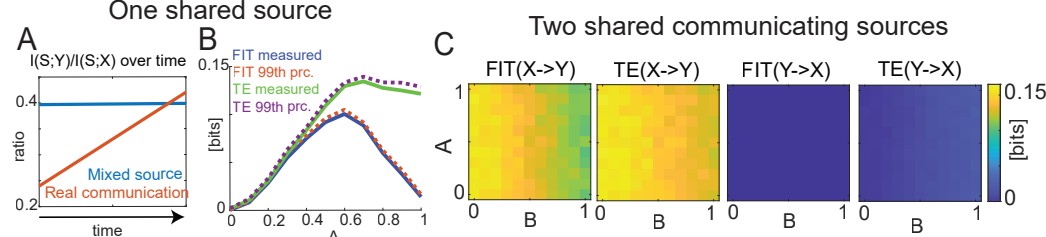

Figure S8: Simulated tests of FIT and TE in the presence of source mixing. A-B: results of the null model with unequal sharing of one feature-informative source $Z$. (A) Proportion of stimulus-feature info with only the null model (blue line, described in SM2.7 and eq. S43) and with real transmission of feature information $X \rightarrow Y$ added to it (red line). (B) FIT and TE of null model (solid lines; see eq. S43) fall below the permutation's test 99th percentile (dashed lines) for any proportion of sharing A. (C) FIT and TE computed from a model with two stimulus-feature informative sources $Z_1$, $Z_2$ communicating $Z_1 \rightarrow Z_2$ and mixed into $X$ and $Y$, vs mixing proportions $A, B$ in $X$ and $Y$ (see eq. S44). All panels plot results averaged over 50 simulations.

.

## SM3    Details and further analyses of experimental data

### SM3.1    MEG data

#### SM3.1.1    Behavioral task and MEG recordings

We analyzed a publicly available MEG dataset [39], with source-reconstructed data available at [https://doi.org/ 10.6084/m9.figshare.12770366]. Full details (including details of approval from responsible ethical review board) are reported in the original publication and are briefly summarized here. The MEG data were recorded from 15 participants performing a visual contrast comparison decision-making task. Each participant´ performed four experimental sessions with on average 429 trials per session. At the beginning of each trial, a reference stimulus was shown. The reference stimulus was a circular, expanding or contracting (randomized across trials) grating with a fixed contrast of 50% and a duration of 400 ms. The reference was followed by a test sequence made up of ten sample gratings (100ms duration each). The contrast of each sample in the sequence was drawn from a Gaussian distribution, the mean of which was either larger or smaller than 50% (randomly selected per trial). At the end of each trial, participants reported whether the average contrast of the sample gratings was higher or lower than the reference contrast. A staircase procedure was used to adjust the mean of the Gaussian distribution setting the average contrast of the 10 samples in each trial, by making decisions harder (mean of the Gaussian closer to 50%) or easier (mean of the Gaussian further from 50%) depending on the behavioral performance of the participant until that moment in the experimental session. The staircase was set to obtain a behavioral performance of approximately 75% correct on each session. The Regions Of Interest (ROIs) in MEG source space used to identify signal from the considered brain areas were defined based on the atlas from Glasser and colleagues[13]. All ROIs were co-registered to individual structural Magnetic Resonance Imaging (MRI) data. Source reconstruction was performed using linearly constrained minimum variance (LCMV) beamforming, by combining leadfield matrices and data covariance matrices (CMs) into a spatial filter for each source position (i.e., vertex) that was applied to the sensor-level data to compute the source estimate. To this end, the leadfield matrices were computed from 3-layer boundary element head-model (conductivity 0.3, 0.3, 0.006 $S/m$ for scalp, brain, skull respectively) based on the individual structural MRIs. CMs were computed for the interval 0s to 1.35s from stimulus onset from the pre-processes and artifact-cleaned [39] broadband MEG data (275x275 sensors) with a regularization of 5% of CM. The source space was constrained to the cortical sheet (4096 vertices per hemisphere), and source orientations were chosen to maximize the power at each vertex. To illustrate the spatial resolution of source reconstruction, in Fig. S9A we plotted the correlation between LCMV spatial filters of neighboring sources vs their distance, finding a very small correlation ($< 0.02$) at distances larger than 2.5cm (as expected from theoretical considerations [15]). To compute FIT and TE, the time-frequency representation of sensor data was projected into source space and averaged over vertices within the ROI (80,20,10 vertices for V1, V3A, LO3, respectively). All FIT analyses

presented in this paper focused on three visual cortical ROIs that were all more than 2.8cm apart from one another (i.e., minimizing spatial filter leakage) and encoded large stimulus information: primary visual cortex (V1), area V3A (which carried maximal stimulus information in the "Dorsal Stream Visual Cortex" group [13]) , and area LO3 (which carried high stimulus information in the "MT+ and Neighboring Visual Areas" group [13]).

### SM3.1.2  Parameters and details of the Information theoretic analyses

For the analysis of FIT and TE we used gamma-band instantaneous power obtained by computing the time-frequency representations of single-trial data via the multi-taper method and then averaging the obtained powers in the $[40 - 75]$ Hz band, exactly as described in the original publication [39]. To estimate the joint probability distributions of the neural activity and the stimulus (or the choice) used to compute the information theoretic quantities, we binned the MEG gamma power from each ROI into 2 equally populated bins and then computed empirically the frequency of occurrence of each response bin across all available trials. The stimulus features used for the information analyses was the average contrast of the 10 samples presented on each trial, discretized into two values. (We coded 0 the average contrast if it was below the reference contrast and 1 the average contrast if it was above the reference contrast). The choice feature used for the information analyses was the binary choice reported by the participant in each trial, with the choice "average contrast stronger than reference" coded as 1 and the choice "average contrast weaker than reference" coded as 0. We computed the information quantities for both the feedforward and the feedback direction for the left and the right hemisphere separately and then averaged the two.

Unless otherwise stated, we computed information quantities using all available trials (correct and error trials). For the specific set of information analyses comparing correct and error trials (Fig. 3G,H), we randomly subsampled correct trials so that the number of correct and error trials used to compute the information quantities was the same for each session. In this way, the information values for correct and error trials can be compared fairly because their difference cannot reflect possible differences in limited-sampling biases due to different data numerosity [26].

### SM3.1.3  Statistical analyses

We established significance of the information measures in the time-delay space using a cluster-based nonparametric statistical test described in Section SM1.7, see [24; 8].

To provide a quantification values of TE and FIT across participants, sessions and network links (Fig. 3D,F,H), we selected a rectangular region in the time-delay domain to select the TE and FIT values for the across sessions statistics, centered around the FIT significant cluster (FIT-specific region). We computed the average over delays and then picked the maximum over time within this region. This gave us one single TE and one single FIT value for each hemisphere in each session. The comparisons of values across participants, subjects and links was performed using two-tailed paired t-tests.

### SM3.1.4  Additional results

Here we list the results of a number of additional analyses that could not be inserted in the main text due to lack of space but that are helpful to better understand and support the conclusions presented in the main text.

The first set of results regards the encoding of information in individual regions of the visual network, rather than the transmission across regions of information about the stimulus. We reported temporal profiles of stimulus information in the three selected ROIs in Fig. S9B. Instantaneous information profiles showed a clear lag in the onset of stimulus information that could not be explained by instantaneous source mixing (see Section SM2.7). The amount of mutual information about the stimulus carried by the power of the gamma band in the visual cortical network is larger in the first half of the presentation of the stimulus ([0-500]ms peri-stimulus, denoted as 'early') than in second half of the presentation of the stimulus ([500-1000]ms peri-stimulus, denoted as 'late') within the trial (Fig. S9C, left). This is why we concentrated the FIT analyses on the first part of the trial (the early window). In the early part of the trial, the gamma band activity in the visual cortical network carries more stimulus than choice information (Fig. S9C, middle) and this information is higher in correct compared to error trials (Fig. S9C, right) . These results are useful to confirm that stimulus

information coding is of more prominent importance in the visual network and that the presence of this information is key to perform accurate perceptual discriminations.

The second set of results regards additional findings about the transmission across regions of information about the stimulus. We produced network representation showing the relative strength of individual TE and stimulus FIT links contributing to the observed differences in directionality (Fig. S9D, compare with Fig.3D) and feedforward behavioral relevance (Fig. S9E, compare with Fig.3E) of information transmission. We found no difference of FIT stimulus nor TE in the feedback direction between correct and error trials (Fig. S9F). For the example pair V1-V3A we identified a significant cluster of stimulus FIT feedforward (V1 to V3A) but not feedback (V3A to V1) in the time-delay domain (Fig. S9G; cluster statistics, p<0.01). These results suggest that feedforward propagation of stimulus information, but not feedback propagation, is specifically key for correct behavior.

In the main text we indicated that the time delay region in which FIT about the stimulus was significant was in the interval 200 to 400ms after the stimulus onset, with an inter-area communication delay between 65 and 250ms. This statement is supported by the plot in Fig. S9G of the time-delay map points that are significant according to the cluster permutation test.

Finally, we performed a control analysis to quantify TE in a time-delay region around which TE was maximal. This is of interest because in the main text analyses (Figure3E-H) we compared FIT and TE using a time-delay region around the peak of FIT. The TE panel (Fig. S9H) shows that TE peaks in a different time-delay region with respect to stimulus FIT (Fig. 3C). Taking a DI-specific box centered around the TE peak in time-delay to select information values we could not assess the direction nor the behavioral relevance of information transmission (Fig. S9I).

### SM3.2  EEG data

### SM3.2.1  Behavioral task and EEG recordings

We next analyzed a publicly available EEG dataset [31]. Data are available at [https://datadryad.org/stash/dataset/doi:10.5061/dryad.8m2g3].  Full details (including details of approvals from Ethical Committees) are reported in the original publication. Here we summarize them briefly.  The EEG data were recorded while participants (N=16) performed a face detection task. Participants were presented with an image hidden behind a bubble mask that was randomly generated in each trial. The presented image was a image of a face in half of the trials and a random texture in the other half of the trials. Participants were instructed to report whether a face was present or not.  In our analyses, we only considered correct trials where the face was correctly detected by the participants (approximately 1000 trials per subject). Following the recommendations of the original publications analysing these data [31; 18], we excluded one participant from the analysis due to a poor EEG signal that did not contain significant eye visibility information in any of the electrodes.  All analyses in our paper are based on the N=15 selected participants.  EEGs were recorded by fitting participants with a Biosemi head cap comprising 128 EEG electrodes. EEG data were re-referenced offline to an average reference, band-pass filtered between 1 Hz and 30 Hz using a fourth order Butterworth filter, down-sampled to 500 Hz sampling rate and baseline corrected using the average activity between 300ms pre-stimulus and stimulus presentation. ICA was performed to reduce blink and eye- movement artifacts, as implemented in the infomax algorithm from EEGLAB [10]. Components representing blinks and eye movements were identified by visual inspection of their topographies, time courses, and amplitude spectra.

### SM3.2.2  Details of the information theoretic analyses and additional results

For the analyses of TE and FIT, we selected the EEG electrodes in the left and the right Occipito-Temporal regions that had the highest mutual information about the visibility of the contra-lateral eye, exactly as done in previous papers [18]. (Specifically, in Ref. [18], the authors used the following criteria to select one electrode in LOT and one in ROT for each participant, see Fig. S13A. For LOT they selected the electrode with maximum right eye MI from electrodes on the radial axes of P07, P7, and TP7, excluding midline Oz and neighboring O1 radial axes. On the right hemisphere, for ROT the author selected the EEG electrode with maximum left eye information from sensors on the radial axes of PO8, P8, TP8, excluding midline Oz and neighboring O2 radial axes). We computed the first derivatives of the EEG signal for both Occipito-Temporal sensors and used both its absolute values

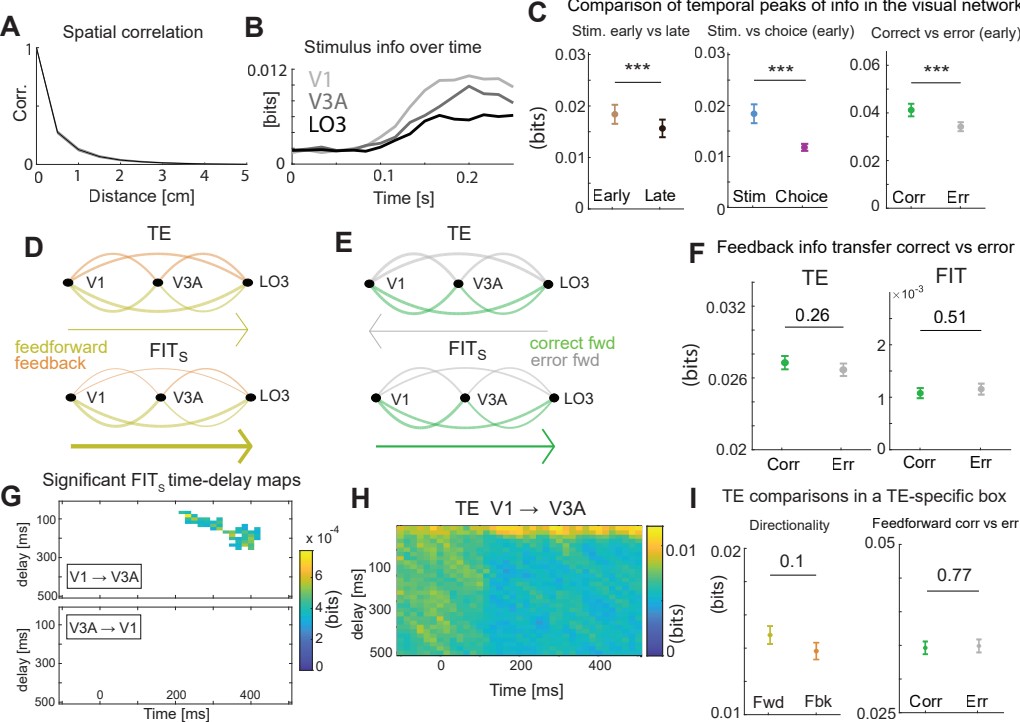

Figure S9: Additional analyses of the MEG dataset. (A) Correlation between LCMV spatial filters of neighboring sources as a function of their distance. (B) Time course of stimulus information in each ROI. (C) Properties of mutual information between stimulus and MEG activity. Left: Peak values of stimulus information over time in the first (early time window, from 0 to 500ms post-stimulus onset) and second half (late time window, 500 to 1000ms after stimulus onset) of the stimulus presentation window. Middle: Peak of stimulus and choice information carried by MEG gamma activity in the early window. Right: Peak of stimulus information values encoded by MEG gamma activity in the early window in correct and incorrect trials, respectively. (D) Graphs representing the strength of feedforward (yellow) and feedback (orange) information transmission in the network for TE (top) and stimulus FIT (bottom). Links are weighted proportionally to the communication strength between each pair. The arrows on the bottom points toward the dominant direction of overall transmission, and are weighted proportionally to the difference between feedforward and feedback transmission. (E) Same as D but for feedforward transmission in correct (green) vs error (gray) trials. (F) Values of TE (left) and of FIT about the stimulus (right) computed in the feedback direction separately in correct and in error trials. (G) Plot of the points with significant values of the stimulus FIT between V1 and V3A (top) and V3A and V1 (bottom) according to a cluster permutation test. Only points that are significant are colored. Color scale is the same as the Fig. 3C. (H) TE time-delay map in the V1 to V3A direction. (I) Values of TE using a time-delay box centered around the TE peak in the time-delay map. In all panels, lines and image plots show averages and errorbars SEM across participants, experimental sessions and regions pairs (in case of FIT and TE) or regions (in case of mutual information). *: p<0.05, **: p<0.01, ***: p<0.001. All information-theoretic quantities were computed from power time courses of source-level MEG signals in the gamma-band ([40-75]Hz), first computed separately for left and right hemisphere and then averaged.

and first derivatives to compute the information quantities, for consistency with the information-encoding analyses performed in a previous study [18].As stimulus feature for the computation of mutual information and FIT, we used the visibility of an eye (defined as the fraction of pixels within the eye region that were not hidden by the bubble mask). This feature was discretized using 2 equipopulated bins. We computed the information quantities for all combinations of directionality of flow across hemispheres (left to right, right to left) and eye identity (left or right eye). We computed significance of FIT in the time-delay using the cluster-based permutation test described in Section SM1.7. This analysis revealed a significant cluster of FIT about the left eye in the right-to-left

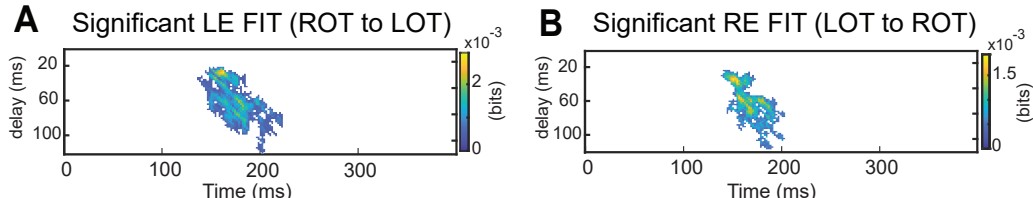

Figure S10: (A) Plot of the points with significant values of the left-eye (LE) visibility FIT from the EEG of Right Occipito-Temporal (ROT) to Left Occipito-Temporal (LOT) electrodes, according to a cluster permutation test. Only points that are significant are colored. (B) Same as A but for the Right-Eye (RE) visibility FIT from the EEG of Left Occipito-Temporal (LOT) to Right Occipito-Temporal (ROT) electrodes. The color scale of panel A and B is the same as the corresponding plots in Figure 3B and C, respectively.

direction (Fig. S10A) and a about the right eye in the left-to-right direction (Fig. S10B). To provide a quantification values of TE and FIT across participants (Fig. 4D), we selected a rectangular region in the time-delay domain to select the TE and FIT values for the across participants statistics, centered around the contra-lateral FIT significant cluster (same for both eyes, as they were significant in very similar time-delay regions). We computed the average over delays and then picked the maximum over time within this region. This gave us one single TE and one single FIT value for each subject. The comparisons of values across participants was performed using two-tailed paired t-tests.

## SM3.3    Analysis of spiking activity in a thalamocortical network

### SM3.3.1    Electrophsyiological experiments

We analysed previously published [5] recordings of multi-unit spiking activity simultaneously obtained from electrodes placed in the primary visual cortex (V1), primary somatosensory cortex (S1), first-order visual thalamus (the lateral geniculate nucleus, LGN), and the first-order somatosensory thalamus (the ventral posteromedial nucleus, VPM) of anaesthetized rats (Fig. 3A). Data are made available with this NeurIPS submission as Supplemental Material. Full details (including details of approvals from Ethical Committees and Local Authorities) are reported in the original publication. Here we summarize them briefly.

These data were recorded from N=6 rats (using one-shank Silicon Michigan probes, Neuronexus Technologies; 100-$\mu$m intersite spacing) in three stimulation conditions: visual stimulation, whiskers tactile stimulation and bimodal stimulation (simultaneous visual and tactile). All experiments were conducted under urethane anesthesia. The visual stimuli consisted of a light flash (50-ms-long LED light flashes at 300 lux). The unimodal somatosensory stimulus consisted of a whisker deflection. For bimodal stimulation, whisker deflection and light flashes were applied in the same hemifield. Stimuli were randomly presented across trials. In our analysis, we considered only stimulation contra-lateral to the recorded brain areas. Each type of stimulus was presented 100 times. The non-stimulated eye was covered with an aluminum foil patch. Neural activity was recorded at a sampling rate of 32 kHz, bandpass filtered (0.1 Hz and 5 kHz) then down-sampled to 8 kHz. In the current work, we used the recordings from infragranular layers of S1 and V1 and from VPM and LGN. Multi-unit spike times were first detected from the band-passed (400–3,000 Hz, fourth-order IIR Butterworth Filter) extracellular potential in each electrode by threshold crossing (>3 SD). A spikes train was obtained for all channels using a temporal binning of 0.125 ms (1/8kHz). For each brain region, spiking activity was then pooled together using all recorded spikes form all electrodes related to that region.

### SM3.3.2    Parameters and details of the information analyses

To compute mutual information and FIT, we defined two different stimulus set of interest. To measure information related to tactile discrimination, we used a "tactile-discriminative set" made of the unimodal visual and the bimodal visual-tactile stimulus (the two stimuli in the set are discriminated by the presence or absence of a tactile stimulus). Similarly, to measure information related to visual discrimination, we used a "visual-discriminative set" made of unimodal tactile and the bimodal

visual-tactile stimulus (the two stimuli in the set are discriminated by the presence or absence of a visual stimulus).

As stimulus feature for the computation of mutual information and FIT for the tactile (visual) discriminative set, we used a binary value indicating either the delivery of a visual (tactile) unimodal or a bimodal stimulation. We computed the information quantities for all combinations of directionality of flow across the visual (LGN to V1 and V1 to LGN) and somatosensory (VPM to S1 and S1 to VPM) thalamo-cortical pathways and stimulus-discriminative sets (visual or tactile). To provide a quantification values of TE and FIT across animals (Fig. S11E,F,H,I), we selected a rectangular region in the time-delay domain to select the TE and FIT values for the across animals statistics, centered around the FIT peaks about the tactile stimulus in the VPM to S1 direction (for the somatosensory pathway) and about the visual stimulus in the LGN to V1 direction (for the visual pathway). We computed the average over delays and then picked the maximum over time within this region. This gave us one single TE and one single FIT value for each animal. The comparisons of values across animals was performed using two-tailed paired t-tests.

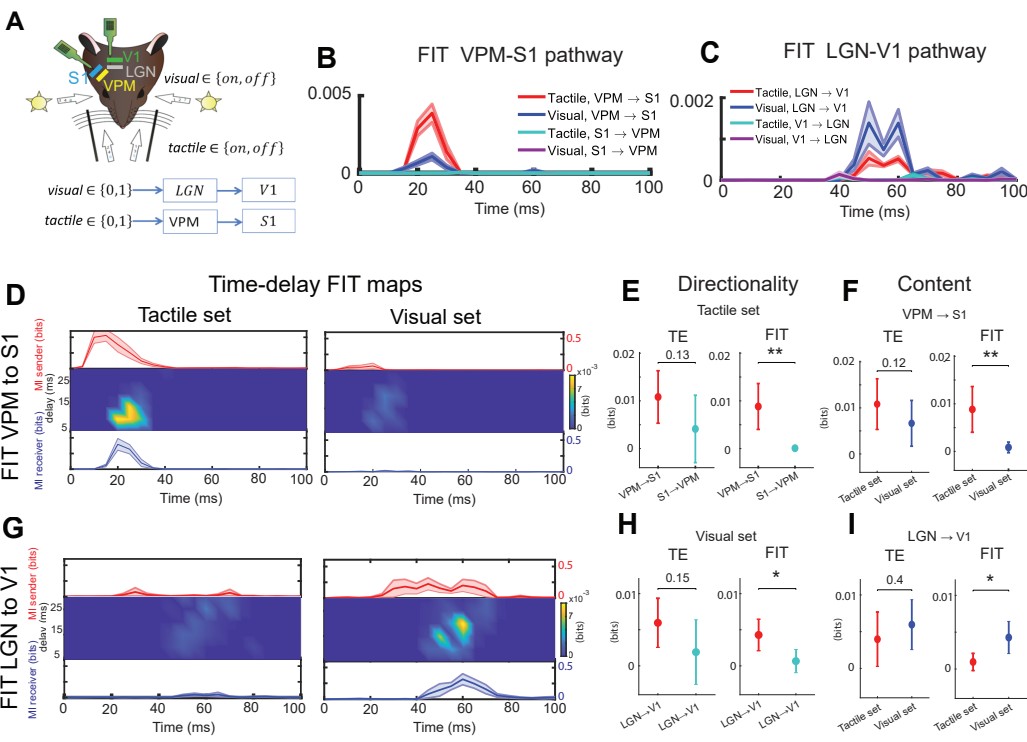

Figure S11: Sensory related info transfer carried by multi unit actity (MUA). (A) Schematic of the experimental setup. MUA was recorded in rats from the Ventral Posteromedial nucleus (VPM) of the thalamus, the Lateral Geniculate Nucleus (LGN) of the thalamus, the primary somatosensory (S1) and visual (V1) cortex simultaneously. During the recording either a unimodal tactile, a unimodal visual, or a bimodal (visual and tactile) stimulus was presented. (B) FIT values averaged across delays for all combinations of transfer direction and stimulation type on the somatosensory pathway (VPM-S1). (C) Same as B but for the visual pathway. (D) FIT from VPM to S1 (mean across subjects) for each value of delay and post-stimulus time. Line plots above and below show Mutual Information (MI) between the presented stimulus and the recorded MUA in the VPM and S1, respectively. Width of shaded errorbars show the SEM of the measure across subjects. The left panel reports values of information and FIT for the tactile-discriminative set, whereas the right panels report values of information and FIT about the visual stimulus set. (E) Directional sensitivity for TE (left) and FIT (right) between VPM and S1, for the Tactile stimulus-set. (F) Comparisons between tactile- and visual-discriminative set, for the TE (left) and the FIT (right) from VPM to S1. (G) Same as panel D but from LGN to V1. (H) Same as panel E but between LGN and V1, for the Visual stimulus-set. (I) Same as panel F but for TE and FIT from LGN to V1.

### SM3.3.3  Further details about the information-theoretic results

We first focused on the somatosensory pathway (VPM and S1). We found that, on this pathway, the tactile FIT in the VPM to S1 direction was visibly higher than the visual FIT in the same direction and both tactile and visual FIT in the S1 to VPM direction (Fig. S11B) We found that the timing of tactile FIT from VPM to S1 was consistent with the one of tactile information in neural activity, that was present in the 5-30ms and 15-30ms post-stimulus intervals in VPM and S1, respectively (Fig. S11D left, top and bottom lines). We found that FIT revealed the directionality of tactile information flow, which was significantly larger in the feedforward (VPM to S1) compared to the feedback (S1 to VPM) direction (Fig. S11E right, p = 0.0065). On the contrary, TE computed for the tactile set was not significantly different in the two directions (Fig. S11E left, p = 0.13). FIT also revealed the content of communication from VPM to S1, being significantly larger for the tactile set than for the visual set (Fig. S11F, right; p = 0.0084), while TE from VPM to S1 was not significantly different in the two directions (Fig. S11F, left; p = 0.12).

Complementary results were found in the visual pathway (LGN and V1). On this pathway, the visual FIT in the LGN to V1 direction was visibly higher than the tactile FIT in the same direction and both tactile and visual FIT in the V1 to LGN direction (Fig. S11C). FIT for the visual-discriminative set in the LGN to V1 direction peaked in a time interval of approximately 45 to 65ms after stimulus-onset and with a transfer delay of approximately 10-25ms Fig. S11G). The visual FIT values were larger in the feedforward than in the feedback direction (Fig. S11H, right; p = 0.033), while TE was not sensitive to the directionality of visual information (Fig. S11H, left; p = 0.15). Moreover, visual FIT values were significantly larger than the tactile ones from LGN to V1 (Fig. S11I, right; p = 0.013), while TE did not capture these sensory modality-specific differences (Fig. S11I, left; p = 0.4). Taken together, these results highlight the power of the FIT in revealing feature- and direction- specific transfers of information with high temporal precision, beyond what is achievable using methods that measure the total propagation of neural activity such as TE.

### SM3.4  Applications of cFIT to real neural data

We tested the effectiveness of the conditional FIT (cFIT) in the analysis of neurophysiological data by applying it to perform further analyses on the EEG and the MUA datasets (Fig. S12). For simplicity, when computing cFIT from $X$ to $Y$ conditioned on the stimulus information of $Z$, we always considered the same communication delay (that is, the time difference between the present and past activity used to compute the information theoretic measures) for the past activity of the sender $X$ and the past activity of the third region $Z$. However the definition of conditional FIT holds for an arbitrary representation of $Z_{past}$, potentially including multiple time points or a communication delay that is different from the one of $X_{past}$.

We first applied cFIT to the EEG dataset. We investigated whether the contra-lateral Occipito-Temporal electrodes used to compute TE and FIT in Figure 4 where the sole senders of eye-specific information across hemispheres. We selected two different sets of putative alternative senders of eye-specific information and used cFIT to remove the contribution of the putative alternative senders from the contra-lateral FIT that we measured. Namely, we selected the third location to be conditioned upon from either a set of weak or a set of strong alternative senders for both the left and for the right eye (Fig. S13). For each participant, we defined the two weak alternative sender locations (one for the left an one for the right eye) as those electrodes carrying the lowest amount of stimulus information about the left or the right eye, respectively, in the frontal lobe of the brain. The expectation was that removing the contribution of these electrodes using cFIT would not change appreciably the results obtained with the contra-lateral unconditional FIT reported in Figure 4.

For each participant, we defined the strong alternative senders locations (one for the left an one for the right eye) as those electrodes carrying the second-largest amount of information about the left eye in ROT or about the right eye in LOT (ROT and LOT defined as in Ref [18]). We found that FIT conditioned on the contra-lateral-eye information of one of the weak alternative senders (the orange lines in Fig. S12A) did not reduce FIT, as the cFIT was virtually equal to the unconditional FIT (the blue trends in Fig. S12A). However, FIT conditioned on the contra-lateral-eye information of one of the strong alternative senders (the green trends in Fig. S12A) was lower than unconditional FIT. However, both cFIT given the weak and the strong alternative senders were significant (cluster statistics over time, p<0.01). The fact that cFIT was lower than FIT when conditioning on informative electrodes but not when conditioning on weakly informative electrodes suggests that cFIT is effective

at removing influences related to similar feature-specific (but not un-specific) information present already in the past activity of other regions. The fact that the inter-hemisperic Occipito-Temporal controlateral-eye-specific cFIT is still highly significant and is only marginally smaller than the original unconditional FIT suggests that most eye-specific information flows across hemispheres through the contra-lateral Occipito-Temporal electrodes selected in [18].

Next, we analyzed the spiking activity dataset. We examined tactile-discriminative information flowing through the somatosensory pathway, and the visual-discriminative information flowing though the visual pathway.

We first used cFIT to test whether the tactile-discriminative FIT from the somatosensory thalamus VPM to the somatosensory cortex S1 could have actually been relayed through the visual thalamus LGN. The neurophysiological expectation is that all tactile-discriminative information flows within the somatosensory pathway, without contributions from visual stations. Consistent with this expectation, we found that the tactile-discriminative cFIT from VPM to S1 conditioned on the visaul thalamus LGN was equal to the unconditional tactile-discrminative FIT from VPM to S1.

We then used cFIT to test whether the visually-discriminative FIT from the visual thalamus LGN to the visual cortex V1 could have actually been relayed through the somatosensory thalamus. The neurophysiological expectation is that all visual-discriminative information flows within the visual pathway, without contributions from somatosensory stations. Consistent with this expectation, we found that the visually-discriminative cFIT from LGN to V1 conditioned on the somatosensory thalamus VPM was equal to unconditional visually-discriminative FIT from LGN to V1.

Together, these results suggest that cFIT is useful to remove contributions from alternative pathways specifically with regard to the transmission of feature-specific information.

## SM4    Comparison with other possible or previously published measures

We examine how FIT differs with respect to other possible or previously published algorithms that were designed to identify the information flow across regions about behavioral or stimulus features of interest. We first consider two measures that implement the Wiener-Granger discounting of the information present in the past activity of the sender. We then consider two other methods, that did not implement this principle.

### SM4.1    Comparison with variations in transfer entropy $\Delta TE$

As mentioned in the main text, one simple-minded proxy for identifying feature-specific information flow could be quantifying how the total amount of transmitted information (TE) is modulated by the stimulus-feature [4]. For the case of two stimuli, this amounts to the difference of TE computed for each stimulus-feature value.

We now show, using simulations, that this measure can fail in capturing feature-related information flow. We performed simulations in a scenario having variable degrees of both feature-specific and feature-unrelated information transfer. The encoded and transmitted stimulus feature $S$ was a stimulus-intensity integer value (1 or 2). The activity of the sender $X$ was a two-dimensional variable with one stimulus-feature-informative $X_{stim}$ and one stimulus-uninformative component $X_{noise}$. The feature-informative dimension had a temporally-localized stimulus-dependent bump in the activity (from 200 to 250ms) and additive Gaussian noise. The stimulus-unrelated component was, at any time point, a zero-mean Gaussian noise. The activity of the receiver $Y$ was the weighted sum of $X_{stim}$ and $X_{noise}$ with a delay $\delta$, plus Gaussian noise. The delay $\delta$ was chosen randomly in each simulation repetition (N=50) in the range 40-60ms. We tested whether $\Delta TE$ across simulation repetitions was significantly different from zero using a two-tailed t-test.

As we did in Figs. 2 and S3 for FIT and TE, we studied the behavior of $\Delta TE$ as a function of the simulation parameters $W_{stim}$ (which increases the amount of information transferred about the stimulus feature) and $W_{noise}$ (which increases the amount of feature-unspecific information that is transferred from $X$ to Y). We found (Fig. S14A) that $\Delta TE$ had almost no relationship with the values of $W_{stim}$ and $W_{noise}$, unlike FIT which individuated stimulus-feature-specific transfer correctly because it increased with $W_{stim}$ but not with $W_{noise}$ (Fig. S3A).

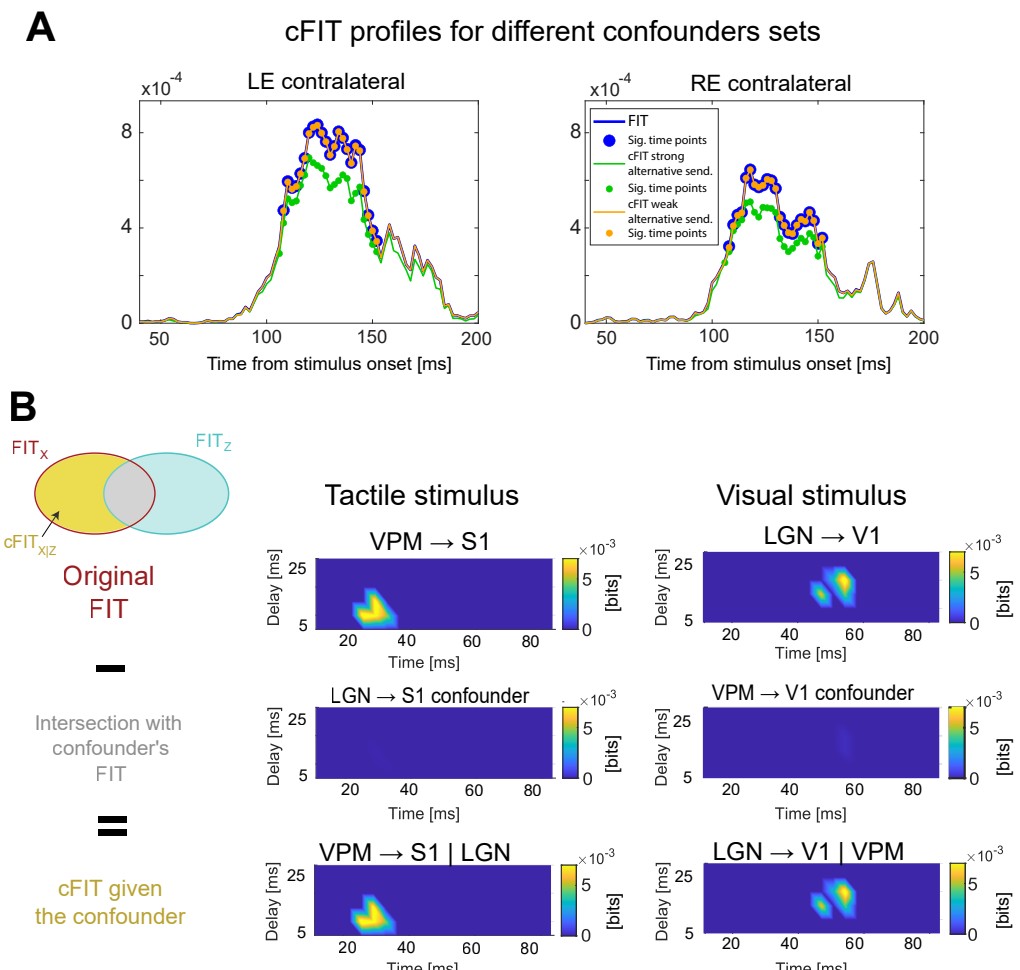

Figure S12: Application of cFIT to experimental data. (A) cFIT application to the EEG data. We conditioned the contra-lateral FIT about the left and the right eye (see Fig. 4) to the activity of two either weak or strong alternative eye-visibility information senders. Temporal profiles of unconditional FIT (in blue) for about the left eye from ROT to LOT (left) and about the right eye from LOT to ROT (right). cFIT temporal profiles when conditioning on weak alternative senders (in orange) and on strong alternative senders (in green). The points where the measures were significant are indicated with a circular marker (p<0.01, cluster statistics). (B) cFIT applied to MUA data. We conditioned tactile- (visual-) discriminative FIT through the somatosensory (visual) pathway (first row) to the activity of the visual (somatosensory) thalamus. The amount of unconditional FIT that was shared with the FIT through the alternative sender (second row) was subtracted from the original FIT to obtain cFIT (third row). The left column shows results for the tactile-discriminative set, the right column for the visual-discriminative set.

Note that we performed also simulations (that were exactly like those of Fig. 2, except that we had 2 rather than 4 stimulus intensity values) in which the noise in $X$ was multiplicative rather than additive. In this case (results not shown) $\Delta TE$ increased with both $W_{stim}$ and $W_{noise}$. Thus, $\Delta TE$ had limited capabilities of identifying some stimulus-feature-specific information transfer in some specific case, but it dot reflect it in general.

The reason that $\Delta TE$ cannot capture feature-specific information flow is, in our view, that $\Delta TE$ is a measure of variation of information strength across stimulus-feature conditions rather than a measure of stimulus-feature-specific information transfer.

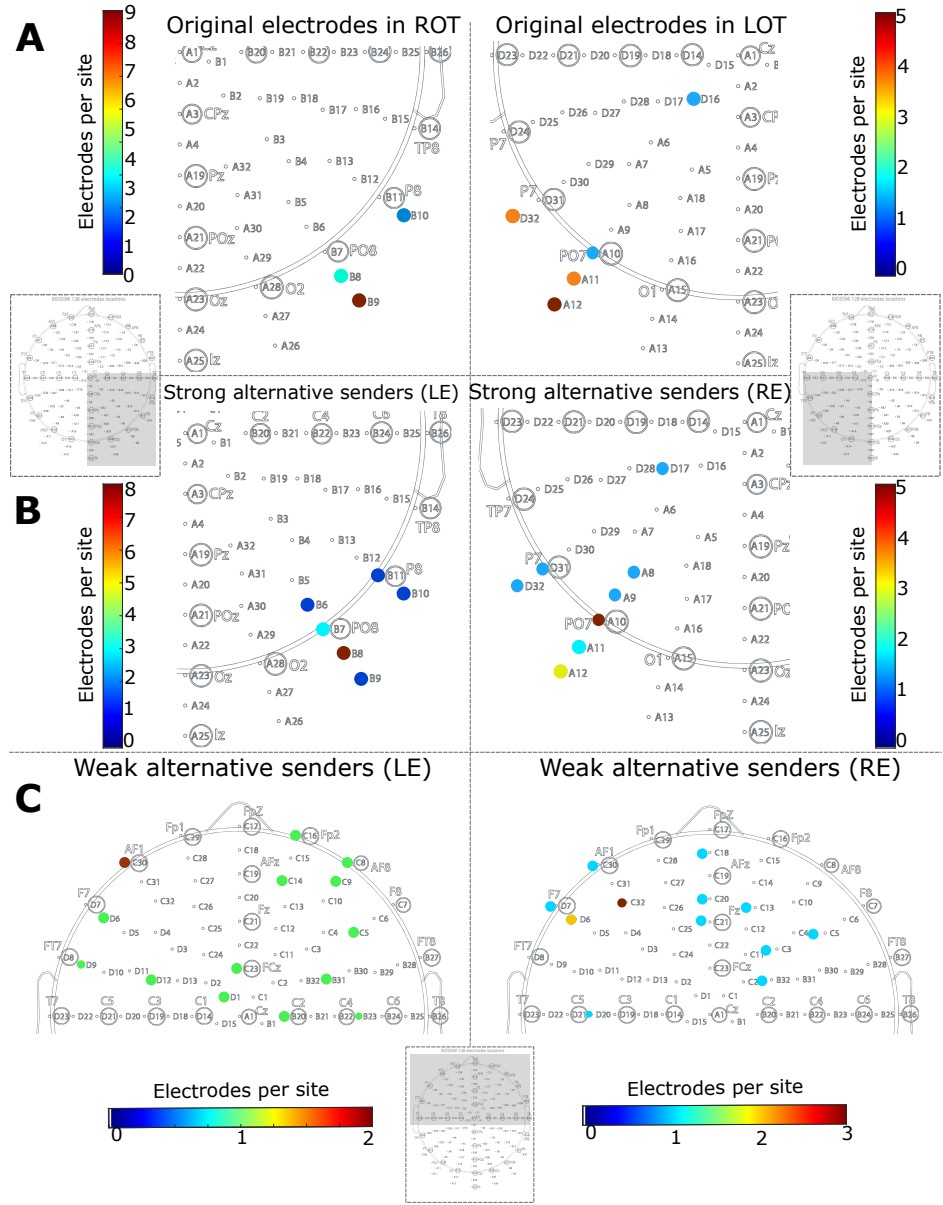

Figure S13: Sets of electrodes used in the EEG data analysis. (A) Electrodes used to measure FIT and TE and for all the analyses in Fig.4, same electrodes used in [18]. (B) Set of 'strong alternative senders', selected as the electrodes carrying the second maximum amount of information about the left eye in the ROT (left panel) and about the right eye in the LOT (right panel). (C) Sets of 'weak alternative senders', selected as the electrodes in the frontal lobe carrying the minimal amount of information about the left eye (left panel) and about the right eye (right panel).

Additionally, we tested $\Delta TE$ on MEG data. We first binarized the stimulus feature into two classes (average contrast either greater, S=1, or lower, S=0, than the reference contrast). We computed TE for all pairs of visual regions in the visual cortical network separately in trials with the same value of the binary stimulus we and computed the difference $\Delta TE$ between these values. Fig. S14B shows that $\Delta TE$ in the visual cortical network had the same strength in the feedforward and feedback direction, unlike FIT that showed a clear directionality of communication of stimulus information (stronger in the feedforward than in the feedback direction). Finally, when computing TE on the spiking activity data of the rat thalamocortical network, we found that TE from thalamus to cortex did not vary between the tactile-discriminative and visually-discriminative stimuli set (see Fig. S11E-I).

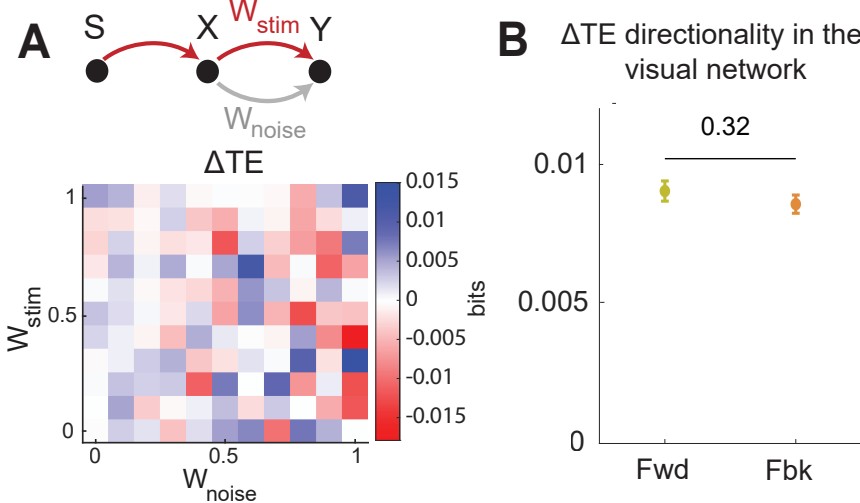

Figure S14: Performance of the Transfer Entropy Difference across stimuli ($\Delta TE$) on simulated and real MEG data. (A) Simulated data. Values (mean across 50 simulations) of $\Delta TE$ from $X$ to $Y$ computed on the same simulated data used in Fig. S3A as a function of the simulation parameters $W_{stim}$ (which increases the amount of information transferred about the stimulus feature) and $W_{noise}$ (which increases the amount of feature-unspecific information that is transferred from $X$ to $Y$). We found that $\Delta TE$ had almost no relationship with the values of $W_{stim}$ and $W_{noise}$, unlike FIT which increases only with $W_{stim}$. All $\Delta TE$ values were not significantly different from zero (two sided t-test across 50 simulations, significance threshold at p = 0.01). (B) Real MEG data. Average across participants, sessions and pairs of regions of the values of $\Delta TE$ (reported values were obtained taking the average over delays and then the maximum over time in the same time-delay region used for the results in Fig. 3D,E,F).

In contrast, on the same data FIT could distinguish tactile-discriminative from visually-discriminative information flow from thalamus to cortex (see Fig. S11E-I).

## SM4.2 Comparisons with Directed Feature Information (DFI)

A previous study [17] defined a measure, Directed Feature Information (DFI), which computes feature-specific information redundant between the present activity of the receiver and the past activity of the sender, conditioned on the past activity of the receiver. However, DFI used a measure of redundancy that actually conflated the effects of redundancy and synergy (see Section SM1.5 where we consider in detail its definition and its PID decomposition). Because of this, DFI can be negative and thus not interpretable as measure of information flow. Moreover, because DFI discounts only past activity of the sender rather than its feature-specific information, it is less precise and less conservative in localizing direction and timing of feature-specific information flow.

The above properties are expected from theoretical considerations but were also demonstrated by us in the following numerical simulations. We computed DFI in the two simulations described in Section SM2.1 (Fig. S15). We found that, in general, DFI had a trend similar to FIT, increasing with the amount of stimulus-feature-related transfer from $X$ to $Y$ ($W_{stim}$) and decreasing with the amount of stimulus-unrelated transfer ($W_{noise}$). However, DFI had several false positives (cases when there was no transmission in the ground truth of the simulated data but it was detected as significant by the algorithm) and also had several false negatives (cases when there was transmission in the ground truth of the simulated data but it resulted as non significant by the algorithm). In comparison, FIT under the same conditions and same simulations had none, see Fig. 2A). More importantly, as a consequence of its inability to include only redundancy and discard synergy, DFI values were very often negative, and could be negative over time both at baseline and during stimulus-feature-related transmission (Fig. S15B).

The limitations of DFI for individuating directed well time-resolved flow of information about specific stimulus features were further tested with the bidirectional information transfer simulations described in detail in Section SM2.4. We remind briefly that in these simulations we simulated a scenario with bidirectional communication between $X$ and $Y$ with stimulus-feature-related transfer from $X$ and stimulus-unrelated transfer from $Y$ to $X$ (Fig. S5A). In brief, both $X$ and $Y$ received information directly from a feature-information-sending node $S$. $X$ received feature information from $S$ early on (between 50 and 90 ms) and $Y$ received feature information from $S$ at a later time (between 110 and 150 ms). $X$ sent its entire activity to $Y$ (therefore communicating its feature information when it became available). $Y$ instead only sent to $Y$ a part of its activity that did not carry feature information. We found that while DFI had a significant positive bump from $X$ to $Y$ in the $[60, 100]ms$ time window, it also had a significant negative bump from $Y$ to $X$ in the time window in which $X$ encoded the feature $[50, 90]ms$. Crucially, the presence of significant DFI from $Y$ to $X$ preceding in time the DFI from $X$ to $Y$ would be interpreted that there is a bidirectional flow of stimulus-feature information, occurring first from $Y$ to $X$ and then from $X$ to $Y$. Therefore, DFI could not capture correctly neither the directionality nor the timing of the stimulus-feature information flow that we put in the simulations.

For FIT, which is a non-negative measure, we always used one-tailed tests to determine whether the measured values were significantly larger than the 99th percentile of the null hypothesis distribution obtained as described in Section SM1.7.

For DFI, which is an unsigned measure, we implemented a two-tailed test. Analogous to our method for FIT, we computed two null hypothesis distributions: one by shuffling $S$ across all trials, and one by shuffling $X$ for fixed values of $S$. We then tested whether DFI was either above the 99.5th percentile of the element-wise maximum or below the 0.5th percentile of the element-wise minimum of these two null hypothesis distributions. If one of these conditions was met, we assigned significance to DFI.

Lastly, we computed DFI on the three real datasets (MEG, EEG and spiking activity) presented in the main text. We found (Fig. S5C-E) that the problems with DFI predicted by mathematics (see Section SM1.5) and encountered simulations are also found in the neural datasets. On real data, DFI was very often negative and it did not detect directionality or feature specificity in cases in which we would expect from previous literature that specificity or directionality should exist.

In the MEG dataset (Fig. S5D), DFI had negative values and thus not interpretable as measure of information transfer. Unlike FIT, DFI could not detect that (as predicted by previous studies) stimulus information is stronger in the feedforward than in the feedback direction, and DFI could not detect that feedforward stimulus information is stronger in correct than error trials (an important result found by FIT).

In the EEG dataset (Fig. S5C), DFI was negative and thus not interpretable as measure of information transfer. The comparison of the DFI results between eye visibility features and directionally of cross-hemispheric transfer could not support the conclusion (predicted by findings in previous literature and confirmed by the FIT analysis) that across-hemisphere information transfer is directional from contra- to ipsi-lateral (DFI does not detected a leading direction of RE information transfer) and is feature specific (DFI does not detected a difference between LE and RE information in the R to L hemisphere communication).

In the thalamocortical spikes data (Fig. S5E), DFI has mostly positive values which are thus interpretable in terms of information transmitted. DFI confirms (though with lower statistical power) the FIT results than in both the somatosensory and visual corticothalamic pathway more information is transmitted feedforward about the corresponding sensory modality (more visual than somatosensory information transmitted from visual thalamus to visual cortex, and more somatosensory than visual information transmitted from somatosensory thalamus to somatosensory cortex). However, DFI failed to demonstrate that, as expected from well-established neurophysiological findings, more information about such simple stimulus features is transmitted from thalamus to cortex than from cortex to thalamus.

In sum, our results lead us to conclude that the definition of redundancy used in DFI that, unlike the more refined one arising from PID, conflates synergistic and redundant effects, leads to major problems predicted by theory and confirmed by simulation and in real data. Our results suggest that DFI is not robust or refined enough to be applied generally and systematically to brain data, and that

the advances provided by FIT with respect to DFI are important not only conceptually but also for the analysis of empirical datasets.

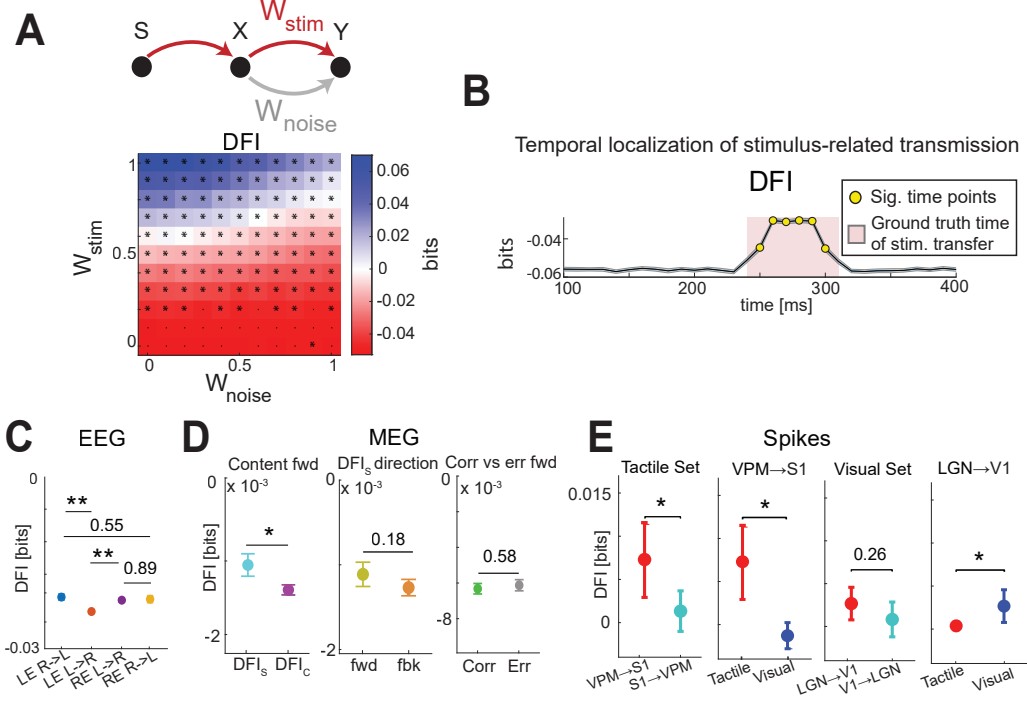

Figure S15: DFI tested on simulated and real brain data. In Panels A,B DFI is tested on the same simulations used for Fig 2A-B. Panel C-E report DFI results on real brain data. (A) DFI as function of stimulus-feature-related ($W_{stim}$) and -unrelated ($w_{noise}$) transmission strength. * indicate significant values ($p < 0.01$, permutation test) for the considered parameter set. (B) Dynamics of DFI in a simulation with time-localized feature-information transmission. Red area shows the window of feature-related information transfer. Yellow dots show time points with significant information (p < 0.01, permutation test). Results plot mean (lines) and SEM (shaded area) across 50 simulations (2000 trials each). (C) EEG DFI between L, R hemisphere about L and R eye visibility (cf with FIT in Fig4D of main paper). (D) MEG DFI about stimulus or choice feedback or feedforward and in correct vs error trials (cf with FIT in Fig. 3D-F). (E) Spikes DFI for the thalamocortical pathway and tactile- or visual-discriminative stimulus set (cf with FIT in FigS8E-I). P-values: 2-tailed paired t-test. *: p<0.05, **: p<0.01.

### SM4.3    Comparison with measures not discounting past information in the receiver as in the Wiener-Granger Causality principle

We finally consider the suitability for identifying flow of information about specific features of possible alternative measures that, although have relevance to feature information coding across areas, *do not* implement the Wiener-Granger discounting of the information present in the past activity of the sender. In brief, methods that do not implement this (and thus just correlate past information of the sender with present information of the receiver), erroneously identify information already encoded in the past activity of the receiver as information transmitted from a sender. This concern would apply in general to all measures that compute time-lagged cross-correlations of activity across areas [14]. In the following, we consider briefly some possible methods that have been used to infer feature-specific information transfer but that do not consider the Wiener-Granger Causality principle.

One possibility would be to measure the presence and timing of feature information (using mutual information between the feature and the activity of the individual area at each instant of time, as in Eq. S1) and then inferring transfer of feature information from $X$ to $Y$ if information about $S$ arises first in $X$ and $Y$. Inferring processing hierarchies on the basis of response selectivity latencies

is a long-established practice in neuroscience [32; 36; 39]. However, the presence of time-lagged information in two areas does not mean that the information in the second area comes from the first area. Indeed, in the simulations to test FIT we created several such simulated scenarios of information present in each area with a different timing but without actual communication between the areas (Fig. S7C, Section SM2.6), and we created non-parametric tests to rule out this possibility based on FIT measures (Section SM1.7). Thus, differential response latencies can be used to hypothesize the presence of processing hierarchies but not to prove transfer of information between specific nodes of the putative information processing network.

Another possibility would be to use PID to measure the presence of shared (or redundant) feature information encoded in both $X$ and $Y$ at different temporal lags. PID can specifically isolate only the information about a feature that is the same, e.g., redundantly encoded, in $X$ and $Y$. Thus, measuring time-lagged shared information goes beyond computing a simple time-lagged correlation between the amount of reach-to-grasp information in $X$ and $Y$, which would not consider whether the time-lagged information content is the same. However, this measure would not discount the presence of the same information in the past activity of $Y$, and it would thus be prone to detecting false communications in case the information was already present in the past of $Y$ and thus could not have come from $X$. (In other words, it would erroneously identify information already encoded in the past activity of the receiver as information transmitted from a sender.) These types of problems of not discounting the past have been illustrated and discussed extensively in the Granger-Causality literature. These considerations apply to a previous study [29] which used PID to attempt to define feature-specific information transmission using the so-called Intersection Information [28], computing information shared between the past activity of the sender and present activity of the receiver (not considering the information already present in the past activity of the receiver).

## SM5 Computational resources

Each of the simulations in Figures 2, S3, S4, S6, S7, S8, S14, and S15 ran in approximately 30 minutes on a personal computer equipped with an Intel i7-10510U processor (4x 1.80GHz CPUs) and 16Gb of RAM, running Windows 10, using MATLAB R2021a. Simulations in Fig. S5 took approximately 3 hours on the same machine.

Real neural dataset analyses ran on a server with an AMD Ryzen Threadripper 3970X processor (32x 3.7GHz CPUs) and 256Gb of RAM, running Ubuntu 18.04, using MATLAB R2019b. The EEG and MEG analyses ran in parallel (using the Parallel Computing Toolbox) over participants or links in the visual cortical network, respectively. Each analysis of the full real datasets (across all participants and experimental sessions) took 12-28 hours depending on the usage of the server.

Our MATLAB codes to compute Feature-specific Information Transfer are provided with this submission and are released under the MIT license. The routines to compute FIT, TE and cFIT are also available at [https://github.com/mcelotto/Feature_Info_Transfer].

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
