# OpenReview forum: "An information-theoretic quantification of the content of communication between brain regions"
_NeurIPS.cc/2023/Conference — NeurIPS 2023 poster_

### Official Review · Reviewer_bTa8 · 2023-07-05

**Soundness:** 3 good
**Presentation:** 3 good
**Contribution:** 3 good
**Rating:** 6
**Confidence:** 4

**Summary:**

In the present work, building on recent advances in Partial Information Decomposition or PID, the authors develop a novel information theoretic measure which they term Feature-specific Information Transfer (FIT) and that the authors claim can capture the feature-specific information transfer between brain regions. The authors validate FIT using synthetic and real world neural recordings.

**Strengths:**

- Originality:  The authors introduce a new measure that even though is based on recent advances on PID is still a significant novel contribution.

- Quality:  the theoretical derivations are solid and there are no obvious flaws that I can see. The introduction of the novel FIT measure is backed by sufficient experimental evidence both with simulated and real world data.

- Clarity: The paper is very clearly written and the illustrations are very good.

- Significance: I think the paper introduces an important measure that can help study stimulus-driven communication between brain areas.

**Weaknesses:**

I think that stating that the FIT measure is feature-specific can be a bit misleading. I would probably have called this measure stimulus-specific rather than feature-specific. I personally reserve the term "feature" to refer to internal neural representations that may be triggered by external stimuli. By that definition, features cannot be observed and it is very hard to attribute certain neural activity to a specific meaningful feature.

**Questions:**

What is the difference between feature and stimulus in this work?

**Limitations:**

Limitations have been properly addressed by the authors.

---

> ### Author Rebuttal · Authors · 2023-08-09
>
> We agree that the nomenclature of the new metric is important and should be chosen with care. We are thus grateful to the Reviewer for raising the issue of how best to name it.
>
> In the present study, we consider as “feature” any variable of interest that is external to the considered neural network. This can be the feature defining an external sensory stimulus (color, contrast, etc) or the feature defining a behavioral output (choice made by the subject, motor or kinematic variable of the subject’s behavior). While some people reserve the use of “feature” for some specific quantification of neural activity (e.g. the timing of spikes, the spike rate, etc), many people especially in vision or hearing reserve the word feature for features of a sensory stimulus. We will better define in the paper’s revision what we mean by feature. We also agree that in the current version of the paper we used the words ‘stimulus’ and ‘feature’ somehow interchangeably, leading to confusion. We will revise the text to avoid any ambiguity between ‘stimulus’ and ‘feature’. E.g. we could replace the letter ‘S’ that we currently use to refer to the external feature in the maths with an ‘F’, we could formulate better sentences such as ‘shuffling X across trials at fixed stimulus’ and replace them with more precise wording such as ‘shuffling X across trials with the same value of the feature’.
>
> However, we would like also to consider and discuss alternatives. We like the suggestion of this referee to call the measure “Stimulus-information transmission” or “Stimulus—specific-information transmission”. However, we define the measure such that it can be applied to information contents that are more general than sensory stimuli (as explained above). In our paper, we applied FIT to compute transmission of information about behavioral choice (Fig 3) as well as applying it to compute transmission of information about sensory stimuli. Using the terminology “stimulus specific” could thus incorrectly suggest that the applicability is limited to sensory function.
>
> Another possibility would be to use “content-specific” rather than “feature’-specific” and call the measure “Content-specific information transfer – CSIT”.
>
> We would prefer to use the current terminology with better clarifications, as explained above. However, we are open to change if it is felt by the reviewers that a change would enhance the clarity about what the measure quantifies. We would warmly appreciate feedback from all reviewers about this issue.
>
> We also hope that the Reviewer will appreciate the advances we presented in this rebuttal in response to the other Reviewer’s suggestions (simulation studies of source mixing, simulation studies of simultaneous encoding of multiple features, computation of DFI on real data, simulation studies scaling of FIT with data size and dimensionality, etc).

---

> ### Author Response · Authors · 2023-08-14
>
> Dear Reviewer, Thanks again for your insights and suggestions, which we greatly appreciated. We wonder whether you received our rebuttal containing the clarifications you requested and the response to your suggestions for improving clarity, as well as other clarifications and extra analyses in response to suggestions of other reviewers. The week in which we can interact is getting to a close and we would really appreciate the opportunity to receive your feedback on our work. Thanks so much!

---

> > ### Comment · Reviewer_bTa8 · 2023-08-18
> >
> > I think that adding clarifications in the text would suffice to address my concern regarding the nomenclature used in the paper.

---

> > > ### Author Response · Authors · 2023-08-21
> > >
> > > Dear Reviewer, thanks again for your valuable insights, which we appreciate and which help us improving the paper. With regard to improving nomenclature, we will add the clarifications about what we mean by feature, as described in our rebuttal.

---

### Official Review · Reviewer_TXME · 2023-07-05

**Soundness:** 3 good
**Presentation:** 3 good
**Contribution:** 3 good
**Rating:** 7
**Confidence:** 3

**Summary:**

Exploring the content and direction of communication between brain regions is key to understanding the brain. This paper proposes a method called Feature-specific Information Transfer (FIT) to investigate the feature-specific content and direction of information flow in multi-region brain recordings. To isolate feature-specific information, the authors use the Partial Information Decomposition (PID) concepts of redundancy and uniqueness of information and the Wiener-Granger causality principle to find the feature-related flow of information. The authors evaluate the ability of FIT to measure feature-specific information flow using synthetic datasets. They show that FIT performs well in both cases: (1) brain regions encode stimulus with communication; (2) brain regions encode stimulus independently without actual communication occurring between them. Finally, the authors compare FIT with Directed Feature Information (DFI) and evaluate the performance of FIT by three neural recordings spanning the range of electrophysiological recordings (spiking activity, MEG, and EEG).

**Strengths:**

* The problem formulation is clear, giving a clear definition and motivation of the problem the authors want to solve.
* The authors' exposition of their method and experiments is straightforward and comprehensive.
* The proposed method, FIT, can understand how the brain works with complex brain functions and the flow of different types of information. FIT would provide an essential set of contributions to the field.

**Weaknesses:**

* The authors should highlight and summarize the contributions of this work at the end of Section 1.
* The comparison of FIT and Directed Feature Information (DFI) is only performed on simlulated dataset.

**Questions:**

* What will happen if multiple feature-related information is transferred from sender $X$ to receiver $Y$ simultaneously? One may come from stimulus, but the other may come from hidden sources, e.g., other brain regions. Could FIT distinguish different types of information flows that occur simultaneously?
* If possible, could you please discuss the stability of FIT over the increasing dimensionality of data? One major limitation of applying PID to neuroscience is the computational difficulty of estimating PID for high-dimensional neural data.

**Limitations:**

I have highlighted technical limitations and weaknesses above. I have nothing further to add here.

---

> ### Author Rebuttal · Authors · 2023-08-09
>
> Compute DFI on real data.
> We computed DFI on the 3 real-data sets (Fig. R3).
> In the MEG dataset (Fig. R3B), DFI was negative values and thus not interpretable as measure of information transfer. Unlike FIT, DFI could not detect that (as predicted by previous studies) stimulus information is stronger in the feedforward than in the feedback direction, and DFI could not detect that feedforward stimulus information is stronger in correct than error trials (an important result found by FIT).
> In the EEG dataset (Fig. R3A), DFI was negative and thus not interpretable as measure of information transfer. The comparison of the DFI results between eye visibility features and directionally of cross-hemispheric transfer couldn’t support the conclusion (predicted by findings in previous literature and confirmed by the FIT analysis) that across-hemisphere information transfer is directional from contra- to -ipsilateral (DFI does not detected a leading direction of RE information transfer) and is feature specific (DFI does not detected a difference between LE and RE information in the R to L hemisphere communication).
> In the thalamocortical spikes data (Fig. R3C), DFI has mostly positive values which are thus interpretable in terms of information transmitted. DFI confirms (though with lower statistical power) the FIT results than in both the somatosensory and visual corticothalamic pathway more information is transmitted feedforward about the corresponding sensory modality (more visual than somatosensory information transmitted from visual thalamus to visual cortex, and more somatosensory than visual information transmitted from somatosensory thalamus to somatosensory cortex). However, DFI fails to demonstrate that, as expected from well-established neurophysiological findings, more information about such simple stimulus features is transmitted from thalamus to cortex than from cortex to thalamus.
> We will add these results to SM4.2 where we report the properties of DFI.
> In sum, we recognize that the idea behind DFI is sound. However, the imperfect definition of redundancy used in DFI (conflating synergistic and redundant effects) leads to problems which we have already understood mathematically and characterized by simulation in the previous version of our paper. These new analyses of real data, which we will add to the revision, show that the problems predicted by theory and confirmed by simulation are also found in real data, suggesting that DFI is not robust enough to be applied to brain data, and that the advances provided by FIT are important not only conceptually but also for the analysis of empirical datasets.
>
> Study case of information about multiple features transferred simultaneously.
> In the submitted paper we already showed that FIT can reveal encoding of multiple features. Simultaneous encoding of multiple features was indeed investigated in the analysis of all three datasets (MEG, EEG, spikes). In MEG, we found simultaneous encoding and transmission of stimulus and choice information. In EEG, we found simultaneous encoding and transmission of left-eye and right-eye information. We used the relative information values results to individuate the features most encoded and transmitted (visual contrast in the MEG dataset, left eye in the EEG dataset).
> We now performed, as suggested, a simulated study of encoding and transmission of multiple features (Fig R4B). We simulated two independent features (e.g. of a sensory stimulus) S1,S2 simultaneously encoded and transmitted (S1 more strongly than S2):
>
> X=S1+D*S2+Ex
>
> where S1,S2 are independent binary variables (±1), Ex is Gaussian noise with SD=1, and Y equals X with a time lag plus independent Gaussian noise with SD=1.
> We found (Fig R4B) that FIT identifies correctly that both features are transmitted, and ranks correctly the features about which most information is transmitted. These simulations will be added to the SM of the revised paper.
>
> Scaling of FIT with data size and dimensionality.
> We added simulations that study how FIT scales with the number of trials available in the dataset and with the dimensionality of neural activity (here, the number of bins of the discretized activity). We found (Fig R4A) that FIT behaves much better than Shannon information quantities (e.g. TE) with the data size and dimensionality. The correct value of FIT, computed from large data, is achieved already with smaller number of trials than for the Shannon information quantities. We found that accurate calculations of FIT are possible with the number of trials available in empirical datasets (Fig R4A); for comparison FIT calculations in the paper were done with R=2-4). Our understanding is that the better scaling and sampling properties of FIT w.r.t. to Shannon Information quantities arise because FIT considers a PID part of the total information which has lesser bias compared to other parts of the total information. Given that the PID atoms of FIT do not contain synergistic terms, this is in line with previous work (Montemurro et al, Neural Comput 2007) showing that synergistic components of information have much larger limited sampling bias, and that information quantities that do not include synergistic components have much better sampling properties than full multivariate Shannon information quantities. Thus, FIT can be computed from the datasets in which Shannon information measures typically applied to neural data.
> For completeness, we also added to our code the implementation of information theoretic limited sampling bias corrections that slightly aid (Fig R4A) the calculation of information theoretic quantities from real data by subtracting an estimate of the bias computed with a quadratic extrapolation of the scaling of information when subsampling available data (Strong et al, Phys Rev Lett 1998; Panzeri et al J Neurophysiol 2007).
>
> Writing.
> We will highlight and summarize more sharply the contributions of this work.

---

> ### Author Response · Authors · 2023-08-14
>
> Dear Reviewer,
> Thanks again for your insights and suggestions, which we greatly appreciated.
> We wonder whether you received our rebuttal containing the results of the analysis of DFI on real data (confirming the problems with DFI already highlighted in the submitted paper using mathematical considerations and simulation), and the simulations of FIT with multiple simultaneously transmitted features (complementing the results about the transmission  in the same brain regions of information about different features already presented in the submitted paper with real neural MEG/EEG/spike data),  as well as other clarifications and extra analyses in response to suggestions of other reviewers.  The week in which we can interact is getting to a close and we would really appreciate the opportunity to receive your feedback on our work. Thanks so much!

---

> > ### Comment · Reviewer_TXME · 2023-08-19
> >
> > Thank you for the responses and explanation. The paper has improved as a result of authors efforts to address raised questions and concerns. I have upgraded my score to 7.

---

> > > ### Author Response · Authors · 2023-08-21
> > >
> > > Dear Reviewer, thanks again for your valuable insights, which helped us to improve the paper significantly. Thanks also for raising to 7 the score on the basis of the extra work we performed. We are grateful for your support.

---

### Official Review · Reviewer_sTPd · 2023-07-05

**Soundness:** 3 good
**Presentation:** 3 good
**Contribution:** 4 excellent
**Rating:** 6
**Confidence:** 4

**Summary:**

This paper proposes a novel non-parametric method aimed to quantify the amount of brain communication between (time series representing the activity of) brain regions using information theoretic measures. Concretely, the authors’ method builds on the framework of partial information decomposition of the transfer entropy (TE), which is the mutual information between the past of a presumed sending signal X and the present of a presumed receiving signal Y conditioned on its on past. This can be interpreted as the amount of predictive information the past of X contains about the present of Y that goes beyond the information contained in the past of Y itself. TE can therefore be interpreted as an operationalization of the Wiener-Granger causality principle for general non-linear/non-Gaussian data. Based on quantities defined in this framework, the authors now go one step further to quantify only that part of the directed information flow from X to Y (analogous to TE) that contains information of an external content variable S, which could for example be a stimulus in a cognitive neuroscience experiment. Another extension, conditionining the measure on a fourth, potentially confounding variable, is also presented. The authors provide extensive theoretical derivations of their metrics in a long supplement and prove several of its properties. Moreover, the method is demonstrated in a set of simulations as well as on two real electrophysiological neuroimaging experiments.

**Strengths:**

The paper contains an enormous amount of work, which is signified by a 34page supplement. The mathematical derivations appear to be solid and comprehensive. As far as I can tell, the proposed methods are mathematically sound and the mathematical results provide a good fundament for understanding the properties of the methods.  The paper and supplement are well written, and the paper is well organized. The density of the information contained in the paper is very high, and often the reader is referred to the supplement for key information, which is something to reconsider. The paper contains somewhat comprehensive simulations and two applications to real data. The figures are of very high quality.

**Weaknesses:**

Presentation-wise I do not consider it not so helpful to put all technical details in the supplement. Quantities like shared information and FIT itself are not defined in the main body, which makes it difficult to follow the theoretical arguments. Other technical details seem to be missing completely, at least I could not find them. This concerns the calculations of the various information theoretic metrics themselves.
My main objection is that the proposed metrics are affected if the measured data are mixtures of underlying sources. This is known to be the case for electrophysiological recordings like MEG and EEG. Mixing occurs due to the propagation and superposition of electrical currents and magnetic fields through the head from the brain sources to the sensors. It is most highly pronounced when analyzing data on the M/EEG sensor level (as apparently done in the section on “Eye-specific interhemisperic information flow during face detection”). Also invasive electrophysiological recordings can exhibit strong artifacts of source mixing, if different channels are recorded relative to a common electrical references. This can lead to spurious estimates of information transfer for measures based on the concept of Granger causality (GC). A simple example to illustrate this is a single brain source activity that is measured in two channels due to source mixing. That alone would not cause Granger causality. However, if both recorded channels are affected by (to some extent) independent noise, (spurious) GC emerges, the reason for this being that past of both channels together contains more information about the present than the past of any single channel alone, since noise can be better averaged out by combining channels. Not that this behavior is neither overcome by GC’s property to model time-delayed interaction nor by conditioning on the past of the receiver.
In the presence of source mixing, GC is mainly driven by asymmetries in the mixing proportions of different signals and not so much about their interactions. These mixing proportions depend on factors such as the choice of the reference electrode.
The authors are aware of the issue as they mention: “because they were sufficiently far apart to avoid leakage in source reconstruction [57].”. However, it is hard to validate this claim, and I consider it rather unlikely that different sources in the visual system are sufficiently far apart (if that is even possible). Such assessment would also depend on certain parameter choices of the inverse model, which are currently not provided.
I strongly urge the authors to provide some theoretical and empirical evidence for the behavior of their methods in mixed signal or mixed signal settings. Specifically suggest to study the null case of only one source with independent noise: X = Z + E1 ; Y = Z + C*E2 with coefficient  parameter C and independent noise E1 and E2. As well as the case of two interacting signals Z1 and Z2 with FIT(Z1 -> Z2) > 0, where instead of Z1/2 mixed signals X = Z1 + A*Z2 and Y = B*Z1 + B. I suggest to study this case as a function of the parameters A and B with |A| < 1 and |B| < 1.
Language-wise I found it slightly odd to put some parts of the methods description into past tense (lines 80ff for example). Also some sentences seem to be rather long, see for example line 335ff.


**Questions:**

-	How were the information theoretic quantities calculated? Was this done by constructing histograms? What were the dimensions of the different distributions? How many bins were used and can the authors comment whether the amount of available data was sufficient to estimate these quantities? Or were some kinds of parametric distributions fitted and the information theoretic measures were evaluated analytically?
-	What were the parameters of the source reconstruction in the MEG/EEG examples? Which inverse methods were used, what regularization parameters? How were the forward models calculated? Was source reconstruction done with free orientation dipolar sources or orientation fixed perpendicular to the cortex? How many source dipoles were modeled? How were sources aggregated into regional time series. How many time series per region? What was the dimensionality of the data in each step of the processing chain?
-	Line 318: the study [4] seems to be relevant but is almost not discussed. Why is it so important that the metrics “should be upper bounded by either feature information encoded in the past of the sending region or the total information flowing between regions.” What is the practical disadvantage if this is not the case?
-	What is the relevance of the theoretical properties of FIT shown in the supplement ( in sections SM1.3.4ff)? What would be the practical implications if some of these properties would not hold?
-	P195ff: What are the references for these statements (e.g. “area V3A (carrying maximal stimulus information in the dorsal stream visual cortex)”) ?



**Limitations:**

Despite the comprehensive theoretical content the paper is lacking some technical details which should be provided in a revision.
The main limitation for me is the unclear behavior of the proposed methods in the presence of mixed signals. While the paper is overall of high quality, I see a necessity to reduce my score if the issue is not adequately addressed/discussed in a revision.

---

> ### Author Rebuttal · Authors · 2023-08-09
>
> Asymmetric source mixing.
> We simulated source mixing in different proportions in X and Y due to field spread/ common reference. Such source mixing in real cases is (and is assumed to be in our simulations) instantaneous (i.e. zero lag) and with a stable (across time) proportion of source sharing in X,Y.
> We simulated a source Z (informative about a stimulus feature S) shared between X and Y with a different proportion A:
>
> X= Z(s) + Ex ; Y=A*Z(s)+ Ey
>
> with Ex, Ey indep. Gauss. noise. This is the “null model” proposed by the Referee, but we controlled the SNR of X,Y by changing A (which sets the relative level of stimulus signal in X,Y) and fixing noise SD to 1. As predicted, on this model FIT and TE had spurious positive values (Fig R1B). In the submitted paper, we used a null-hypothesis test (randomly permuting trials with the same stimulus feature value) for spurious values induced by X,Y covariations due to stimulus-signal sharing. Here, this test correctly rules out that the from the null model’s FIT and TE are generated only by source sharing with no real transmission (Fig R1B). We already reported in the submitted paper that the real-data FIT were statistically significant with this test.
> Importantly, analysis of the null model also shows that with instantaneous source mixing the ratio between stimulus info in X and Y is constant in time (Fig R1A). This gives a useful heuristic: different timecourses of stimulus info in X vs Y cannot be explained by instantaneous source mixing. We measured all real-data FIT in cases with a delay in stimulus info latencies between X and Y (X to Y info latencies: MEG: 17-35ms between V1 and higher areas, Fig R2C. EEG: 25ms across hemispheres (Fig 4B). In spike data: 20ms from thalamus to cortex, Fig S8). Overall, these findings speaks against dominant mixing of a stimulus-informative source in our analyses.
>
> Finally, as suggested we simulated the case with real FIT between two “pure” signals Z1 and Z2 that are unevenly mixed in the measured X,Y:
>
> X = Z1 + AZ2 + Ex;   Y = Z2 + BZ1 + Ey
>
> Since adding a new stimulus-informative channel (Z1 to Y and Z2 to X) increases the stimulus information in X,Y, we set SD of indep. Gauss. Noise Ex,Ey to equalize SNR of X and Y across the simulated parameters space. We found (Fig R1C) that mixing (A,B>0) reduced FIT and TE compared to the pure case (A=B=0). However, the correct directionality of info transfer was always detected for all mixtures. Thus, FIT is reasonably conservative and robust to this mixing.
>
> Info computation details.
> We will bring into the main text more theory details about PID and FIT. See General and Ref. LrX8 rebuttal for our plan.
> Information was computed by discretizing neural activity into equipopulated bins (3 bins for simulations, see ll 137-8; 2-4 bins for real data, SM p. 21-24). These discretized estimators are widely used in neuroscience and PID was developed mostly for discrete distributions. Apologies if this was unclear. We will add to SM a dedicated Methods section and a Table with the number of bins used for each figure panel.
> We now show in a new simulation that the number of trials we used in simulations and real data analyses is enough for good estimates (see Fig R4 and rebuttal to Ref. TXME).
>
> Importance that FIT satisfies proven info-theoretic bounds.
> That FIT satisfies such bounds is essential to interpret FIT values as transmitted information. If FIT can be larger than the feature information encoded by sender X or receiver Y or than the total information transmitted (TE X→Y), then FIT cannot be interpreted as stimulus feature info transmitted X→Y. If e.g. FIT>I(S;X_past) then some of the stimulus feature info in FIT cannot have been transmitted X→Y. These bound also allow quantifying FIT in meaningful relative terms. They allow e.g. to interpret the ratio FIT(X→Y)/I(S;X_past) as the proportion of stimulus info encoded in past of X that gets transmitted to Y. This will be explained in the revised paper.
>
> Source mixing in MEG/EEG data.
> We used published MEG/EEG data and we kept to the published preprocessing, to avoid introducing changes which confound comparison of FIT with published work. We referred to the original papers for full info, but we provide below (and will add to the SM) some details.
>
> For MEG:
> Source reconstruction: LCMV beamformers based on: (i) leadfield matrices from 3-layer boundary element head-model (conductivity 0.3, 0.3, 0.006 S/m for scalp, brain, skull) based on individual MRIs; (ii) covariance matrix (CM) of broadband data (275x275 sensors); (iii) regularization of 5% of CM. Source space constrained to cortical sheet with 4096 vertices per hemisphere, and source orientations chosen to maximize power at each vertex. To illustrate spatial resolution, we plot correlation between LCMV spatial filters of neighboring sources vs distance, finding <0.02 correlations at d>2.5cm (Fig R2B). To compute FIT & TE, time-frequency repr. of sensor data was projected into source space and averaged over vertices within the ROI (80,20,10 vert. for V1, V3A, LO3). Although our FIT/TE analyses were all correct, we identified an error in rendering ROIs in Fig 3B. Apologies. The correct visualization is in Fig R2A. We also apologize for the over-statement about no leakage, which we will rectify. Leakage of MEG source estimates drops greatly at >2cm for realistic SNR (Gross et al, Neuroimage 2003). We thus ran a more conservative analysis only on V1, V3A, LO3, visual ROIs with high stimulus info (Fig R2C) that are >2.8cm apart. Results (stronger feedforward stimulus transmission) are fully confirmed in this network (Fig R2D-F). Since here real-data analyses serve more for validation than for discovery, we would present the more conservative analysis in the revised paper. We appreciate feedback on this.
>
> For EEG:
> We used sensor data. Distance between electrodes LOT, ROT for computing inter-hemispheric transfer was >10 cm, suggesting that source mixing may be small.

---

> > ### Comment · Reviewer_sTPd · 2023-08-16
> > **Thanks for the clarifications although I am not convinced about some**
> >
> > I would like to thank the authors for the thorough rebuttal. I am mostly satisfied and I would suggest that specifically the technical details of the M/EEG data analyses be added to the manuscript.
> >
> > There are two major points that still do not convince me.
> >
> > 1. The authors use a null-hypothesis test that randomly permutes trials with the same stimulus feature value that is supposed to test for spurious values induced by X,Y covariations due to stimulus-signal sharing. They report that this test correctly rules out "that the from the null model’s FIT and TE are generated only by source sharing with no real transmission". However, I am not convinced that this test can separate interactions between X and Y that are just due to instantaneous mixing from genuine time-delayed interactions between X and Y (taking the issue of stimulus dependence aside). By permuting trials of X and Y, all statistical associations between X and Y are destroyed. This includes those introduced by instantaneous mixing of otherwise independent sources. The test thus seems too liberal, meaning it is too easy to reject the null hypothesis also for mixed independent signals. Or, in other words, the null hypothesis (X and Y independent) is too unrealistic. A more realistic hypothesis would be that X and Y are in fact mixtures of independent signals (see [1]). I agree that precautions of the authors (selecting mutually far away regions) could be the reasons for the results to come out as expected. But I would like to clarify the role of the statistical approach.
> >
> > 2. The authors state "Importantly, analysis of the null model also shows that with instantaneous source mixing the ratio between stimulus info in X and Y is constant in time" . I am surprised by this result as I believe that the stimulus information in X and Y depends on the SNR which could be highly volatile. EEG and MEG measure macroscopic brain signals composed of many individual components (such as different rhythms), and the amplitudes of these components are strongly fluctuating over time and with task. Assuming that some of the activities are informative with rescect either to each other or the stimulus, and some don't, the SNR is constantly changing. An example would be if someone closes there eyes or moves their leg for a second. This would lead to substantial changes in the power of brain rhythms in the alpha and mu band. And even if these actions would be unrelated to the stimulus, this would alter the SNR of stimulus related information in the data in the same way as a change in the mixing coefficients would.
> >
> > [1] Shahbazi, F., Ewald, A., Ziehe, A., & Nolte, G. (2010). Constructing surrogate data to control for artifacts of volume conduction for functional connectivity measures. In 17th International Conference on Biomagnetism Advances in Biomagnetism–Biomag2010: March 28–April 1, 2010 Dubrovnik, Croatia (pp. 207-210). Springer Berlin Heidelberg.

---

> > > ### Author Response · Authors · 2023-08-17
> > >
> > > We are glad that the Reviewer is mostly satisfied by our additional analyses.
> > > As stated in Rebuttal, we will add the details of M/EEG analyses to the paper.
> > >
> > > Thanks for the additional suggestions.
> > >
> > > (1) RE: “by permuting trials of X and Y, all statistical associations between X and Y are destroyed, including those introduced by instantaneous mixing of otherwise independent sources”. Our permutations do not shuffle trials and/or time points indiscriminately. The permutations create surrogate data associating the entire time series of X within a given trial to the entire time series of Y within another randomly chosen trial to the same stimulus feature value. They generate surrogate data that fully preserve auto-correlation in X and Y due to stimulus-info time variations and to autocorrelated noise, stimulus info in X and Y at each time point, and inst. correlations due to similarity of stimulus tuning of instantaneously mixed stimulus informative sources. What they destroy (besides the genuine time-lagged communication between X and Y) is the inst. noise correlations due to inst. mixing of non-stimulus informative sources. Having verified that the value of FIT (see below) is unaffected by the strength of inst. noise correlation between X and Y (because FIT’s PID discards any part of X,Y info that is not about the stimulus feature), the permutation surrogate provides a reasonable initial test of genuine time-lagged communication in the presence of autocorr. in X and Y, stimulus info in X and Y, and inst. correlations due to source mixing (inst. corr. due to stimulus signal mixing are maintained, inst. corr. due to noise mixing are destroyed but do not affect FIT). Apologies if this was not clear. We will clarify it in revision.
> > >
> > > We showed in Rebuttal (Fig R1B) that the surrogate permutation is good for the null model of one informative mixed source.
> > > Following the Reviewer’s new suggestion, we now simulated a new null model in which X and Y are linear mixtures of the same sources with possibly unequal mixing weights between X and Y. Assuming that each source has a stimulus-driven component (which is 0 for non-stimulus informative sources) plus own indep. additive noise, this can be compactly written as
> > > X=F(s) + Nx +Ex
> > > Y=G(s) + Ny +Ey
> > > where F and G are the stimulus tuning functions of X and Y (sum of the stimulus-driven components of all stim-informative sources), Nx and NY are source-mixing noise of X and Y (sum of the stimulus-unrelated components of all sources), and Ex, Ey are indep. noise. Nx, Ny can be instantaneously correlated due to source mixing. We varied across simulations the level of independence between sources  (from partial to full) . Making the sources more independent from each other and/or making source weighing more different between X and Y decrease the inst. noise correlation between Nx and Ny, and also makes the functions F and G more dissimilar.
> > >
> > > Simulating this system, we found that (as explained above) FIT does NOT depend on the strength of inst. noise correlation between X and Y (unlike TE, which does), and that the spurious values of FIT are reduced when F and G are more dissimilar (because there is less redundant stimulus info between past of X and present of Y). Importantly, because FIT does not depend on the strength of the inst. noise correlation (destroyed in the permutation surrogates), for all tested simulations (including those with major mixing strength diff. between X and Y), the permutation test correctly classified these spurious FIT values as non-significant. Thus, the permutation surrogates can reasonably cover this new case.
> > >
> > > We think it is useful to offer readers a simple surrogate permutation test for significance assessment. However, we agree that no surrogate data is perfect.  In the revised paper, we will add the above simulations and clarifications, and carefully avoid overstatements. We will highlight the assumptions and limitations of the simple surrogate permutation, and we will cite [1] as a way to construct more conservative surrogates.  We hope that this effort will contribute to promote more frequent and careful use of surrogates. Such use is currently uncommon in neuroscience and there is no single commonly accepted surrogate generation.
> > >
> > >
> > > (2) RE: We agree that stimulus info timecourse difference in X and Y does not rule out genuine stimulus communication when noise in X and Y is not stationary it was stationary in our Rebuttal’s null model). We will clarify this in revision. Note that we think that examining differences in info timecourses between sites can be used as sanity check to flag possibly spurious FIT values, but NOT to prove/disprove real communication. A time-invariant ratio of stimulus info in X and Y would make it difficult to trust a positive value of FIT as real (it would require contrived scenarios, e.g. level of noise in one site proportional at each instant to the amount of stimulus info transmitted across sites).

---

> > > > ### Author Response · Authors · 2023-08-17
> > > >
> > > > We apologize for the typo at the end of our above comment.
> > > >
> > > > The sentence
> > > >
> > > > “RE: We agree that stimulus info timecourse difference in X and Y does not rule out genuine stimulus communication when noise in X and Y is not stationary it was stationary in our Rebuttal’s null model)”
> > > >
> > > > contained a misprint and should be corrected as:
> > > >
> > > > “RE: We agree that the invariance over time of the ratio of the stimulus info in X and Y does not rule out genuine stimulus communication when noise in X and Y is not stationary (noise was stationary in our Rebuttal’s null model)”

---

> > > > > ### Author Response · Authors · 2023-08-21
> > > > >
> > > > > Dear Reviewer, as the interactive period is ending, we take the opportunity to thank you for your suggestions and insights, which helped us to improve our work.

---

> ### Author Response · Authors · 2023-08-14
>
> Dear Reviewer,
> Thanks again for your insights and suggestions, which we greatly appreciated. We wonder whether you received our rebuttal containing the results of the two requested simulations on source mixing, the reanalysis of MEG data with more conservative inter-ROI distances, the details and evaluation of spatial resolution of the MEG beamformer, the requested numerical investigation of the behavior of the FIT measure with number of trials and dimensionality of the neural response space,  as well as other clarifications and extra analyses in response to suggestions of other reviewers.  The week in which we can interact is getting to a close and we would really appreciate the opportunity to receive your feedback on our work. Thanks so much!

---

### Official Review · Reviewer_LrX8 · 2023-07-26

**Soundness:** 4 excellent
**Presentation:** 3 good
**Contribution:** 3 good
**Rating:** 7
**Confidence:** 3

**Summary:**

The submission proposes a measure called Feature-specific Information Transfer (FIT) which can be used to partial out information transmitted from a sender to a receiver (in this case, two brain regions) about a specific feature (another variable) in a casual way (i.e. it is not present in the history of the receiver. It does so using Partial Information Decomposition (PID), a new way of decomposing mutual information into shared, unique and synergistic components. Basic properties of the measure are established and validated theoretically and by simulation, and it is then applied to analyze three neural datasets of varying modalities, yielding both some recovered sanity checks and novel analysis results.

**Strengths:**

The paper leads with a very strong exposition, which explains exactly what it is doing and why. The overall reasoning about the desiderata of FIT are likewise clear and come at the right pacing in the paper. The simulation and experimental results are comprehensive and show strong benefits over the prior TE approach.

**Weaknesses:**

I think that the paper shunts too much important content to the appendix. PID and related concepts are likely to be less familiar to the NeurIPS audience, and the paper does not give much intuition on why various claims hold beyond a reference to the extensive supplement. Given the audience, I suspect that pushing more detail on the neuroscience data and experiments to the supplement in favor of bringing more theoretical meat to the main text would make for a stronger paper. Even if much of the detail remains in the supplement, papers such as this can benefit from providing at least intuitions or sketches as to why some properties might hold.

I also found minor typos:
* l40 features-specific -> feature-specific?
* l102 what is $Y_t$? Is it intended to be $Y_{pres}$, or am I misunderstanding something here?

**Questions:**

Can you explain in more detail why $SUI(S: \{X_{past}, Y_{pres}\setminus Y_{past}\})$ can be higher than TE, motivating the definition in expr. 2? The paper makes the claim around line 93 and makes a reference to the bottom of Fig1A. I imagine it is true algebraically if one works it out but I'm not sure I see where in the figure this is explained or why it should be true intuitively.

**Limitations:**

The paper does a good job to discuss the computational limitations of actually computing FIT and how it limits the applicability to small numbers of regions and time points.

---

> ### Author Rebuttal · Authors · 2023-08-09
>
> The paper shunts too much important content to the appendix.
>
> We agree that it would be better to bring to the main text more theoretical details about the PID used to compute FIT. In case of acceptance, we will use the extra page allowed by NeurIPS to include more PID theoretical details about FIT derivation and properties. This will include spelling out the definition of shared information, and moving to main text the Figure with the PID lattices, currently Fig S1 (useful to provide immediate intuition of why the upper bound properties of FIT hold).
> If needed to free up sufficient space, as suggested by the reviewer we will push more details about the neuroscience data analysis validation on the SI (e.g. moving current Fig 4 to the SM).
>
> Minor typos.
>
> “l40 features-specific -> feature-specific?.” Thanks, we will correct it.
> “l102 what is $Y_{t}$? Is it intended to be $Y_{pres}$?”. This was a misprint. We intended to write $Y_{pres}$, as the Reviewer correctly guessed. We will correct the misprint in revision.
>
> Why $SUI(S:X_{past},Y_{pres} \backslash Y_{past})$ can be higher than TE, motivating the definition in expr. 2.
>
> We agree that the reference to the bottom of Fig 1A contained in line 93 of the first submission was unclear. In the revision we will remove this and we will reference to Fig S1 (which will be moved to the main text, as mentioned in the reply above).
> The fact that the first PID atom $SUI(S:X_{past},Y_{pres} \backslash Y_{past})$ can be larger than TE was derived formally from the algebra of Eq. S7. In the PID decomposition by construction the unique information is conceptually defined to be a component of (and hence equal or smaller than) the conditional mutual information about the target. However, the target in $SUI(S:X_{past},Y_{pres} \backslash Y_{past})$ is the stimulus feature S, which means that this atom is not constrained to be smaller than the Transfer Entropy $I(X_{past}; Y_{pres}| Y_{past})$, which is not defined in terms of S. Fig S1C illustrates the absence of a relationship between the PID atom $SUI(S:X_{past},Y_{pres} \backslash Y_{past})$ and TE, in which the yellow quantity TE cannot be mapped on the left lattice having S as a target. This was briefly explained in the SM after eq. S23 of the original submission and will be emphasized more in the revised paper, in case of acceptance.
>
> The issue of redundancy about a target potentially exceeding the information between the sources is discussed in the PID literature (Harder et al, Phys Rev E 2013; Pica et al, Entropy 2017). The intuition of why this can happen is because the mechanisms connecting the target with the sources can induce a non-zero component of redundancy (known as mechanistic redundancy), independently of the correlations between the sources. That is, redundancy is created due to the similarity in the mechanisms linking the target and the sources, even for independent sources. Pica et al (Entropy 2017) developed a formalism relating PID atoms with different targets to quantify mechanistic redundancy and isolate the component of redundancy that is already manifested in the mutual information between the sources (termed source redundancy). We built on this formalism to guarantee that FIT only considers a component of source redundancy about S carried by $X_{past}$ and $Y_{pres}$ that is unique with respect to $Y_{past}$.

---

> ### Author Response · Authors · 2023-08-14
>
> Dear Reviewer, Thanks again for your insights and suggestions, which we greatly appreciated. We wonder whether you received our rebuttal containing the clarifications you requested and the response to your suggestions for improving clarity, as well as other clarifications and extra analyses in response to suggestions of other reviewers. The week in which we can interact is getting to a close and we would really appreciate the opportunity to receive your feedback on our work. Thanks so much!

---

> > ### Comment · Reviewer_LrX8 · 2023-08-19
> > **Thank you**
> >
> > Dear Authors -- thank you for the detailed rebuttal and clarifications. I maintain my favorable rating and hope to see the paper at the conference.

---

> > > ### Author Response · Authors · 2023-08-21
> > >
> > > Dear Reviewer, thanks again for your valuable insights, which helped us to improve the paper significantly. Thanks also for your words of appreciation for our work. We are grateful for your support.

---

### Author Rebuttal · Authors · 2023-08-09

We are grateful to the Reviewers for their suggestions and insights. We feel that the new results we obtained in addressing their suggestions (Figs R1-4) significantly elevate the level of conceptual advance provided by our paper. These new results will be included in the revised paper in case of acceptance.

The advances from the new work, and the changes we wish to make to the paper, are summarized below. More details can be found in the replies to the individual Reviewers.

Methods development and validation:

Mixing of sources.
We developed new simulations to evaluate the effects of asymmetric mixing of sources (which can happen in real neural data because e.g. of field spread). First, we developed a simulation of the effect of mixing a stimulus-informative source with unequal proportions in the empirically measured signals X,Y. We found that in such case, as predicted, both FIT and TE can have spurious positive values (Fig R1B). However, we also found that the permutation test that we developed and used for FIT in the submitted paper (shuffling trials with the same value of the stimulus feature) to discount the effect of common covariations in X, Y due to stimulus-signal sharing can also be used to rule out this confounder of mixing a stimulus-informative source (Fig R1B). Importantly, all previously reported real-data and FIT values were already tested against this null-hypothesis, suggesting that our FIT results cannot be explained by this artefact. Our analysis also shows that with instantaneous source mixing the ratio between stimulus information in X and Y is constant in time (Fig R1A). This provides an important heuristic to accept empirical results: data with different time course of stimulus info in X vs Y cannot be explained by instantaneous source mixing. Importantly, all the FIT real-data results in the paper are obtained in presence of a latency difference of >10ms between encoding of stimulus information in the putative sender and putative receiver (e.g. Fig R1B), incompatible with dominant mixing of a stimulus-informative source.
Finally, we simulated the case in which there is real FIT between two “pure” signals Z1 and Z2, but Z1,Z2 are unevenly mixed in the measured X,Y. We found (Fig R1C) that mixing reduced FIT and TE compared to the pure case. However, the correct directionality of info transfer was always detected for all mixtures. Thus, FIT/TE seems conservative and robust to this mixing. To our knowledge, this case was not considered in previous GC/TE literature, so this extra analysis provides progress beyond FIT relevant for the TE literature.

More conservative MEG analysis.
Models of source separation indicate that in MEG mixing becomes negligible at distances > 2cm. We thus run a more conservative MEG analysis in a network of visual areas that are > 2.8cm apart, reducing possible source mixing. All previously reported FIT results are fully confirmed in this more conservative analysis (Fig R2C-F).

Scaling of FIT with data size and dimensionality.
We computed how FIT scales with the size of the dataset and the dimensionality of neural activity, and compared it with the behavior of Shannon information quantities such as TE. We found that FIT behaves much better than TE (smaller limited-sampling bias, less data needed for the accurate estimations, more robust to the curse of dimensionality), see Fig R4A. This is because FIT does not include synergy terms which, according to previous literature, are the most biased and data-hungry ones. We also now implemented in the FIT numerical routine the limited-sampling bias corrections of the neural information theory literature, which further help.

Comparison with DFI on real data.
We now computed DFI, a previously proposed measure of feature-specific information transmission, also on real data. We found (Fig R3) that the problems with DFI predicted in the previous version of our paper by mathematics and simulations are also found in the neural datasets. On real data, DFI was very often negative and it didn’t detect directionality or feature specificity in cases in which we expect from previous literature that specificity or directionality should exist.

Study information about multiple features transferred simultaneously.
In the submitted paper we already showed on real data that FIT can reveal encoding of multiple features. Simultaneous encoding of multiple stimulus features was investigated in all three datasets (MEG, EEG, spikes) and was found in two datasets (MEG, EEG). However, in the submitted paper we did not demonstrated this property with simulations. We now performed a simulated study of encoding and transmission of multiple features (Fig R4B). We simulated two independent stimulus features simultaneously encoded and transmitted (one more and one less strongly). We found that FIT identifies correctly that both features are encoded and transmitted, and ranks correctly the features about which most information is transmitted. These simulations will be added to the SM of the revised paper.

Writing and presentation:

We will bring more details of the FIT maths (shared and unique information definition, PID lattices, mathematical properties of FIT) from the SI to the main text. We will use the extra page allowed for the FIT camera-ready paper. If space is a problem, we will move the EEG analysis figure to the SM.

We will be clearer on how we computed the information quantities (by discretizing data, computing histograms). This information was present yet not prominent or well organized in the submitted paper.

We all add a SM section which explains this clearly and which contains a Table with the number of bins used for each (simulated or real) dataset and Figure panel. We will add to SM3.1 a description of the requested MEG source localization (summarized from the original data publication which reports full details).

We will better define what we mean by “feature”.

---

### Decision · Program_Chairs · 2023-09-21

**Decision:**

Accept (poster)

**Comment:**

This paper is a methodological and applied data analysis contribution that proposes and empirically tests a novel approach for quantifying the information content of communication between brain regions.
The resulting contribution received unanimously positive reviews, with reviewers praising the clarity of the exposition, the soundness of the mathematical derivation, and the completeness of the simulation and experimental results.
Reviewers also emphasized the contribution of this work by mentioning its potential impact in applied branches of neuroscience as a method that could help elucidate complex brain functions by teasing apart the flow of different types of information.